# The AMPK-Sirtuin 1-YAP axis is regulated by fluid flow intensity and controls autophagy flux in kidney epithelial cells

Aurore Claude-Taupin [1] ✉, Pierre Isnard[1], Alessia Bagattin[1], Nicolas Kuperwasser[2], Federica Roccio[1], Biagina Ruscica[1], Nicolas Goudin [2], Meriem Garfa-Traoré[2], Alice Regnier[1], Lisa Turinsky [1], Martine Burtin[1], Marc Foretz [3], Marco Pontoglio [1], Etienne Morel [1], Benoit Viollet [3], Fabiola Terzi[1,4], Patrice Codogno [1,4] & Nicolas Dupont [1,4] ✉

Shear stress generated by urinary fluid flow is an important regulator of renal function. Its dysregulation is observed in various chronic and acute kidney diseases. Previously, we demonstrated that primary cilium-dependent autophagy allows kidney epithelial cells to adapt their metabolism in response to fluid flow. Here, we show that nuclear YAP/TAZ negatively regulates autophagy flux in kidney epithelial cells subjected to fluid flow. This crosstalk is supported by a primary cilium-dependent activation of AMPK and SIRT1, independently of the Hippo pathway. We confirm the relevance of the YAP/TAZ-autophagy molecular dialog in vivo using a zebrafish model of kidney development and a unilateral ureteral obstruction mouse model. In addition, an in vitro assay simulating pathological accelerated flow observed at early stages of chronic kidney disease (CKD) activates YAP, leading to a primary cilium-dependent inhibition of autophagic flux. We confirm this YAP/autophagy relationship in renal biopsies from patients suffering from diabetic kidney disease (DKD), the leading cause of CKD. Our findings demonstrate the importance of YAP/TAZ and autophagy in the translation of fluid flow into cellular and physiological responses. Dysregulation of this pathway is associated with the early onset of CKD.

The development and function of organs is tightly controlled by mechanical forces[1,2]. In the kidney, mechanical forces, such as shear stress induced by urinary fluid flow, are the principal regulators of proximal reabsorption. Indeed, forces regulate active transport mechanisms in the proximal tubule to drive the reabsorption of up to 70% of $Na^+$, $K^+$, $H^+$, $NH_4^+$, $Cl^-$, $HCO_3^-$, $Ca^{2+}$, inorganic phosphate, low-molecular-weight proteins, glucose and water from the glomerular ultrafiltrate into the blood[3]. Altered tissue mechanics are now recognized to have an active role in driving human diseases[4–7]. However,

how mechanical alterations participate in the early stages of renal diseases, such as chronic kidney disease (CKD), is still unclear.

In kidney proximal tubules, the primary cilium, a microtubule-based organelle present at the apical surface of epithelial cells, acts as a flow sensor to integrate variations in flow rates of the glomerular ultrafiltrate. We and others have demonstrated an interplay between macroautophagy (hereafter referred to as autophagy) and primary cilia[8–10]. Recently, we showed that primary cilium-dependent autophagy is activated through an LKB1-AMPK signaling pathway to regulate

[1]Université Paris Cité, INSERM UMR-S1151, CNRS UMR-S8253, Institut Necker Enfants Malades, F-75015 Paris, France. [2]Structure Fédérative de Recherche Necker, US24-UMS3633 Paris, France. [3]Institut Cochin, Inserm U1016 - CNRS UMR8104 – Université Paris Cité, 75014 Paris, France. [4]These authors jointly supervised this work: Fabiola Terzi, Patrice Codogno, Nicolas Dupont. ✉e-mail: aurore.claude-taupin@inserm.fr; nicolas.dupont@inserm.fr

metabolism and cell volume of kidney epithelial cells[11,12]. Autophagy is a catabolic process involved in the maintenance of cellular homeostasis. This self-eating mechanism is initiated by the formation of a double-membrane delimited vacuole, the autophagosome, which ultimately fuses with lysosomes, in order to regulate the turnover of cytoplasmic material, including proteins, lipids and organelles[13,14]. Autophagy is tightly coordinated by AuTophaGy-related (ATG) proteins, which can be regulated by transcriptional and post-transcriptional mechanisms[15].

Yes-associated protein 1 (YAP1; hereafter referred to as YAP) and WW-domain-containing transcription regulator 1 (WWTR1; hereafter referred to as TAZ) are transcriptional coactivators that respond to multiple microenvironmental inputs, including shear stress[16–22]. YAP/TAZ shuttle between the cytosol and the nucleus where they interact with transcription factors, including the TEA domain family member (TEAD) transcription factors, to induce the expression of key target genes involved in cell proliferation, survival, differentiation or organ size regulation[23]. The co-transcriptional activity of YAP/TAZ is regulated by the Hippo pathway (consisting of the MST1/2 and LATS1/2 kinases) in response to various signals, such as adhesion proteins or growth factors. Hippo-independent inputs, corresponding to mechanical stimuli such as cell density, stretching or fluid flow also regulate YAP/TAZ activity[24]. Recently, accumulating evidence has uncovered an interplay between autophagy and the YAP/TAZ pathway[25–30]. In addition, these pathways have been associated with CKD, such as in diabetic kidney diseases[31,32]. However, to our knowledge, the crosstalk between autophagy and the YAP/TAZ pathway has never been reported in the context of kidney physiology and pathophysiology.

Here, we investigated the crosstalk between these two pathways in the context of shear stress in kidney epithelial cells. We show that YAP/TAZ inactivation by physiological fluid flow is mandatory for the activation of autophagy flux in proximal tubule kidney epithelial cells. We demonstrate that a primary cilium-dependent activation of AMPK promotes YAP nuclear exit, relying on Sirtuin 1 (SIRT1), to inhibit the expression of the autophagy repressor Rubicon, a suppressor of autophagosome maturation[33,34]. Moreover, AMPK-dependent phosphorylation of YAP contributes to its retention in the cytoplasm. Lastly, we show the relevance of this interplay in CKD, revealing a potential role of the YAP-autophagy axis dysregulation in renal pathology.

## Results

### YAP/TAZ activity is inhibited by physiological fluid flow
We previously reported that physiological fluid flow stimulates autophagy to direct a metabolic adaptation of proximal tubule kidney epithelial cells (KECs)[11,12]. As YAP/TAZ activity can be regulated by different patterns of blood flow[16], we investigated whether a physiologically relevant constant laminar flow (1 dyn.cm$^{-2}$) could impact YAP/TAZ activity in KECs. We observed a re-localization of YAP and TAZ into the cytoplasm of KECs submitted to physiological shear stress (Fig. 1a–d). YAP/TAZ are co-transcriptional regulators modulating the activity of TEAD transcription factors. Therefore, we used the 8xGTIIC luciferase reporter, previously described as a YAP/TAZ activity reporter[35], to study YAP/TAZ activity upon shear stress. We observed a reduced transactivation activity of YAP/TAZ in cells subjected to fluid flow, as shown by reduced 8xGTIIC luciferase reporter gene activity (Fig. 1e) and downregulation of YAP/TAZ target genes (Fig. 1f, g). YAP/TAZ was also observed in the cytoplasm of cells cultivated under static conditions (Supplementary Fig. 1a–d). This was confirmed by increased levels of YAP phosphorylation at S397 (Supplementary Fig. 1e, f). However, the decrease in YAP/TAZ activity was more important when KECs were subjected to fluid flow than under static conditions. Of note, it has been previously shown that shear stress does not induce cell proliferation but a reduction in cell size[11,36]. The effect of shear on YAP/TAZ activity was also observed in proximal

tubular human kidney epithelial cells (HK-2) (Supplementary Fig. 1g–k). This suggests that an active mechanism inhibiting YAP/TAZ is dependent on the mechanical forces exerted by unilateral fluid flow. Indeed, the expression of other YAP/TAZ target genes (*Ptplad2* and *Cptp*), previously reported as being inhibited by fluid flow[21], were also downregulated in KECs subjected to shear stress (Supplementary Fig. 1l, m). Thus, under physiological fluid flow conditions, YAP/TAZ are retained in the cytoplasm and are transcriptionally inactive in renal epithelial cells.

### Autophagy flux is regulated by YAP/TAZ upon fluid flow
To determine the role of YAP/TAZ activity in fluid flow-induced autophagy, we investigated autophagy in KECs after knocking down the expression of either YAP or TAZ. Of note, the double YAP/TAZ knockdown affected cell viability and proliferation. Physiological fluid flow stimulates autophagy in KECs[11,12], as confirmed by the increase of endogenous LC3 puncta in cells upon shear stress (Fig. 2a–d). The knockdown of YAP (Fig. 2a, b) or TAZ (Fig. 2c, d) increased the accumulation of LC3 positive structures in the presence of the lysosomotropic agent chloroquine (CQ), indicating that autophagy flux is amplified in KECs under flow. Accordingly, the accumulation of LC3-II, the lipidated form of LC3 associated with autophagosome membranes, was further increased in cells treated with CQ after YAP or TAZ inactivation (Fig. 2e–h and Supplementary Fig. 2a–d). This was also observed in human renal epithelial cells (Supplementary Fig. 2e–h). The effect of YAP/TAZ on autophagy flux was further confirmed using cells expressing the mRFP–GFP–LC3 tandem probe (Fig. 2i, j). The loss of YAP or TAZ increased the number of autolysosomes (GFP negative, mRFP positive LC3 puncta) without affecting the number of autophagosomes (double positive mRFP-GFP LC3 puncta). Accordingly, the formation of double-positive ATG16L1-WIPI2 structures, known to be engaged in the formation of autophagosomes was not affected by the loss of YAP or TAZ (Supplementary Fig. 2i, j). These data confirm that YAP and TAZ individually play a role in inhibiting autophagic flux in cells subjected to shear.

Thus, even at high density, a condition known to block autophagy in a YAP-dependent manner in static conditions (Fig. 2a–d and[29,30]), stimulation of autophagy was observed in KECs submitted to fluid flow and was confirmed using a CRISPR/Cas9 inactivation of *Yap* (Fig. 2k, l and Supplementary Fig. 2k, l). These data suggest that a specific interplay between YAP/TAZ and autophagy occurs when kidney epithelial cells are under physiological shear stress.

To confirm the role of YAP/TAZ in the regulation of fluid-flow-dependent autophagy flux, we have generated a stable cell line expressing a constitutively active form of YAP (5SA) to counteract YAP inactivation observed in cells submitted to shear stress (see Fig. 1). Of note, this experiment was performed in YAP-knockout cells (YAP$^{KO}$) to avoid effects of endogenous YAP on autophagy. Cells expressing YAP 5SA presented an inhibition of autophagy flux as no accumulation of LC3-II was observed in the presence of CQ compared to cells expressing a constitutively inactive form of YAP (5SA/S94A) (Fig. 3a, b). Accordingly, the levels of LC3 positive structures were not impacted by the presence of CQ in cells expressing the constitutively active form of YAP, confirming an inhibition of autophagy flux under these conditions (Fig. 3c, d). We obtained similar results when overexpressing a constitutively active form of TAZ (Supplementary Fig. 3a), indicating that YAP/TAZ inhibition by fluid flow is necessary to allow autophagy induction in kidney epithelial cells.

We previously reported that fluid flow induces lipophagy[12], a form of selective autophagy that degrades lipid droplets (LDs)[37]. Therefore, we investigated the role of YAP/TAZ activity on the degradation of LDs in KECs submitted to physiological fluid flow. The expression of YAP 5SA led to the accumulation of LDs upon shear (Fig. 3e, f), supporting the fact that YAP inhibition is required to induce the degradation of LDs in kidney epithelial cells under shear stress.

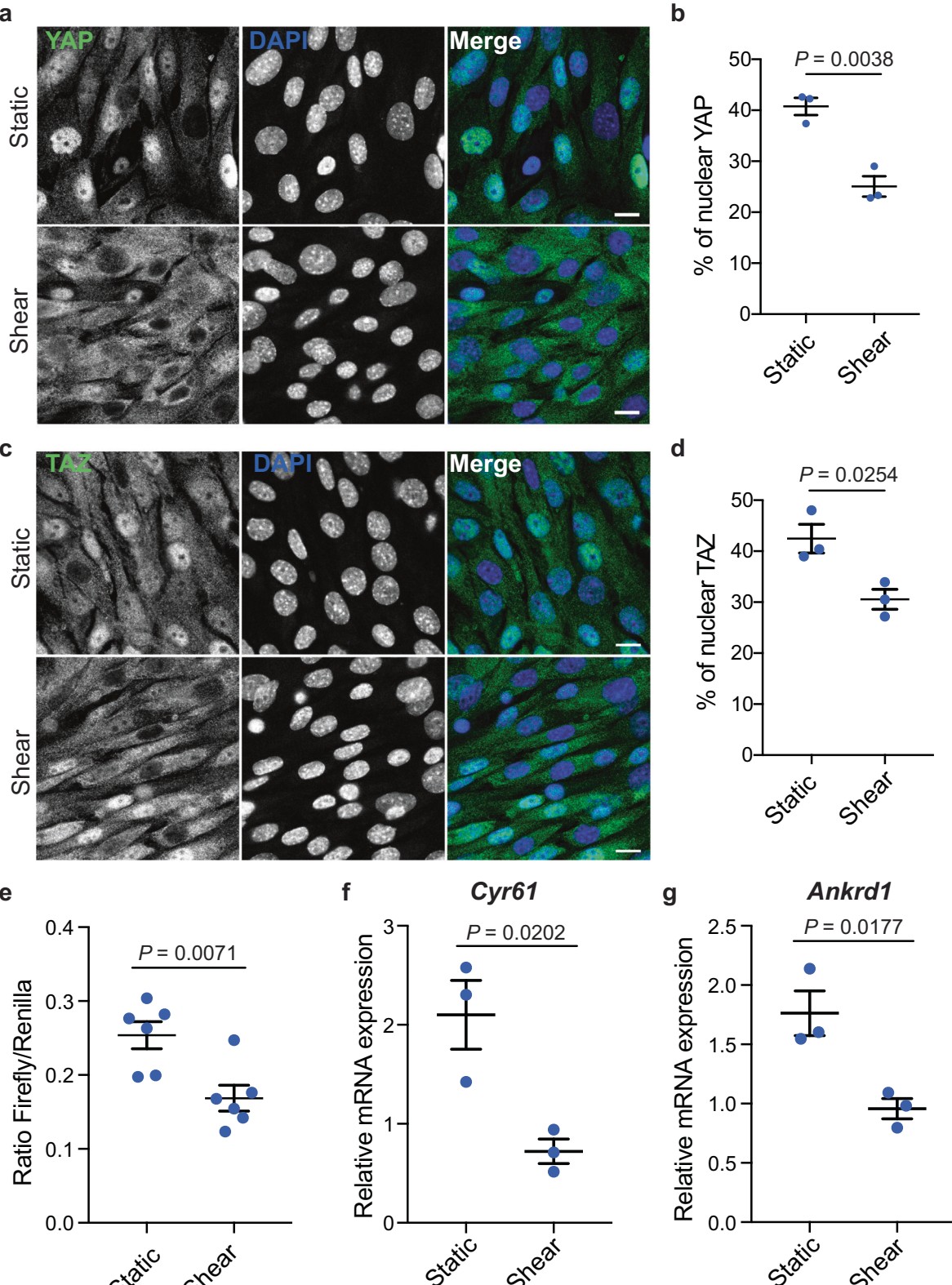

**Fig. 1 | Physiological shear stress inhibits YAP/TAZ. a, b** Representative images (**a**) and quantification (**b**) of YAP localization in the nucleus, labeled with DAPI, in KECs subjected to flow (shear) or not (static) during 24 h. Data show the mean ± s.e.m.; *n* = 3 independent experiments, two-sided *t*-test. Scale bars, 10 μm. **c, d** Representative images (**c**) and quantification (**d**) of TAZ nuclear localization in KECs subjected to flow (shear) or not (static) during 24 h. Data show the mean ± s.e.m.; *n* = 3 independent experiments, two-sided *t*-test. Scale bars, 10 μm.

**e** Luciferase assay for YAP/TAZ activity in KECs subjected to shear stress or not (static) during 24 h. Data show the mean ± s.e.m.; *n* = 6 from 3 independent experiments, two-sided *t*-test. **f, g** Expression of *Cyr61* (**f**) and *Ankrd1* (**g**) in KECs subjected to flow (shear) or not (static) during 24 h. mRNA levels were quantified by real-time RT-qPCR, normalized to β-actin and are presented as fold increases. Data show the mean ± s.e.m.; *n* = 3 independent experiments, two-sided *t*-test. Source data are provided as a Source Data file.

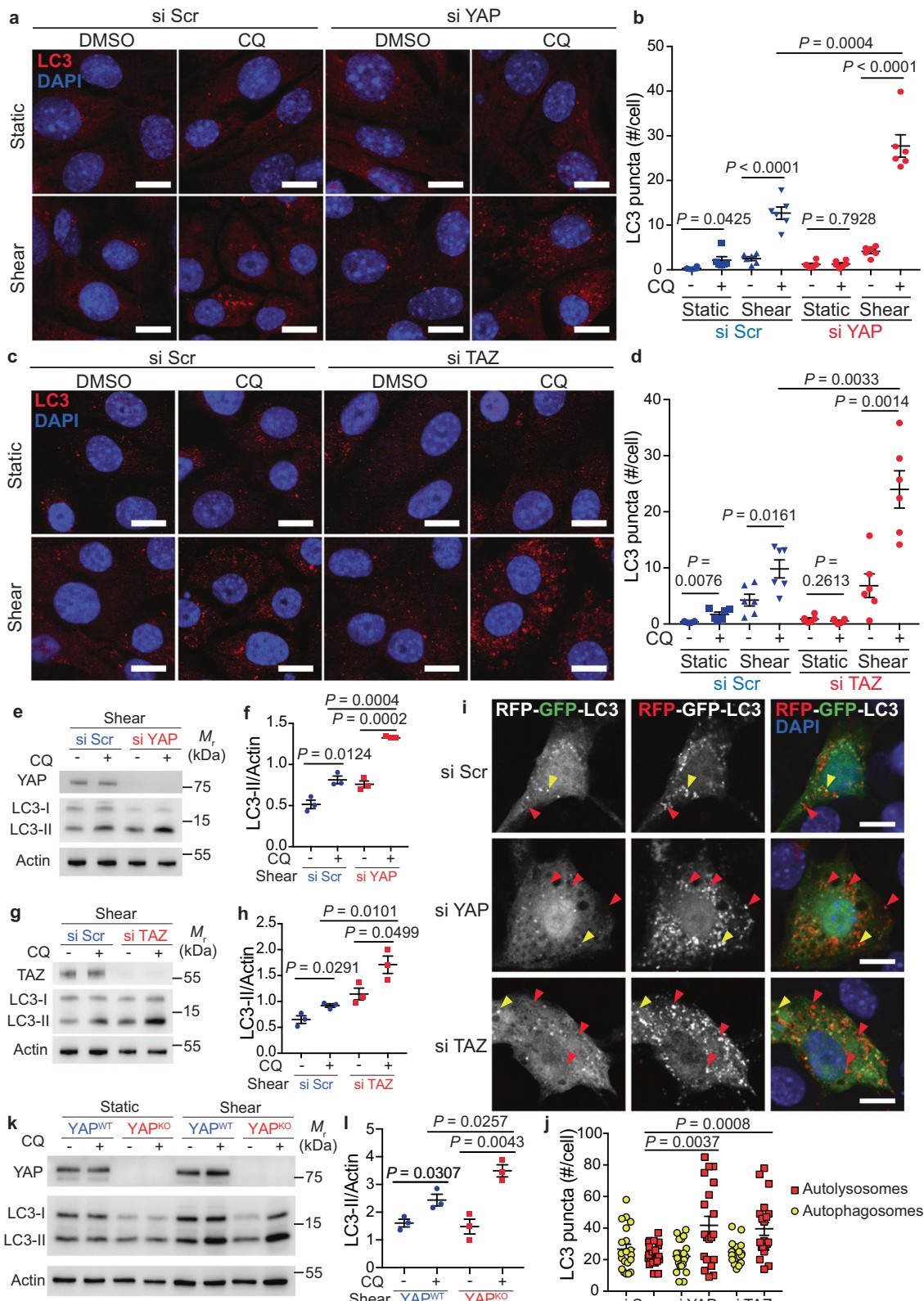

To test whether YAP/TAZ regulate autophagy through TEAD-binding, we overexpressed a genetically encoded fluorescently tagged inhibitor of the interaction between YAP and TAZ with TEAD transcription factors[38]. GFP-TEADi-transfected cells exhibited a high autophagic flux (Supplementary Fig. 3c, d), suggesting that YAP/TAZ regulate autophagy in a manner dependent on their co-transcriptional activity. We thus studied the expression of *Bcl2* and *Rubicon*[33,34,39], as

the genes of these autophagy inhibitors contain TEAD-binding domains. We found that the expression of *Rubicon*, but not *Bcl2*, was significantly reduced in cells submitted to fluid flow (Fig. 3g and Supplementary Fig. 3e). In addition, the expression of *Cptp*, another autophagy inhibitor[40] but also a YAP/TAZ target gene inhibited by fluid flow[21], was reduced in renal epithelial cells during shear stress (Supplementary Fig. 1m). The effect of YAP on the expression of *Rubicon*

**Fig. 2 | The loss of YAP or TAZ stimulates the autophagy flux during shear stress. a, b** Representative images (**a**) and quantification (**b**) of LC3 puncta in KECs after transfection with a control siRNA (si Scr) or siRNA against *YAP*, then subjected to flow (shear) or not (static) during 24 h in the presence or absence of chloroquine (CQ). #: number. **c, d** Representative images (**c**) and quantification (**d**) of LC3 puncta in KECs after transfection with a control siRNA (si Scr) or siRNA against *TAZ*, then subjected to flow (shear) or not (static) during 24 h in the presence or absence of chloroquine (CQ). **e** Representative images of YAP, LC3-I, LC3-II and Actin proteins levels in KECs after transfection with a control siRNA (si Scr) or siRNA against *YAP*, subjected to flow (shear, 24 h) or not (static), in the presence or absence of chloroquine (CQ), by western blot analysis. **f** The ratio of LC3-II to Actin was determined by densitometry, relative to panel **e. g** Representative images of TAZ, LC3-I, LC3-II and Actin proteins levels in KECs after transfection with a control siRNA (si Scr) or siRNA against *TAZ*, subjected to flow (shear, 24 h) or not (static), in the presence or absence of chloroquine (CQ), by western blot analysis. **h** The ratio of LC3-II to Actin was determined by densitometry, relative to panel **g. i, j** Representative images (**i**) and quantification (**j**) of mRFP-GFP-LC3 puncta in KECs transfected with the mRFP–GFP–LC3 tandem probe, subjected to shear stress during 24 h. **k** Representative images of YAP, LC3-I, LC3-II and Actin proteins levels in WT (YAP^WT) and (YAP^KO) KECs (CRISPR Cas9) subjected to flow (shear, 24 h) or not (static), in the presence or absence of chloroquine (CQ), by western blot analysis. **l** The ratio of LC3-II to Actin was determined by densitometry, relative to panel **k. b, d, j** Data show the mean ± s.e.m.; *n* = 6 (**b, d**) or *n* = 20 (**j**) from 3 independent experiments, two-sided *t*-test. Scale bars, 10 μm. **f, h, l** Data show the mean ± s.e.m.; *n* = 3 independent experiments, two-sided *t*-test. Source data are provided as a Source Data file.

was further confirmed in cells expressing the constitutively inactive form of YAP. Indeed, we observed a decrease in the expression of *Rubicon* in cells expressing 5SA/S94A compared to cells expressing a constitutively active (CA) form of YAP (5SA) (Fig. 3h). Accordingly, by chromatin immunoprecipitation experiments, we demonstrated that only the CA form of YAP, and not the constitutively inactive one, was able to bind to the *Rubicon* gene (Fig. 3i, j). Altogether, these findings strongly suggest that Rubicon, an autophagy inhibitor, is transcriptionally controlled by YAP/TAZ. Hence shear stress, by inhibiting YAP/TAZ, downregulates the expression of *Rubicon* and allows the activation of autophagy flux.

### The primary cilium is involved in the control of YAP/TAZ activity upon shear stress

We previously reported that kidney epithelial cells need a functional primary cilium (PC) to sense fluid flow and efficiently activate autophagy[11,12]. To determine the role of PC in regulating the activity of YAP/TAZ upon shear stress, we downregulated the expression of KIF3A, a kinesin family member involved in PC assembly/disassembly and regulation of cilia length[41,42] (Supplementary Fig. 4a–c). Downregulation of KIF3A significantly increased YAP/TAZ nuclear levels upon shear stress, as observed by immunofluorescence experiments (Fig. 4a–d). Accordingly, we observed an increased transactivation activity of YAP/TAZ in cells depleted for KIF3A, compared to control during shear stress (Fig. 4e). The shear stress-dependent nuclear exit of YAP was also inhibited in deciliated cells obtained by knocking down CEP164, a centriole appendage protein required for PC formation (Fig. 4f, g and Supplementary Fig. 4d–f). This correlated with increased transactivation activity of YAP/TAZ (Fig. 4h). Thus, a functional PC is necessary to efficiently inactivate and retain YAP/TAZ in the cytoplasm upon shear stress.

### AMPK-dependent regulation of YAP/TAZ by shear stress

The Hippo/LATS kinases can control YAP/TAZ activity by phosphorylation to induce their cytosolic sequestration or degradation[43]. As we observed cytosolic sequestration of YAP/TAZ upon shear stress, we first analyzed Hippo-dependent phosphorylation of YAP (S127 and S397) and TAZ (S89)[44] when KECs were subjected to fluid flow (Supplementary Fig. 1e and Supplementary 5a–e). Even though we observed a significant increase of TAZ levels upon shear stress, the ratios of TAZ S89/total TAZ, YAP S127/total YAP and YAP S397/total YAP remained unchanged. Accordingly, the levels of active LATS1 (P-LATS1 (T1089)/total LATS1) did not significantly increase during shear stress (Supplementary Fig. 5f, g). These findings were confirmed in *Lats1* and *Lats2* siRNA transfected cells (Supplementary Fig. 5h, i), in which YAP and TAZ nuclear exit was not affected by shear stress (Supplementary Fig. 5j–m). Thus, YAP/TAZ inhibition by fluid flow appears to be Hippo-independent.

Upon shear stress, the PC activates autophagy through an LKB1-AMPK signaling pathway[11,12]. As YAP S61 can be a substrate of AMPK upon glucose starvation[45,46], we hypothesized that AMPK could

phosphorylate YAP upon shear stress. Interestingly, we observed a significant increase in YAP phosphorylation at S61 in KECs subjected to shear stress (Fig. 5a, b). Moreover, YAP phosphorylation at S61 is sufficient to control autophagy induction upon shear stress, as the expression of YAP S61A in YAP^KO-KECs inhibited the accumulation of LC3 puncta and LC3 -II in the presence of CQ, compared to the expression of YAP WT (Fig. 5c, d and Supplementary Fig. 6a, b). These findings raise the possibility that AMPK-dependent phosphorylation of YAP at S61 could regulate YAP cytoplasmic retention and control autophagy when KECs are cultivated under fluid flow. To confirm this hypothesis, we first inhibited the activity of AMPK using dorsomorphin[47], which reduced YAP S61 levels (Supplementary Fig. 6c, d). Under these conditions, we observed a significant increase of nuclear YAP upon shear stress (Fig. 5e, f) alongside the inhibition of lipophagy, as no additive effect was detected in cells treated with dorsomorphin upon Atg5 silencing (Fig. 5g, h). The effect of AMPK on the phosphorylation of YAP was further confirmed in *LKB1* siRNA transfected cells (Fig. 5i, j) and in kidneys from AMPK α1 mutant mice (Fig. 5k–m). In fact, YAP S61 phosphorylation levels were decreased in both in vitro and in vivo conditions. Moreover, YAP nuclear levels were increased in *LKB1* siRNA transfected cells (Supplementary Fig. 6e, f). Altogether, these data show that the AMPK-dependent phosphorylation of YAP controls its cytoplasmic retention as well as autophagic processes during shear stress.

### SIRT1 controls YAP exit from the nucleus in response to fluid flow

To better understand the mechanisms responsible for YAP exit from the nucleus upon fluid flow, we studied the potential role of SIRT1, as it has been previously reported as a negative regulator of the co-transcriptional activity of YAP in endothelial cells subjected to laminar flow[48]. In addition, the deacetylase activity of SIRT1 is dependent on its phosphorylation by AMPK[49]. Moreover, SIRT1 participates in autophagy activation by controlling the nucleocytoplasmic transport of LC3 upon starvation[50]. We first analyzed the subcellular localization of SIRT1 and observed an increase of SIRT1 nuclear levels upon shear stress (Fig. 6a, b), suggesting that SIRT1 is activated in these conditions. Indeed, shear stress decreased the levels of the SIRT1 substrate H3K9ac[51], while the use of EX527, a SIRT1 inhibitor, preserved H3K9ac levels upon shear (Fig. 6c, d). This suggests that SIRT1 is activated in kidney epithelial cells following shear stress. To investigate its role on YAP translocation to the cytosol upon shear, we inhibited SIRT1 using EX527 in mouse KEC and human HK-2 cells. SIRT1 inhibition led to the accumulation of nuclear YAP upon shear, as observed by immunofluorescence experiments (Fig. 6e, f and Supplementary Fig. 7a, b). Accordingly, EX527 treatment during shear stress increased the transactivation activity of YAP/TAZ (Supplementary Fig. 7c) and inhibited both autophagy flux (Fig. 6g, h) and LD degradation (Fig. 6i, j). The effect of SIRT1 on YAP/TAZ activity was further confirmed in cells expressing a mutant form of SIRT1 unable to enter the nucleus (SIRT1-NLS^mut, as previously

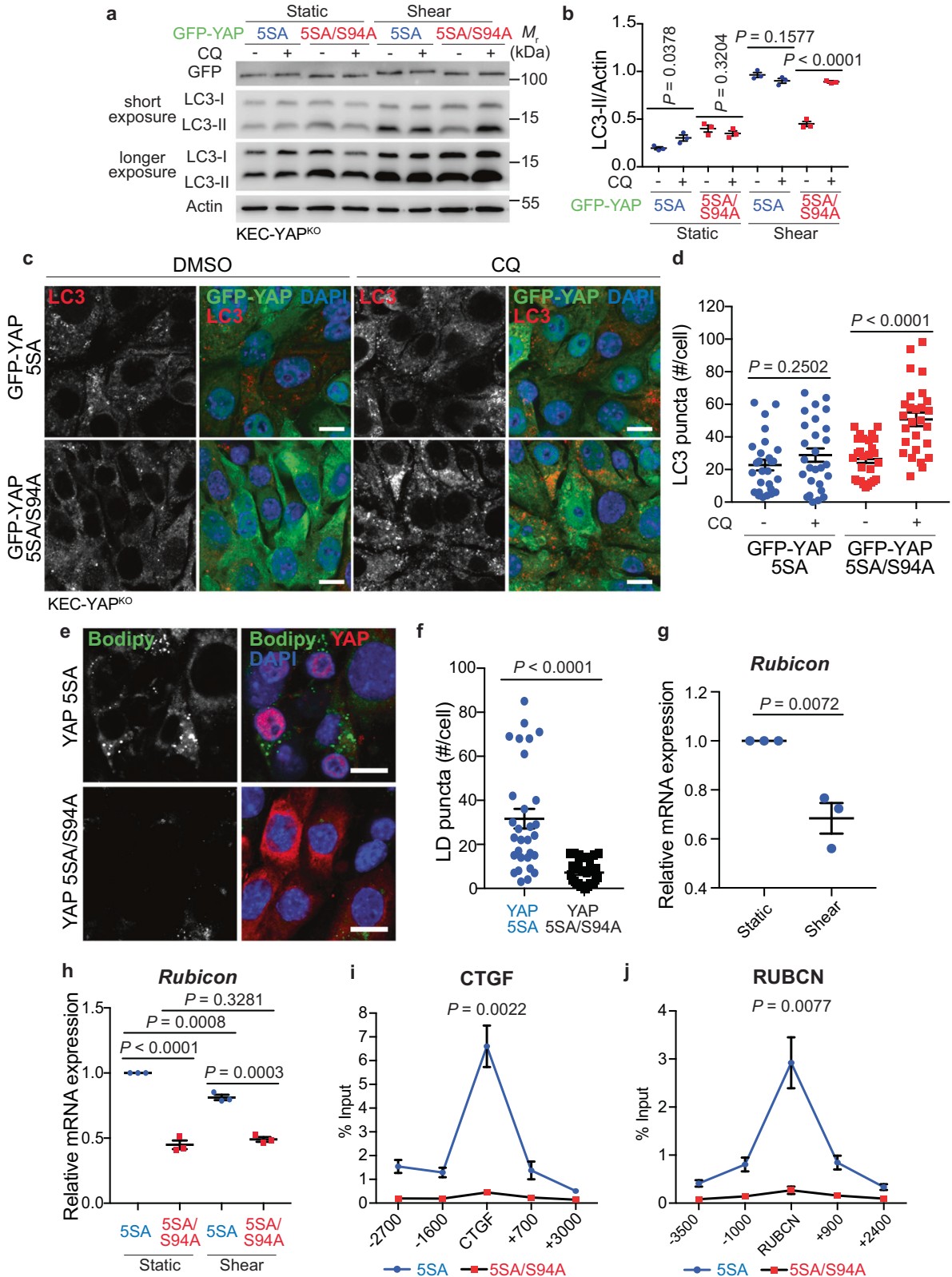

described[52]). Indeed, we observed more YAP/TAZ activity in cells expressing SIRT1-NLS[mut], which was retained in the cytosol (Fig. 6k, Supplementary Fig. 7d), compared to cells expressing SIRT1 WT or a mutant form unable to be transported back to the cytoplasm (SIRT1-NES[mut], Fig. 6k, Supplementary Fig. 7d). Of note, the expression of SIRT1 or SIRT1-NES[mut] negatively affected TEAD promoter activity in static conditions (Fig. 6k, Supplementary Fig. 7d), thus confirming a

role for nuclear SIRT1 in controlling YAP/TAZ activity. As we observed a decrease of acetylated YAP proteins levels upon shear compared to the static condition (Fig. 6l), our data highly suggest a role for the deacetylase SIRT1 in regulating YAP/TAZ activity during shear stress. Previously, we showed that shear stress induced an AMPK-dependent mitochondria biogenesis[12]. EX527-dependent YAP/TAZ transactivation did not alter the shear stress-dependent expression of *Pgc1a* and

**Fig. 3 | The expression of a constitutively active form of YAP inhibits autophagy flux during shear stress. a** Representative images of GFP, LC3-I, LC3-II and Actin proteins levels in KECs expressing a constitutively active (5SA) or inactive (5SA/S94A) form of GFP-YAP, then subjected to flow (shear, 24 h) or not (static), in the presence or absence of chloroquine (CQ), by western blot analysis. **b** The ratio of LC3-II to Actin was determined by densitometry, relative to panel a. Data show the mean ± s.e.m.; *n* = 3 independent experiments, two-sided *t*-test. **c, d** Representative images (**c**) and quantification (**d**) of LC3 puncta in KECs expressing a constitutively active (5SA) or inactive (5SA/S94A) form of GFP-YAP, then subjected to shear stress during 24 h in the presence or absence of chloroquine (CQ). Data show the mean ± s.e.m.; *n* = 27 individual data points from 3 independent experiments, two-sided *t*-test. Scale bars, 10 μm. **e, f** Representative images (**e**) and quantification (**f**) of LDs (Bodipy) in KECs expressing a constitutively active (5SA) or inactive (5SA/S94A) form of YAP, then subjected to shear stress during 24 h. Data show the

mean ± s.e.m.; *n* = 30 individual data points from 3 independent experiments, two-sided *t*-test. Scale bars, 10 μm. **g** Expression of *Rubicon* in KECs subjected to flow (shear) or not (static) during 24 h. mRNA levels were quantified by real-time RT-qPCR, normalized to β-actin and are presented as fold increases. **h** Expression of *Rubicon* in KECs expressing a constitutively active (5SA) or inactive (5SA/S94A) form of YAP, then subjected to flow (shear) or not (static) during 24 h. mRNA levels were quantified by real-time RT-qPCR, normalized to β-actin and are presented as fold increases. **i, j** Relative enrichment of YAP (ChIP GFP-YAP, % input) on the *CTGF* (**i**) and *RUBICON* (**j**) genes in KECs expressing a constitutively active (5SA) or inactive (5SA/S94A) form of GFP-YAP. The *P*-value represent the statistical difference between the 5SA and 5SA/S94A condition at the binding sequence. **g–j** Data show the mean ± s.e.m.; *n* = 3 independent experiments, two-sided *t*-test. Source data are provided as a Source Data file.

*Tfam*, two master genes involved in mitochondria biogenesis (Supplementary Fig. 7e, f). These data suggest that upon shear stress, SIRT1-dependent YAP nuclear exclusion is not involved in the regulation of mitochondria biogenesis but is required for lipo/autophagy induction.

### Effect of the modulation of fluid flow in vivo on YAP localization and autophagy

We next aimed to study the relevance of our in vitro data using a zebrafish model of kidney development. Indeed, many features of the human nephron are shared with zebrafish pronephros, which begin to filter blood at ~48 h post-fertilization (hpf)[53]. We analyzed autophagic readouts in the proximal tubule (PT) using wt1b:GFP;LC3:RFP embryos, at 24 hpf when the zebrafish pronephros possess a lumen but no functionality, and at 48 hpf, when the pronephros begins to operate[53]. We observed that the number of LC3 puncta increased in the PT at 48 hpf compared to 24 hpf (Fig. 7a, b and Supplementary Fig. 8), suggesting that autophagy is induced concomitantly with blood filtration in the kidney pronephros.

In our in vitro model, autophagy induction is dependent on YAP exit from the nucleus. Therefore, we performed immunofluorescence experiments to analyze the subcellular localization of YAP[54] at 24- and 48-hpf in Wt1b⁺ proximal tubules. Using machine learning (see Methods), we delimited the nuclei inside the PT and quantified the percentage of YAP fluorescence in the nuclei and cytoplasm and observed a decrease in the quantity of nuclear YAP at 48-hpf (Fig. 7c, d). Thus, when zebrafish kidney pronephros begin to be functional, YAP translocates to the cytosol and autophagy is induced.

We previously reported that interruption of urinary flow in the kidney of mice by unilateral ureteral obstruction (UUO) resulted in a downregulation of autophagy and accumulation of LDs, at 24 h after UUO[11,12]. Therefore, we investigated whether YAP subcellular localization in the renal tubule (WGA⁺) could be modified during UUO. At this time point, we observed a significant increase of nuclear YAP compared to sham-operated controls (Fig. 7e, f). This result further confirms an important role of fluid flow in controlling YAP subcellular localization.

Altogether, using two independent in vivo models, we confirmed the relevance of the interplay between YAP and autophagy in the functional kidney proximal tubules.

### Pathological fluid flow induces YAP nuclear accumulation and leads to the inhibition of the autophagy flux

Upon kidney injury, the loss of functional nephrons triggers compensatory events to maintain renal function[55,56]. Among those events, residual nephrons adapt by increasing their individual nephron filtration and urinary fluid flow rates. However, over time, those compensatory events lead to a vicious circle in which the loss of nephrons results in the damage of healthy remaining nephrons. To mimic these pathological conditions, we increased the fluid flow rate from 1 to

4 dyn.cm² (as previously detailed[57]), using our microfluidic system. In these conditions, a decrease of SIRT1 protein levels (Supplementary Fig. 9a, b) and a significant increase of nuclear YAP was detected, compared to KECs subjected to physiological flow conditions (1 dyn.cm⁻²) (Fig. 8a, b). Accordingly, the transactivation activity of TEAD was increased in cells subjected to 4 dyn.cm⁻² flow (Fig. 8c). Of note, at the same time point, TAZ subcellular localization was unchanged (Supplementary Fig. 9c, d), suggesting that an early response of YAP, but not TAZ, is induced by pathological flow. The effect of pathological shear on YAP activity was also confirmed in human HK-2 cell line (Supplementary Fig. 9e, f). As several studies have demonstrated a protective role of autophagy in renal proximal tubule cells during diabetic nephropathy[58,59], we hypothesized that YAP activation by pathological conditions could be associated with autophagy inhibition. Indeed, autophagy flux was inhibited in cells submitted to high shear stress (Fig. 8d–g). Moreover, it has been shown that under fluid flow, a pool of activated AMPK is present at the primary cilium in kidney epithelial cells[36]. We observed a decrease of activated AMPK species at the base of primary cilia under high shear stress (4 dyn.cm⁻²) (Fig. 8h, i). However, TEAD activity was reduced in cilia-deficient cells subjected to pathological flow, suggesting that a functional PC is necessary to allow efficient TEAD activation in these conditions (Supplementary Fig. 9g). Thus, YAP activity is regulated by fluid flow intensity and its reactivation by pathological fluid flow leads to autophagy inhibition in renal epithelial cells.

YAP dysregulation has also recently been linked to CKD in humans, including diabetic kidney diseases (DKD), a leading cause of CKD[32]. In fact, previous studies showed the nuclear localization of YAP[32] and the blockade of the autophagy flux[60] in kidney biopsies from DKD patients. To confirm the relevance of the YAP/autophagy interplay in the context of CKD, we investigated YAP subcellular localization and autophagy on biopsies from patients suffering from DKD. Results confirmed that the increased nuclear localization of YAP correlated with decreased fusion of autophagosomes (LC3-positive structures) with acidic compartments (LAMP1-positive structures), suggesting a blockade of the autophagy flux in DKD (Fig. 9a–d). Based on these results and on the in vitro results reported above, these data strongly suggest a causal link between the rate of fluid flow, YAP activity and blockade of the autophagy flux.

### The YAP/autophagy crosstalk promotes renal interstitial fibrosis during unilateral ureteral obstruction

To investigate the role of the YAP/autophagy crosstalk in renal deterioration process during CKD, we took advantage of the well-described unilateral ureteral obstruction (UUO) model of renal fibrosis[61,62]. In fact, 14 days after obstruction, we confirmed the presence of severe tubular lesions and interstitial fibrosis (Fig. 10a–d). Interestingly, at this experimental time point, we observed increased levels of nuclear YAP in tubular epithelial cells (Fig. 10e, f), which correlated with the accumulation of large LC3-positive structures (Fig. 10g–i). However, we also

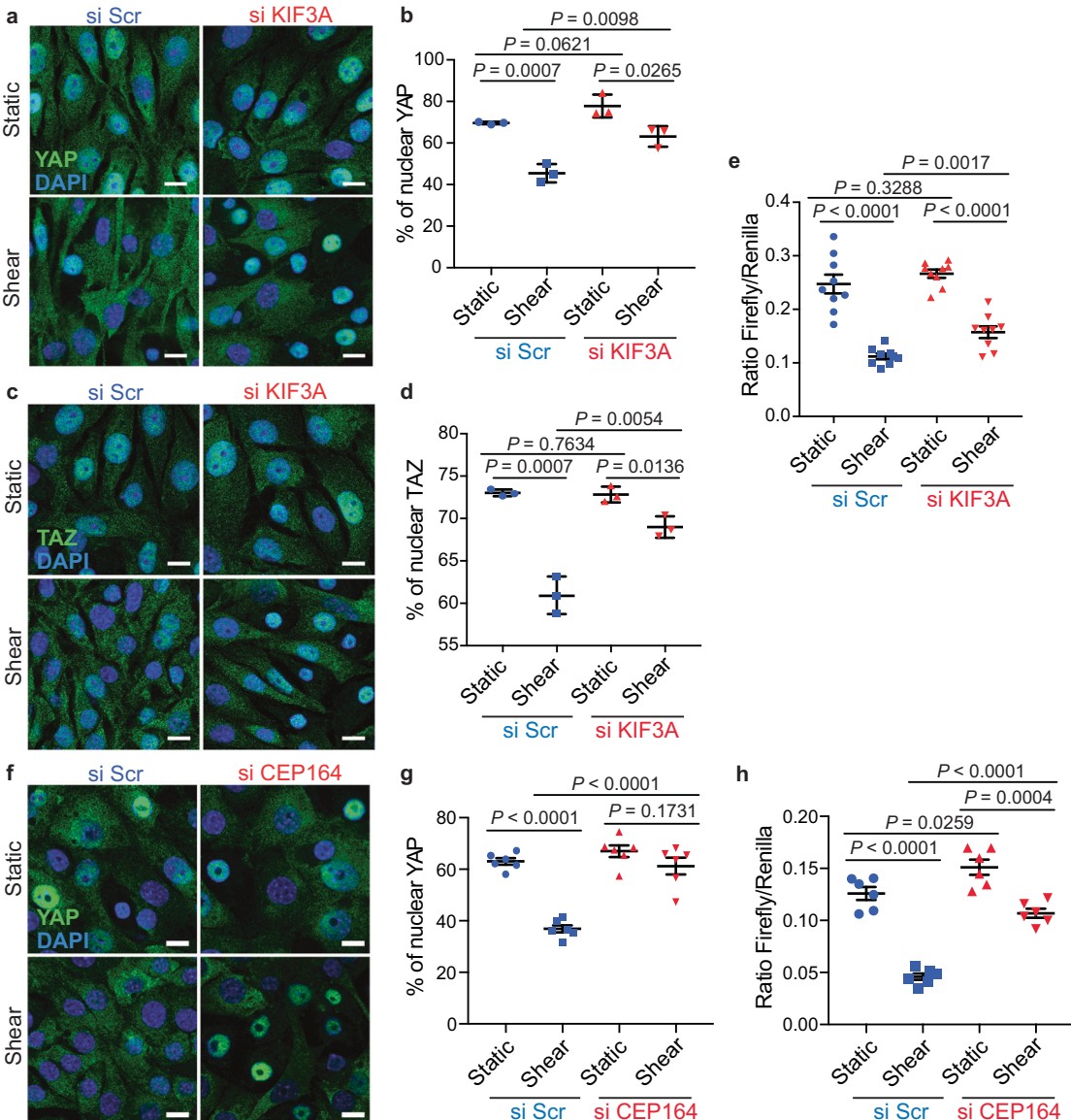

**Fig. 4 | YAP/TAZ inactivation by shear stress requires a functional primary cilium. a, b** Representative images (**a**) and quantification (**b**) of YAP nuclear localization in KECs after transfection with a control siRNA (si Scr) or siRNA against *KIF3A*, subjected to flow (shear) or not (static) during 24 h. Data show the mean ± s.e.m.; *n* = 3 independent experiments, two-sided *t*-test. Scale bars, 10 μm. **c, d** Representative images (**c**) and quantification (**d**) of TAZ nuclear localization in KECs after transfection with a control siRNA (si Scr) or siRNA against *KIF3A*, subjected to flow (shear) or not (static) during 24 h. Data show the mean ± s.e.m.; *n* = 3 independent experiments, two-sided *t*-test. Scale bars, 10 μm. **e** Luciferase assay for YAP/TAZ activity in KECs subjected to shear stress or not (static) during 24 h, after a

knockdown of KIF3A. Data show the mean ± s.e.m.; *n* = 9 from 3 independent experiments, two-sided *t*-test. **f, g** Representative images (**f**) and quantification (**g**) of YAP nuclear localization in KECs after transfection with a control siRNA (si Scr) or siRNA against *CEP164*, subjected to flow (shear) or not (static) during 24 h. Data show the mean ± s.e.m.; *n* = 6 from 3 independent experiments, two-sided *t*-test. Scale bars, 10 μm. **h** Luciferase assay for YAP/TAZ activity in KECs subjected to shear stress or not (static) during 24 h, after a knockdown of *CEP164*. Data show the mean ± s.e.m.; *n* = 6 from 3 independent experiments, two-sided *t*-test. Source data are provided as a Source Data file.

observed a significant accumulation of the autophagy substrate p62/SQSTM1 in obstructed kidneys (Fig. 10j, k), suggesting that the autophagosomes accumulating during UUO could not undergo lysosomal degradation.

To determine whether YAP activation upon kidney damage induced by UUO could be responsible for autophagy inhibition, we selectively inactivated *Yap* in tubular cells by crossing *Yap*^fl/fl^ mice with KSP-CreER mice (Supplementary Fig. 10a, b). Both the score of tubular lesions and the severity of interstitial fibrosis were significantly reduced in kidneys from *Yap*^ΔTub^ mice as compared to *Yap*^fl/fl^ littermates (Fig. 10a–d and Supplementary Fig. 10c, d). Even though we did not observe any significant difference in the number of LC3-positive structures accumulating in *Yap*^ΔTub^ obstructed kidneys (Fig. 10g, h), we

found that they were smaller in *Yap*^ΔTub^ mice compared to *Yap*^fl/fl^ littermates (Fig. 10g, i), suggesting that the autophagy blockade in *Yap*^ΔTub^ is less pronounced than in control littermates. Accordingly, the accumulation of the autophagy substrate p62/SQSTM1 was not significantly increased in *Yap*^ΔTub^ obstructed kidneys (Fig. 10j, k), suggesting that the accumulating autophagosomes observed during UUO could still undergo some lysosomal degradation. Altogether, these data strongly suggest that the YAP/autophagy interplay is instrumental in the renal deterioration process during CKD.

## Discussion

In this study, we uncovered molecular mechano-transduction mechanisms connecting YAP/TAZ and autophagy in mouse and

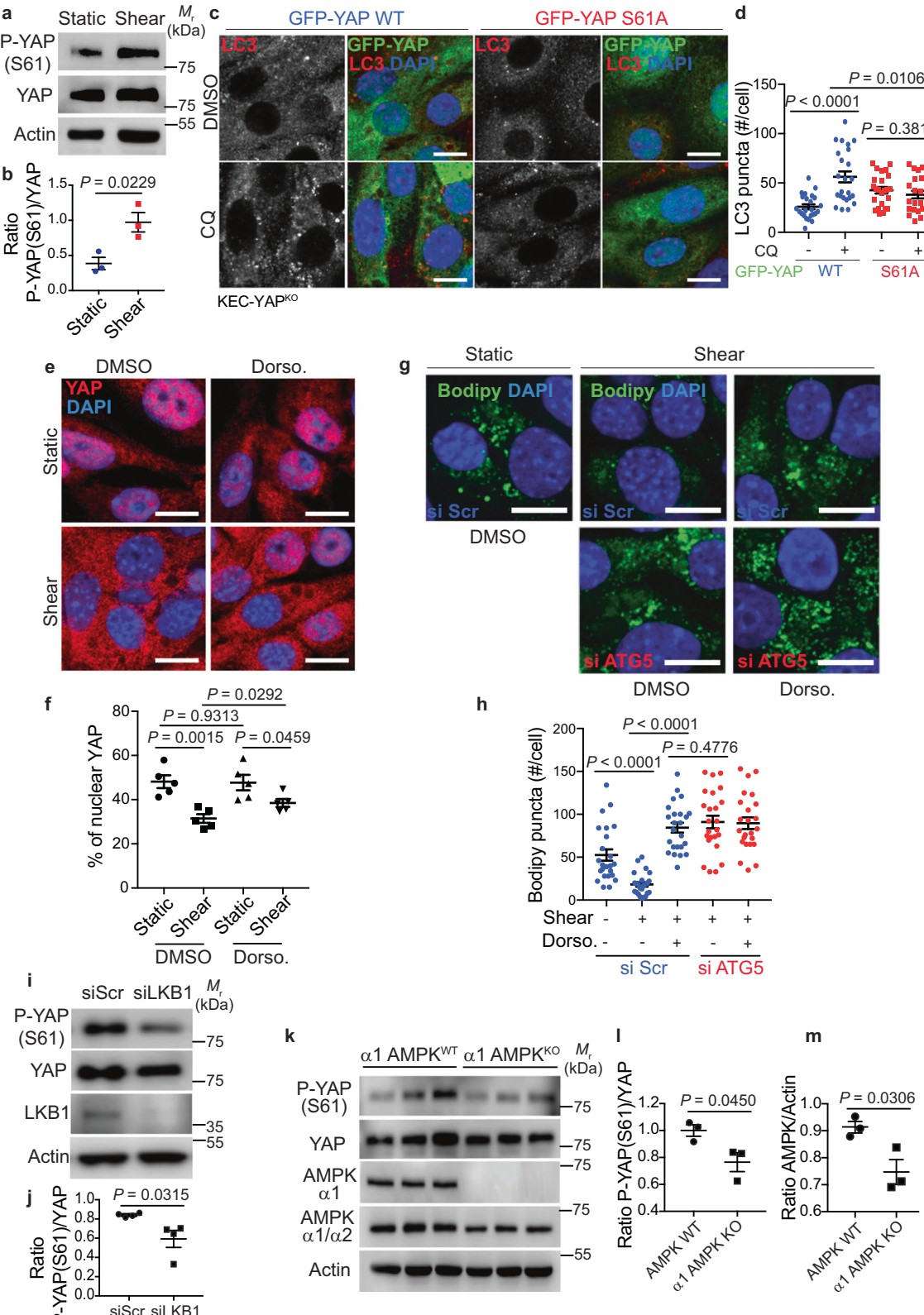

human kidney epithelial cells under shear stress. This process requires the activation of AMPK, which allows efficient inhibition of YAP/TAZ transcriptional activity, upstream of the expression of autophagy inhibitors CPTP and Rubicon. This transcriptional regulation is achieved by a direct phosphorylation of YAP on S61 by AMPK and by AMPK- and SIRT1-dependent YAP nuclear exclusion (Supplementary Fig. 11). While AMPK directly triggers mitochondrial biogenesis[12], the

metabolic adaptation of proximal tubule cells to shear stress relies on YAP to promote autophagy, thus supporting energy-consuming cellular processes such as glucose reabsorption and gluconeogenesis[12]. Altogether, our data and previous results from our laboratory[11,12] and others[36,63] demonstrate that the activation of AMPK plays a central role in maintaining the homeostasis of renal tubular cells, which constantly face physiological fluid flow.

**Fig. 5 | AMPK-dependent phosphorylation of YAP at S61 regulates autophagy upon fluid flow. a** Representative images of P-YAP (S61), YAP and Actin proteins levels in KECs subjected to flow (shear) or not (static) during 24 h, by western blot analysis. **b** The ratio of P-YAP (S61) to YAP was determined by densitometry, relative to panel **a**. Data show the mean ± s.e.m.; $n = 3$ independent experiments, two-sided $t$-test. **c, d** Representative images (**c**) and quantification (**d**) of LC3 puncta in KECs expressing a wild-type (WT) or a mutant (S61A) form of GFP-YAP, subjected to shear stress during 24 h in the presence or absence of chloroquine (CQ). **e, f** Representative images (**e**) and quantification (**f**) of YAP nuclear localization in KECs treated or not with dorsomorphin (Dorso.), subjected to flow (shear) or not (static) during 24 h. Scale bars, 10 μm. **g, h** Representative images (**g**) and quantification (**h**) of lipid droplets (Bodipy) in KECs, after transfection with a control siRNA (si Scr) or siRNA against *ATG5*, treated or not with dorsomorphin (Dorso.) and subjected to flow (shear) or not (static) during 24 h. **i** Representative images of P-YAP (S61), YAP, LKB1 and Actin proteins levels in KECs after transfection with a control siRNA (si Scr) or siRNA against *LKB1* subjected to flow (shear) during 24 h, by western blot analysis. **j** The ratio of P-YAP (S61) to YAP was determined by densitometry, relative to panel **i**. **k** Representative images of P-YAP (S61), YAP, α1 AMPK, total AMPK (α1/α2) and Actin proteins levels in kidney samples from wild-type (WT) or AMPK α1 KO mice, by western blot analysis. **l** The ratio of P-YAP (S61) to YAP was determined by densitometry, relative to panel **k**. **m** The ratio of AMPK to Actin was determined by densitometry, relative to panel **k**. **d, h** Data show the mean ± s.e.m.; $n = 24$ individual data points from 3 independent experiments, two-sided $t$-test. Scale bars, 10 μm. **f, j** Data show the mean ± s.e.m.; $n = 5$ (**f**) or $n = 4$ (**j**) independent experiments, two-sided $t$-test. **l, m** Data show the mean ± s.e.m.; $n = 3$ kidney samples from WT or AMPK α1 KO mice, two-sided $t$-test. Source data are provided as a Source Data file.

In addition, we highlighted here a novel LKB1-AMPK-YAP pathway stimulating autophagic flux in renal cells under specific conditions. Here, we showed that the inhibition of AMPK impairs the phosphorylation of YAP, its nuclear exclusion and the autophagic flux *via* a transcriptional regulation of Rubicon. Taking into account studies from us[11,12] and others[36] reporting the role of an LKB1-AMPK-mTOR axis in regulating autophagy upon shear stress, the present work raises the interesting possibility that AMPK has a dual role on autophagy in KECs subjected to fluid flow: First by controlling autophagic initiation and second by controlling autophagic flux. Whether such a dual AMPK-dependent molecular signalization pathway regulates autophagy in other conditions needs further investigations.

Here, we also reported that the PC is important to drive fluid flow-dependent YAP/TAZ nuclear exclusion. AMPK activity at the PC is dependent on the upstream kinase LKB1[11]. However, how PC control the activation of LKB1-AMPK axis is still unclear. Based on literature showing the interaction between polycystin-1 (PC1) and LKB1[64], we can hypothesize that this molecular pathway could be a good candidate to control AMPK activity in KECs subjected to fluid flow.

YAP/TAZ nuclear exclusion is a crucial step to understand the role of the PC-autophagy axis in the control of cell homeostasis. We reported that YAP/TAZ exit from the nucleus requires SIRT1 activity. However, other proteins, such as XPO1 (known as exportin-1), previously shown to regulate YAP nucleo-cytoplasmic shuttling[65], could also participate in the control of YAP/TAZ localization.

Upon shear stress, the cytoplasmic accumulation of YAP/TAZ decreases the levels of *Rubicon* to control autophagy and metabolism. Indeed, a knockdown of YAP/TAZ led to high autophagic flux levels. These results are in agreement with a recent study, which showed that Rubicon-deficient kidney proximal tubular cells exhibit high autophagy flux and accelerated LDs degradation[66], suggesting the importance of balancing the activity of YAP/TAZ and thus autophagy, to maintain kidney tubular cells homeostasis.

In this study, we revealed the importance of mechanical stimuli to control YAP/TAZ and autophagy. Previous reports have shown that matrix stiffness[30] and contact inhibition[28,29] are important factors regulating the YAP/TAZ-autophagy axis. In these contexts, autophagy is compromised upon YAP/TAZ inhibition (cytoplasmic sequestration) and we confirmed this in renal epithelial cells cultivated under static conditions (See Fig. 2). However, simulating physiological shear stress in vitro stimulated autophagy in a cytoplasmic YAP/TAZ-dependent manner. Interestingly, Hippo-dependent cytoplasmic retention of YAP/TAZ induced by high density or matrix stiffness, inhibits autophagy by repressing the expression of proteins engaged in the formation[28,29] and maturation[30] of autophagosomes. In this work, we show that under shear stress, a Hippo-independent cytoplasmic retention of YAP/TAZ stimulates autophagy by repressing the expression of autophagy inhibitors (*Rubicon* and *Cptp*). It should be kept in mind that the different cytoplasmic forms of phospho-YAP could recruit different partners[67,68], to control autophagy in a posttranscriptional manner. Thus, according to the mechanical stress sensed by the cell, a specific response will occur to translate these forces into the appropriate metabolic adaptation.

AMPK and autophagy are well known to be dysregulated in renal diseases, including in CKD[69]. In addition, several GWAS studies of CKD have identified several loci, including a locus containing *PRKAG2*, the gene encoding the AMPK γ2 subunit. Of note, this gene has been shown to be associated with rapid kidney function decline[70,71]. Based on this literature, different molecules that directly or indirectly activate AMPK, such as metformin (already widely used to treat type 2 diabetes mellitus), have been identified as efficient inhibitors of the progression of renal diseases, including polycystic kidney disease, renal cancer or acute kidney injury[69,72,73]. Despite promising results, it is necessary to be careful with the therapeutic use of metformin in humans, because of side effects such as lactic acidosis observed in some CKD patients. Here, we reported that pathological fluid flow, mimicking the shear stress observed in CKD, is sufficient to re-localize YAP to the nucleus and block autophagy flux. We also uncovered that the loss of YAP in renal tubules is able to protect against UUO-induced tubular injury and fibrosis. These data are in line with previous studies showing that tubular YAP is hyperactivated and deleterious in CKD mouse models[32,61,74,75] but also with other studies reporting the protective role of autophagy in UUO mouse models. It is worth noting, however, that the role of autophagy in UUO-induced fibrosis seems to be more complex, since the loss of tubular *Atg7* has been shown to reduce renal fibrosis[76,77]. Similarly, the role of YAP seems to be beneficial after an acute injury such as ischemia-reperfusion[78]. Whether the type of lesions or the difference in flow intensity can explain this discrepancy deserves further investigation. Lastly, we have also shown that the effect of pathological fluid flow on YAP and autophagy are correlated with a defect in AMPK activation at the base of the primary cilia. Thus, our study has shed light on the role of a specific ciliary pool of AMPK in maintaining the homeostasis of renal tubular cells and opens new avenues to develop therapeutical strategies for CKD patients.

## Methods

### Patients
Renal biopsies were routinely performed in patients with diabetic nephropathy and analyzed and conserved in the pathology Department of Hôpital Necker-Enfants Malades (Paris, France). Renal biopsies from patients with normal kidney histology (minimal change disease) were used as controls. In total, we studied as controls 6 patients with normal kidney histology and 6 patients with moderate diabetic kidney disease. Patient characteristics are depicted in Supplementary Table 1. Kidney biopsies were fixed in formalin, alcohol, and acetic acid and paraffin-embedded. 4-μm sections of paraffin-embedded kidneys were

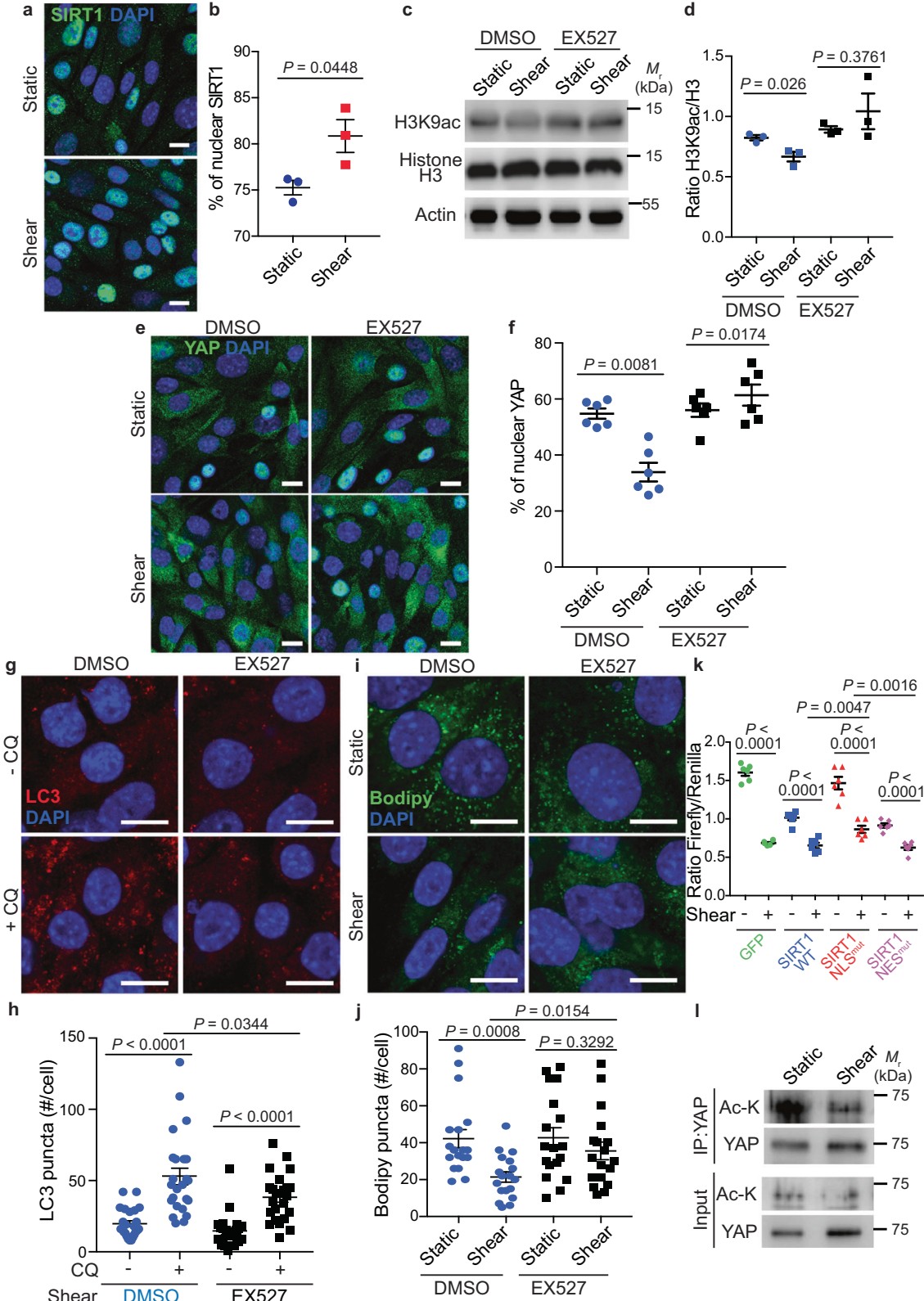

submitted to the appropriate antigen retrieval. Then, sections were incubated with either YAP (Cell Signaling Technology, #14074, 1:200) or LAMP1 (Abcam, ab24170, 1:100) and LC3B (MBL, #M152.3, 1:200 dilution), antibody overnight at 4 °C, followed with the appropriate secondary antibody. DAPI was used to stain nuclei. Whole kidney sections were then scanned using a nanozoomer 2.0 HT (Hamamatsu) with a X40 objective. Our human kidney biopsy samples were taken in

the context of patient care and then repositioned for biomedical research purposes. Written consent has been obtained from each patient, or their parents in the case of minors. No financial compensation is foreseen as these samples are taken in the context of care and not initially taken for research purposes. The protocol was approved by the Institutional Review Board of Hôpital Necker-Enfants Malades, APHP, Paris, France.

**Fig. 6 | SIRT1 induces YAP nuclear exclusion upon fluid flow. a, b** Representative images (**a**) and quantification (**b**) of SIRT1 nuclear localization in KECs subjected to shear stress during 24 h. Data show the mean ± s.e.m.; $n = 3$ independent experiments, two-sided $t$-test. Scale bars, 10 μm. **c** Representative images of H3K9ac, Histone H3 and Actin proteins levels in KECs subjected to flow (shear) or not (static) during 24 h, treated or not with EX527, by western blot analysis. **d** The ratio of H3K9ac to Histone H3 was determined by densitometry. Data show the mean ± s.e.m.; $n = 3$ independent experiments, two-sided $t$-test. **e, f** Representative images (**e**) and quantification (**f**) of YAP nuclear localization in KECs treated or not with EX527, subjected to flow (shear) or not (static) during 24 h. Data show the mean ± s.e.m.; $n = 6$ individual data points from 3 independent experiments, two-sided $t$-test. Scale bars, 10 μm. **g, h** Representative images (**g**) and quantification (**h**) of LC3 puncta in KECs subjected to flow (shear) or not (static) during 24 h, in the

presence or absence of EX527 and chloroquine (CQ). Data show the mean ± s.e.m.; $n = 24$ individual data points from 3 independent experiments, two-sided $t$-test. Scale bars, 10 μm. **i, j** Representative images (**i**) and quantification (**j**) of lipid droplets (Bodipy) in KECs treated or not with EX527, subjected to flow (shear) or not (static) during 24 h. Data show the mean ± s.e.m.; $n = 18$ individual data points from 3 independent experiments, two-sided $t$-test. Scale bars, 10 μm. **k** Luciferase assay for YAP/TAZ activity in KECs transfected with GFP or GFP-SIRT1 WT or its mutated forms (SIRT1 NLS$^{mut}$ or SIRT1 NES$^{mut}$), subjected to shear stress or not (static) during 24 h. Data show the mean ± s.e.m.; $n = 6$ from 3 independent experiments, two-sided $t$-test. **l** Immunoprecipitation analysis of YAP in KECs subjected to flow (shear) or not (static) during 24 h. Representative images of YAP and acetylated proteins are shown from $n = 3$ independent experiments. Source data are provided as a Source Data file.

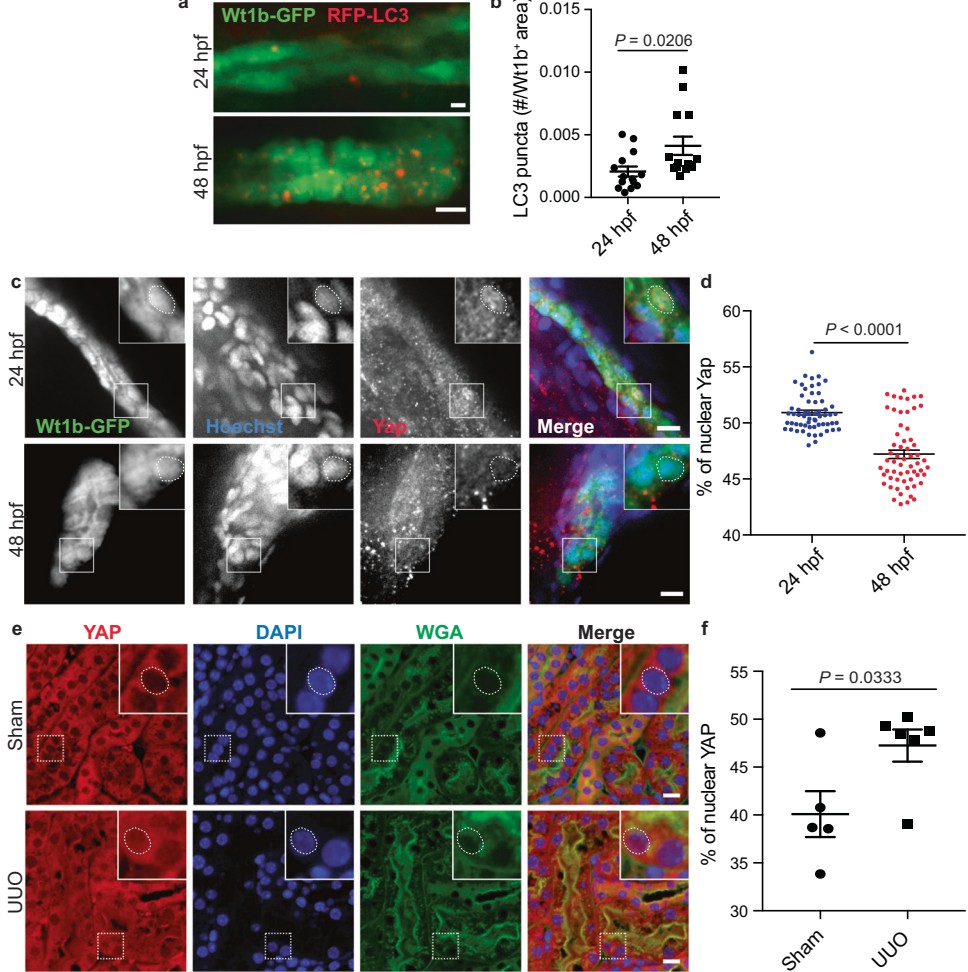

**Fig. 7 | YAP subcellular localization is associated to autophagy activity in vivo. a, b** Representative images (**a**) and quantification (**b**) of RFP-LC3 puncta in Wt1b-GFP$^+$ pronephros, at 24 h post fertilization (hpf) and 48 hpf. Data show the mean ± s.e.m.; $n = 18$ individual data points from 3 independent experiments, two-sided $t$-test. Scale bars, 10 μm. **c, d** Representative images (**c**) and quantification (**d**) of YAP nuclear localization in Wt1b-GFP$^+$ pronephros at 24 and 48 hpf. Nucleus were labeled with Hoechst. Data show the mean ± s.e.m.; $n = 60$ individual data points

from 3 independent experiments, two-sided $t$-test. Scale bars, 10 μm. **e, f** Mice subjected to unilateral ureteral obstruction (UUO) or sham operation were euthanized 24 h after surgery and kidney sections subjected to immunohistochemistry for YAP and renal tubule marker Wheat Germ Agglutinin (WGA). **e** Representative images. Scale bars, 10 μm. **f** Quantification of YAP nuclear levels in the renal tubules (WGA$^+$). Data show the mean ± s.e.m.; $n = 5$ (sham) and $n = 6$ (UUO) different mice. two-sided $t$-test. Source data are provided as a Source Data file.

## Animals and protocols

Animal procedures were approved by the departmental director of the Services Vétérinaires de la Préfecture de Police de Paris and by the ethical committee of the Paris Descartes University. Animals were housed in a specific pathogen-free facility, fed *ad libitum* (standard diet; 18% proteins, 6% fat, #2018, Inotiv produced from Mucedola,

Milan, Italy), and housed at constant ambient temperature (between 20 and 21 °C) and humidity ranges (50–60%) in a 12 h light cycle.

Nine-week-old female mice (C57BL/6 strain, Janvier Laboratories) were subjected to unilateral ureteral obstruction (UUO; $n = 6$) or sham operation (controls; $n = 5$) as previously described[79] and euthanized 24 h after surgery. During the same period (24 h), mice were deprived

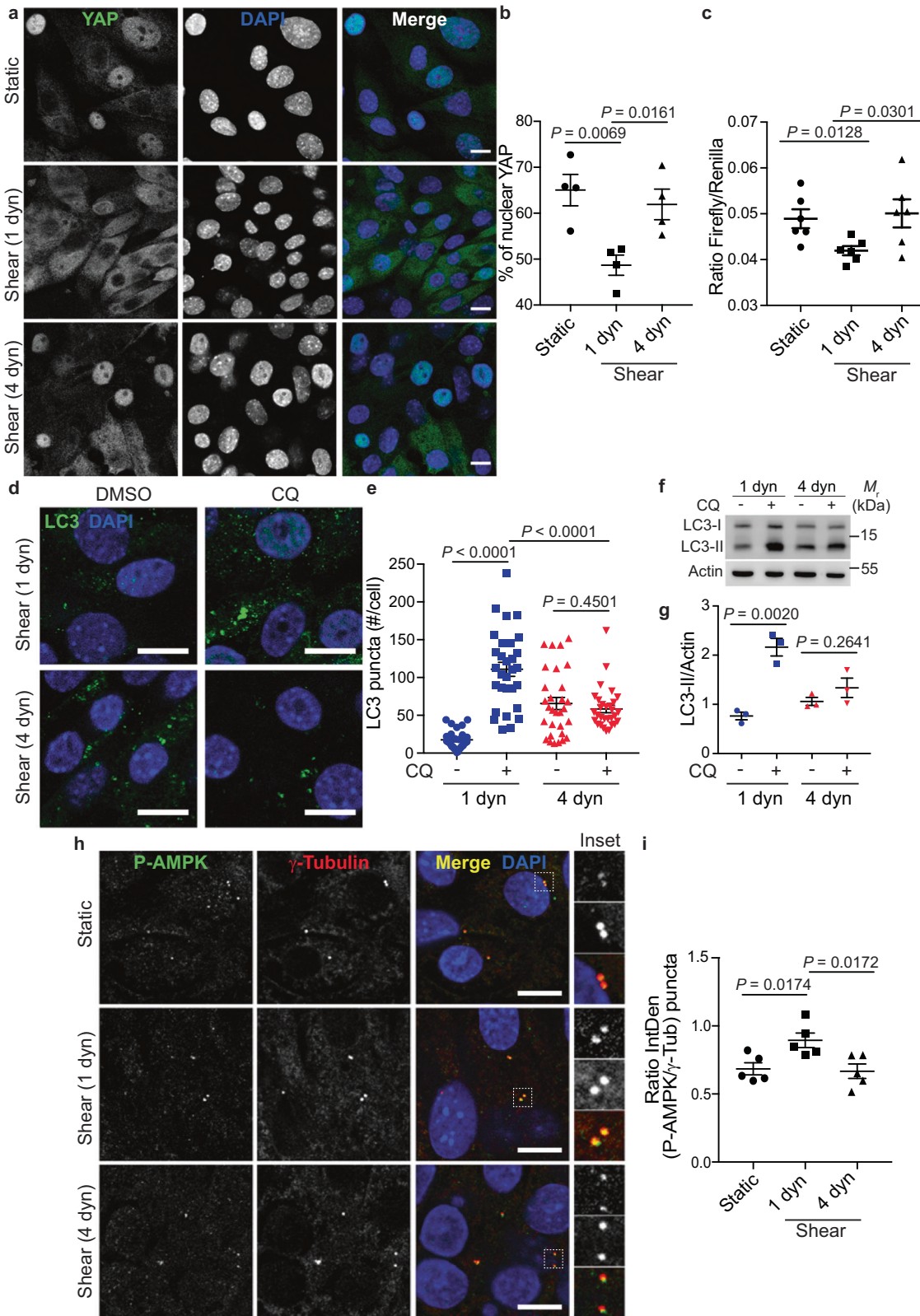

of food. At euthanasia, kidneys were removed for immuno-fluorescence analysis. Briefly, 4-µm sections of paraffin-embedded kidneys were submitted to the appropriate antigen retrieval. Then, sections were incubated with YAP (Cell Signaling Technology, #14074, 1:200) antibody overnight at 4 °C, followed with the appropriate secondary antibody and WGA-Rhodamine (Vector Laboratories, RL-1022, 1:200). DAPI was used to stain nuclei. Whole kidney sections were then

scanned using a nanozoomer 2.0 HT (Hamamatsu) with a X40 immersion objective. ImageJ software was used to quantify the percentage of nuclear YAP (Intensity Ratio Nuclei Cytoplasm Tool, RRID:SCR_018573) in the proximal kidney tubules.

To generate AMPKα1 KO mice, AMPKα1 [f/f] mice[80] (C57BL/6 strain) were crossed to EIIa-Cre mice (C57BL/6 strain, Jackson laboratory)[81]. Kidneys from 6-month-old female mice were harvested

**Fig. 8 | Pathological flow induces YAP nuclear translocation and inhibits autophagy. a**, **b** Representative images (**a**) and quantification (**b**) of YAP nuclear localization in KECs subjected to physiological flow (shear 1 dyn) or not (static) during 48 h. For pathological flow (4 dyn), KECs were subjected to physiological shear stress (1 dyn) during 24 h before increasing the flow rate to 4 dyn.cm$^{-2}$ during one more day. Data show the mean ± s.e.m.; $n = 4$ independent experiments, two-sided $t$-test. Scale bars, 10 μm. **c** Luciferase assay for YAP/TAZ activity in KECs subjected to shear stress or not (static) during 48 h, as described for **a**. 1 dyn = Physiological flow. 4 dyn = pathological flow. Data show the mean ± s.e.m.; $n = 6$ from 3 independent experiments, two-sided $t$-test. **d**, **e** Representative images (**d**) and quantification (**e**) of LC3 puncta in KECs subjected to physiological flow (shear 1 dyn) or pathological flow (shear 4 dyn, as described in panel **a**), in the presence or absence chloroquine (CQ). Data show the mean ± s.e.m.; $n = 30$ individual data

points from 3 independent experiments, two-sided $t$-test. Scale bars, 10 μm. **f**, **g** Representative images of LC3-I, LC3-II and Actin proteins levels in KECs subjected physiological flow (shear 1 dyn) or pathological flow (shear 4 dyn, as described in panel **a**), in the presence or absence chloroquine (CQ) by western blot analysis. **g** The ratio of LC3-II to Actin was determined by densitometry, relative to panel **f**. Data show the mean ± s.e.m.; $n = 3$ independent experiments, two-sided $t$-test. **h**, **i** Representative images (**h**) and quantification (**i**) of colocalization between phosphorylated AMPK (P-AMPK) and centrioles (γ-Tubulin) in KECs subjected to physiological flow (shear 1 dyn), pathological flow (shear 4 dyn, as described in panel **a**), or not (Static). Data show the mean ± s.e.m. of P-AMPK intensity, compared with γ-tubulin intensity; $n = 5$ independent experiments, two-sided $t$-test. Scale bars, 10 μm. Source data are provided as a Source Data file.

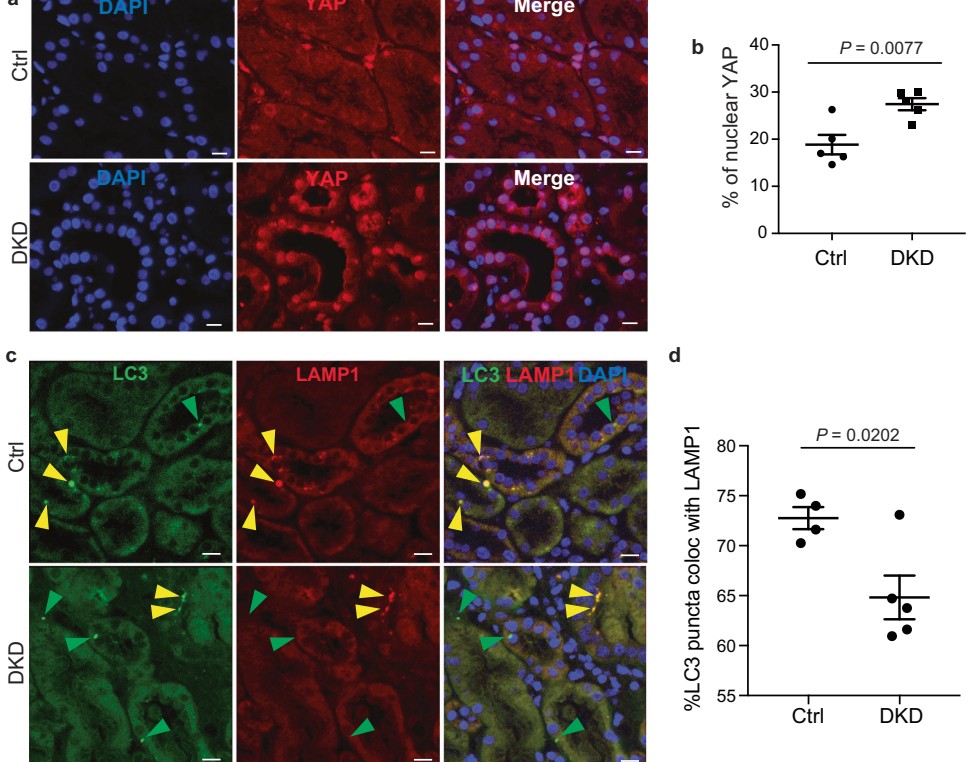

**Fig. 9 | YAP reactivation upon chronic kidney disease is correlated with autophagy inhibition. a** Representative images of kidney sections subjected to immunohistochemistry for YAP from controls ($n = 5$) and patients with diabetic nephropathy (DKD, $n = 5$) Scale bars, 30 μm. **b** Quantification of YAP nuclear levels in the renal tubules. Data show the mean ± s.e.m.; $n = 5$. two-sided $t$-test. **c** Representative images of kidney sections subjected to immunohistochemistry for LC3 and LAMP1 from controls and patients with diabetic nephropathy ($n = 5$) Scale bars, 30 μm. **d** Quantification of colocalization between LC3 and LAMP1 in the renal tubules. Data show the mean ± s.e.m.; $n = 4$ (controls) and $n = 5$ (diabetic nephropathy). two-sided $t$-test. Source data are provided as a Source Data file.

and frozen at −80 °C until further processing for protein expression analysis.

For the *Yap* tubular deletion model, all experiments were performed on 9-week-old female mice (FVB/N strain). *Ksp/CreERT2* mice that express an inducible (tamoxifen) Cre recombinase under the control of the *Ksp-cadherin* (Cdh16) promoter and *Yap^{f/f}* mice containing loxP sites have been described previously[82,83]. Cre-negative, *Yap^{f/f}* littermates were used as negative controls. *Ksp/CreERT2XYap^{flox}* mice (*Yap^{Δtub}*, $n = 10$) and *Yap^{f/f}* littermates (*Yap^{flox}*, $n = 10$) were subjected to unilateral ureteral obstruction (UUO) and sacrificed 14 days after surgery. At the time of sacrifice, the right control contralateral kidney and the left obstructed kidney were removed, weighed, and analyzed by morphological studies. Kidneys were fixed in 4% paraformaldehyde and paraffin-embedded. Four-μm sections were stained with periodic acid-Schiff (PAS) and picrosirius red. Images were acquired using a digital slide scanner (Nanozoomer S210 Digital Slide

Scanner, Hamamatsu). Renal lesions were blindly examined by a renal pathologist who was unaware of the group studied. The degree of tubular atrophy and the extent of interstitial fibrosis was evaluated using a semiquantitative injury score on PAS and picrosirius red staining sections, respectively, as previously described[84] in up to 10 random selected fields (original magnification, X100) across the kidney cortex. For immunofluorescence analysis, 4-μm sections of paraffin-embedded kidneys were submitted to the appropriate antigen retrieval. Then, sections were incubated with either LC3B (MBL, #M152.3, 1:200 dilution) or SQSTM1 (PROGEN, GP62-C, 1:100) antibody overnight at 4 °C, followed with the appropriate secondary antibody. For YAP detection, sections were first treated with avidin/biotin kit (Origene, E08-18) before using YAP primary antibody (Cell Signaling Technology, #14074, 1:200) overnight at 4 °C. Sections were then incubated with biotinylated anti-rabbit antibody (GEHealthcare, RPN1004V) followed by Alexa Fluor™ 488 streptavidin. DAPI was used

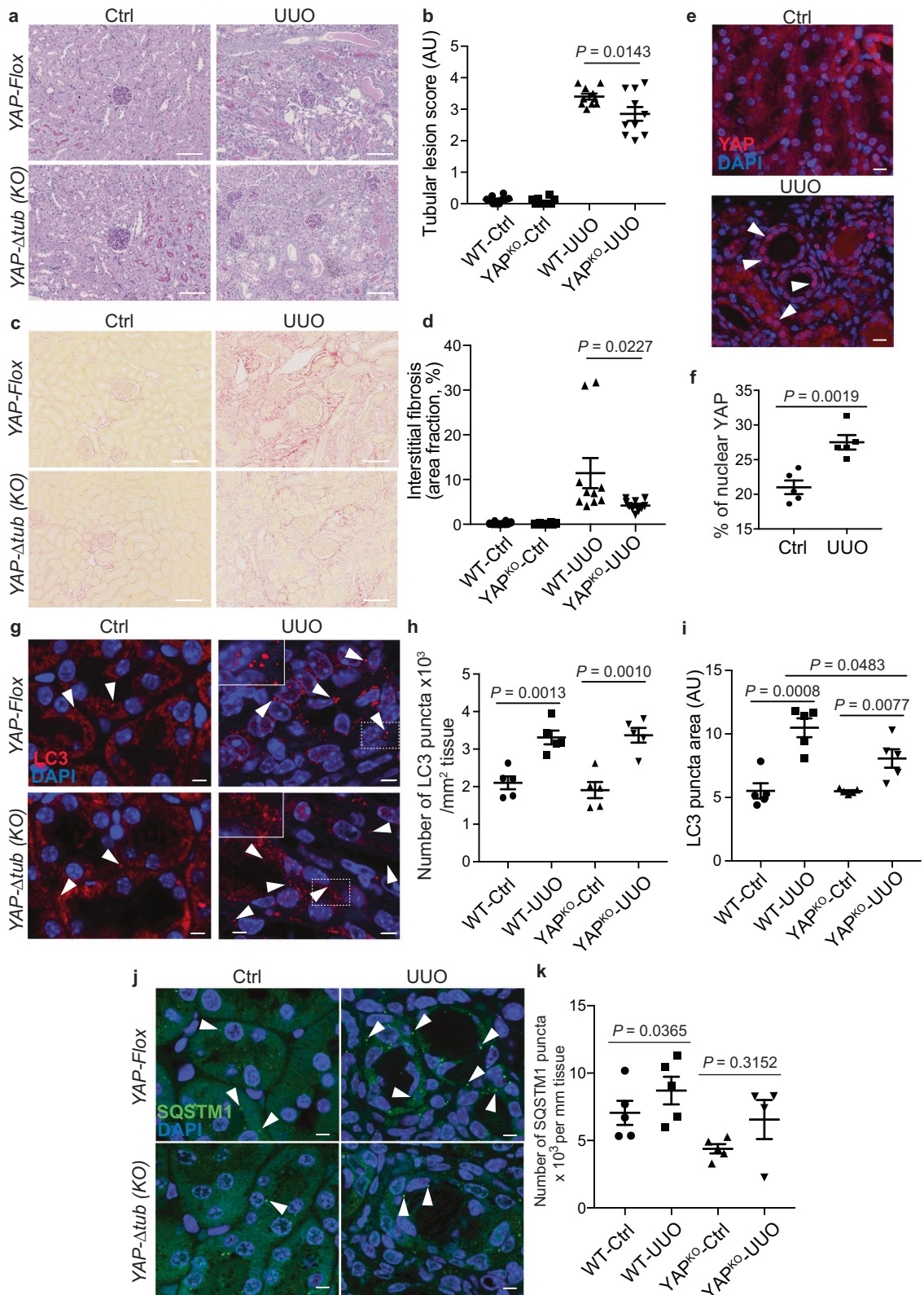

to stain nuclei. Whole kidney sections were then scanned using a nanozoomer 2.0 HT (Hamamatsu) with a X40 immersion objective and 6 random selected fields across the kidney cortex were analyzed with ImageJ software for YAP, Icy software for LC3 and SQSTM1 puncta quantification.

The Zebrafish Tg(hsp70l:RFP-Rno.Map1lc3b) line (RFP-LC3) was obtained from Dr. Enrico Moro (University of Padova, Italy)[85] and the

Tg(wt1b:GFP) line[86] was used to label proximal pronephros. As the experiments performed for this study were using zebrafish embryos <120 hpf, no ethical permission was needed. Embryos were obtained by natural mating and raised at 28.5 °C in petri dishes containing fish water. To induce the expression of RFP-LC3, larvae were collected 6 hpf and heat shocked by placing in 42 °C fish water and then incubating for 60 min at 37 °C. Fish were then placed back at 28.5 °C until

**Fig. 10 | The YAP/autophagy crosstalk promotes renal interstitial fibrosis during unilateral ureteral obstruction. a, b** Representative images (**a**) of the kidney cortex of control contralateral kidneys and obstructed kidneys (UUO kidney) 14 days after surgery in *Yap^Δtub* mice and *Yap^flox* littermates (PAS staining, original magnification X200) and quantification (**b**) of the tubular injury score. Data show the mean ± s.e.m.; *n* = 10 for each experimental group. ANOVA, followed by the two-sided Tukey-Kramer test. **c, d** Representative images (**c**) of the kidney cortex of control contralateral kidneys and obstructed kidneys (UUO kidney) 14 days after surgery in *Yap^Δtub* mice and *Yap^flox* littermates (picrosirius staining, original magnification X200) and quantification (**d**) of the interstitial fibrosis area. Data show the mean ± s.e.m.; *n* = 10 for each experimental group. ANOVA, followed by the two-sided Tukey-Kramer test. **e, f** Representative images (**e**) of kidney

sections subjected to immunohistochemistry for YAP. Scale bars, 30 μm (**f**) Quantification of YAP nuclear levels in the renal tubules. Data show the mean ± s.e.m.; *n* = 5 mice. two-sided *t*-test. **g–i** Representative images of control contralateral kidneys and obstructed kidneys (UUO kidney) in *Yap^Δtub* mice and *Yap^flox* littermates subjected to immunohistochemistry for LC3. Scale bars, 10 μm. **h, i** Quantification of LC3 puncta number (**h**) and size (**i**) in the renal tubules. Data show the mean ± s.e.m.; *n* = 5 for each experimental group. two-sided *t*-test. **j, k** Representative images (**j**) of kidney sections subjected to immunohistochemistry for SQSTM1. Scale bars, 10 μm. **k** Quantification of SQSTM1 puncta in the renal tubules. Data show the mean ± s.e.m.; *n* = 5 for each experimental group. two-sided *t*-test. Source data are provided as a Source Data file.

live imaging at 24 and 48 hpf. To avoid pigmentation of embryos, 0.003% 1-phenyl-2-thiourea (PTU) was added to the embryo medium 24 hpf. Immunofluorescence experiments were performed as described previously[87]. Briefly, embryos were fixed overnight at 4 °C in 4% PFA, then washed 3 times for 20 min in PBS 1% Triton X-100 (PBS-Triton). Embryos were incubated for 1 hr in blocking buffer (PBS-Triton, FBS 10%) then overnight at 4 °C with anti-Yap antibody (Cell Signaling, #4912, 1:200) in blocking buffer. Embryos were then washed 3 times for 20 min in PBS-Triton before incubation with anti-rabbit Alexa Fluor 546 used at 1:500 dilution in blocking buffer containing Hoechst 33342 (Sigma) to label the nuclei. After 3 washes in PBS-Triton, embryos were mounted in 2% low melting agarose. Z-stacks were acquired using a Zeiss Spinning disk piloted by Zen software (20X objective). For image analysis, Ilastik Pixel classification (v1.3.3post3)[88] models were created for each channel (i.e Hoechst (nuclei, Blue) and Wt1b-GFP (tubules, green)) with live prediction using at least 3 representative images of all data analyzed. The model was then validated when all predictions on representative images were visually correct. Using Fiji (v2.3.0/1.53f51)[89] was a macro designed by Nicolas Goudin (available upon request), we then verified each image (one by one) and corrected if necessary the Wt1b-GFP and the Hoechst masks created with Ilastik, in order to finally reach the ground truth. Finally, we measured different local background to estimate the mean background and measured the ratio of the mean YAP intensity with background substraction into corrected 3D ROIs, inside and outside the nucleus.

## Cell culture, transfections and plasmids

Wild-type mouse KECs, generated by G.J. Pazour (University of Massachusetts), were kindly provided by A.M. Cuervo (Albert Einstein College) and HK-2 cells (ATCC) were cultured in Dulbecco's Modified Eagle Medium (DMEM), supplemented with 10% FBS at 37 °C and 5% $CO_2$. The cells used in this study were free of mycoplasma contamination. No cell lines used in this study were found in the database of commonly misidentified cell lines maintained by ICLAC and NCBI Biosample.

To generate YAP^KO in KECs, oligonucleotides targeting Exon 2 were annealed and clones into pSpCas9(BB)−2A-Puro (Addgene # 62988, a kind gift from Feng Zhang) into the BbsI sites by golden gate cloning (pCas9-sgYAP). The guide sequence was GCTGGCAGTGGTA-CATCATC (Exon 2). The cloned guides were sequence verified. KECs were co-transfected with pCas9-sgYAP and pEGFP (Addgene # 31796, a kind gift from Robin Shaw) using Lipofectamine 2000 (Thermo Fisher) and 48 h later clonally sorted by FACS based on GFP expression. Single-cell clones were then expanded and protein expression was assessed by immunoblotting.

To create the lentiviral GFP YAP constructs, the SV40-Puromycin expression cassette was PCR amplified from the pLX303 backbone and ligated into the pBA904 (kind gift from Jonathan Weissman, Addgene #122238) backbone at the XbaI site by Gibson assembly to create pLNK_Puro. The GFP-YAP and GFP-YAP mutant constructs were then assembled by first PCR amplifying

mGreenLantern (a kind gift from Gregory Petsko, Addgene # 161912) and the YAP elements, which were inserted between the NheI and EcoR1 sites of pLNK_Puro by Gibson assembly. Lentivirus was packaged in HEK293T cells by PEI transfection with the transfer vector, psPAX2 and pVSVG.

To generate stable KECs expressing GFP-YAP or a GFP-YAP mutant (constitutively active 5SA and constitutively inactive 5SA/S94A, as previously described[90] or S61A), KECs YAP^KO were infected with the appropriate lentiviral vectors (MOI 2.5), produced in HEK293T cells. The supernatant containing lentiviruses was collected, centrifuged, and filtered through 0.45 μm filter and stored in aliquots at −80 °C. Two days after infection, cells were subjected to puromycin selection for 48 h (1 μg/mL) to enrich for targeted cells. The expression of GFP-YAP (WT or mutants) was assessed by immunofluorescence and immunoblotting. To validate the cell lines, a TEAD luciferase assay was performed, to confirm the increase in TEAD promoter activity in the YAP 5SA and S61A mutants, compared to YAP 5SA/S94A and WT expressing cells, respectively.

To inhibit the expression of YAP, TAZ, KIF3A, CEP164, ATG5, LKB1 and LATS1/2, cells were doubly transfected using RNAiMAX (Invitrogen) according to the manufacturer's instructions with siRNAs targeting mRNAs encoding for YAP, TAZ, KIF3A, CEP164, ATG5, LKB1 and LATS1/2 from Qiagen (listed in Supplementary Table 2). After transfection, cells were seeded onto microslides and subjected to shear stress the following day. Non-targeting siRNA pool (Scramble) was used as a control.

DNA transfections were done with Lipofectamine 2000 (Invitrogen). Plasmids used in this study are listed in Supplementary Table 3. YAP S61A mutant was generated utilizing a QuikChange site-directed mutagenesis kit (Agilent) and confirmed by sequencing (Eurofins Genomics).

## Flow chamber

KECs and HK2 were subjected to shear stress as previously described[11]. Briefly, cells were seeded at high density in closed perfusion chambers (Microslide I^{0.6} Luer; channel dimensions 50 × 5 x 0.4 mm coated with iBiTreat, Ibidi) and cultured for at least 1 day at full confluency to allow polarization and maximum ciliogenesis. Medium was changed twice a day before flow induction. The chamber was then connected to a computer-controlled set-up containing an air-pressure pump and a two-way switching valve (Ibidi pump system #10902), piloted by a Pump control software (Ibidi) using a perfusion set (Ibidi Blue #10961, length 15 cm, ID 0.8 mm). Ten mL of cell culture medium was pumped unidirectionally between two reservoirs through the flow channel at a rate corresponding to a shear stress of 1 dyn.cm^{-2} (physiological fluid flow) for 1 day for KECs and 2 days for HK2. For cells cultivated in static conditions, medium was changed twice a day for 24 h (KECs) or 48 h (HK2). To simulate pathological fluid flow, cells were subjected to 1 dyn.cm^{-2} shear stress for 1 day to allow cells' differentiation. Then, the fluid flow was increased to 4 dyn.cm^{-2} for 1 day before analysis, to simulate the flow rate observed soon after subtotal nephrectomy by micropuncture studies[55,91].

## Pharmacological inhibitors

All autophagic measurements and treatments were performed as described previously[92]. Briefly, where indicated, cells were treated with chloroquine (CQ, 10 μM, Sigma-Aldrich) to inhibit autophagic maturation. To inhibit AMPK and SIRT1, cells were respectively treated with dorsomorphin (Dorso., 10 μM, Sigma-Aldrich) or EX527 (10 μM, Sigma-Aldrich).

## Immunoblotting, antibodies and detection assays

Blots were labeled with antibodies against LC3B (Sigma, #L7543, 1:5000 dilution), YAP (Cell Signaling Technology, #4912, 1:1000), Phospho-YAP_S127 (Cell Signaling Technology, #13008, 1:1000), Phospho-YAP_S61 (Cell Signaling Technology, #75784, 1:1000), Phospho-YAP_S397 (Cell Signaling Technology, #13619, 1:1000), TAZ (Sigma, #HPA007415, 1:1000), Phospho-TAZ_S89 (Cell Signaling Technology, #59971, 1:1000), LATS1 (Cell Signaling Technology, #9153, 1:1000), Phospho-LATS1_T1079 (Cell Signaling Technology, #8654, 1:1000), H3K9ac (Sigma, #07-352, 1:1000), Histone H3 (Proteintech, #17168-1-AP, 1:1000), LKB1 (Cell Signaling, #3050, 1:1000), AMPK (Cell Signaling, #2532, 1:1000), AMPKα1 (a gift from Grahame Hardie, University of Dundee, UK, 1:10000), acetylated lysine (Millipore, #06-933, 1:1000), SIRT1 (Millipore, #07-131, 1:1000), and Actin (Millipore, Clone C4 MAB1501, 1:20,000 dilution). Appropriate HRP-labeled anti-rabbit (Millipore, #AP307P, 1:10000), anti-mouse (Millipore, #AP308P, 1:10000) and anti-sheep (Calbiochem, #402100, 1:10000) were then used and revealed with Super Signal West Dura chemiluminescent substrate (Pierce). Images were taken with the ChemiDoc-MP Imaging System and quantified using ImageJ software.

## YAP immunoprecipitation

KECs cells submitted to 1 day of shear stress (1 dyn.cm$^{-2}$) or not (static) were trypsinised (pool from 3 microslides) and lysed in RIPA buffer supplemented with protease and phosphatase inhibitor cocktail (Halt™, Thermo Fisher) and 1 mM phenylmethylsulfonyl fluoride (Sigma) for 30 min on ice. After centrifugation 5 min at 15,300 g, supernatants were incubated with 2 μL of anti-YAP antibody (Cell Signaling, #14074) overnight at 4 °C. The immune complexes were captured by adding 50 μL of Dynabeads (Thermo Fisher Scientific) for 2 h at 4 °C. Dynabeads were washed three times with PBS and bound proteins eluted with 2X Laemmli sample buffer (Bio-Rad) before being subjected to immunoblot analysis. Immunoblotting images were visualized and analyzed using ImageLab v.6.0.0.

## qRT–PCR

Total RNA was extracted from cells using the Cells-to-Ct Kit (Applied Biosystems), according to the manufacturers' instructions. Real-time PCR was performed using SYBR Green Master Mix (Applied Biosystems), and products were detected on a qTOWER 3 G Real-Time PCR System (Analytik Jena) piloted by qPCRsoft 4.0. Relative expressions of *Cyr61, Ankrd1, Rubicon, Ptplad2, Cptp, Bcl2* and *β-actin* were calculated using the $2^{\Delta\Delta C(t)}$ method. β-actin was used as normalization control. Conditions for real-time PCR were as follows: initial denaturation for 10 min at 95 °C, followed by amplification cycles of 15 s at 95 °C, and 1 min at 60 °C. The qPCR primers used in this study are listed in Supplementary Table 4.

## Conventional microscopy and antibodies

Immunofluorescence microscopy was carried out using a Zeiss Spinning disk piloted with Zen software with a X63 oil immersion objective. Cells grown on microslides were fixed for 10 min at RT in 4% paraformaldehyde (PFA), permeabilized (5 min, 0.1% Triton X-100), blocked (30 min, 10% FBS) and incubated with the following primary antibodies in blocking buffer for 2 h at RT: YAP (Cell Signaling Technology, #14074, 1:100), TAZ (Sigma, #HPA007415, 1:100), LC3B (MBL, #PM036 or #M152.3, 1:200), FLAG-M2 (Sigma, # F1804, 1:200), SIRT (Millipore,

#07-131, 1:100), WIPI2 (Abcam, # ab105459; 1:150) and ATG16L1 (MBL, #PM040, 1:150) followed by incubation with fluorophore-conjugated secondary antibodies (Molecular Probes) for 1 h. For primary cilia imaging, cells were fixed with cold methanol for 10 min at −20 °C, blocked (30 min, 10% FBS) and incubated with the following primary antibodies in blocking buffer containing 0.05% Saponin for 2 h at RT: ARL13B (Proteintech, # 66739, 1:100), γ-tubulin (Sigma, #T5326, 1:200), Phospho-AMPK_T172 (Cell Signaling Technology, #2535, 1:200), followed by incubation with fluorophore-conjugated secondary antibodies (Molecular Probes) for 1 h.

The cells were also labeled with Bodipy 493/503 (Thermo Fisher Scientific, #D3922, 1:200) to detect the LDs by fluorescence microscopy, as described previously[93]. Mounting medium containing 4′,6-diamidino-2-phenylindole (DAPI) was used when required. ImageJ software v2.1.0/1.53c was used to quantify the percentage of nuclear YAP/TAZ (Intensity Ratio Nuclei Cytoplasm Tool, RRID:SCR_018573) and colocalization between Phospho-AMPK and γ-tubulin (Red and Green Puncta colocalization Macro, D. J. Swiwarski modified by R.K. Dagda). Icy software (licence GPLv3[94]) was used to quantify LC3 and Bodipy puncta (Plugin spot detector, objects measured: scale 2 (spots -3 pixels), https://gitlab.pasteur.fr/bia/spot-detector), and cilia length (Plugin ROI statistics, https://gitlab.pasteur.fr/bia/roi-statistics).

## Luciferase reporter assay

The luciferase reporter construct 8xGTIIC-Luciferase (Firefly) was a gift from Stefano Piccolo (Addgene plasmid #34615)[35]. KECs were transfected with 8xGTIIC plasmid along with Renilla luciferase (Ratio 10:1). The next day, cells were seeded onto microslides and subjected to shear stress the following day. At the end of shear stress, luciferase activity was measured using Pierce Renilla-Firefly Luciferase Dual Assay Kit (Thermo Scientific, #16185), with a ThermoTriStar2s LB492 luminometer (Berthold) piloted by Mikrowin 2010, according to manufacturer's instructions.

## Chromatin Immunoprecipitation (ChIP)

KECs YAP$^{KO}$ stably expressing constitutively active or inactive GFP-YAP mutants (5SA or 5SA/S94A, respectively) were grown in 15 cm dishes (2 dishes/experiment) for 48 h to reach 80% confluency. Approximately 50 million cells were trypsinized, washed in HBSS (Thermo Fisher Scientific) and incubated in freshly prepared 2 mM DSG (Thermo Fisher Scientific) in DPBS (Thermo Fisher Scientific) for 45 min at room temperature on a rotating wheel. After pelleting and washing in DPBS, cells were crosslinked in 1% formaldehyde (Sigma) in DPBS for 10 min at room temperature on a rotating wheel. Crosslinking was stopped by the addition of 125 mM Glycine (Sigma) for 5 min. After quenching, cells were pelleted, washed with ice-cold DBPS and resuspended in ChIP Buffer (25 mM Tris-HCl pH 8.0, 2 mM EDTA, 150 mM NaCl, 1% Triton X100, 0.3% SDS, protease inhibitors (Roche)). After sonication on a Bioruptor Pico (Diagenode) and clearing by centrifugation, 10 to 30 μg of chromatin were used for immunoprecipitation in 0.1% SDS-ChIP buffer overnight at 4 °C with 40 μl of Dynabeads protein A (Thermo Fisher Scientific) pre-coated with 2 μg of GFP antibody (ab290, Abcam). Beads were washed with five different washing buffers as described in[95]. DNA was then eluted, de-crosslinked and extracted with the IPure kit v2 (Diagenode) and analysed by qRT-PCR with PowerTrack™ SYBR Green Master Mix (Thermo Fisher Scientific) on a QuantStudio 5 machine (Thermo Fisher Scientific). Chromosomal regions with and without binding of TEAD transcription factors on the Rubicon and CTGF genes were selected by browsing publicly available ChIP-seq datasets (GSM1342158[96] and GSM1331248[97]). Primer sequences are listed in Supplementary Table 5.

## Statistics and reproducibility

Statistical analyses were carried out using Prism 7.0a (GraphPad) or Microsoft Excel v16.16.23. *P*-values were calculated using the unpaired

*t*-test or analysis of variance. All values are given as mean ± SEM. For animal studies, no statistical method was used to predetermine sample size. The experiments were not randomized.

### Reporting summary

Further information on research design is available in the Nature Portfolio Reporting Summary linked to this article.

## Data availability

Authors confirm that all relevant data are included in the paper and/or its Supplementary Information files. Source Data for Figs. 1–10 and Supplementary Figs. 1–11 are provided with this paper. Resources and materials are available from Aurore Claude-Taupin (aurore.claude-taupin@inserm.fr) or Nicolas Dupont (nicolas.dupont@inserm.fr). Source data are provided with this paper.

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

## Acknowledgements

We are grateful to Ana Maria Cuervo (Albert Einstein College) for sharing KECs and Sirio Dupont (University of Padova) for sharing FLAG-TAZ WT and 4SA plasmids. We thank Enrico Moro (University of Padova, Italy) for providing Tg(hsp70l:RFP-Rno.Map1lc3b) embryos, as well as Marion Delous and Sophie Saunier (Imagine institute) for sharing their Tg(wt1b:GFP) zebrafish line. We thank Grahame Hardie (University of Dundee, Dundee, UK) for providing anti AMPKα1 antibody. We are grateful to Dorien Peters (Leiden University Medical Center) for providing us the *Ksp/CreERT2* mice. Alexander Benmerah is acknowledged for fruitful discussion. We thank David Romeo-Guittart and Franck Oury for sharing tools to study SIRT1. We also thank Carla Alemany and Juliane Da Graça for technical assistance. We thank the LEAT and Histology platforms of the Structure Federative de Recherche Necker for technical assistance. A.C-T is supported by Fondation Tourre and Fondation L'Oréal-UNESCO For Women in Science. This work was funded by Institut National de la Santé et de la Recherche Médicale (INSERM), Centre National de la Recherche Scientifique (CNRS), Université Paris Cité and Agence Nationale de la Recherche (ANR; ANR-17-CE13-0015-03, R16167KK and ANR-18-CE14-0006-02 to P.C.; ANR-18-CE14-0026-01 and ANR-22-CE13-0004 to N.D.).

## Author contributions

Conceptualization: A.C.-T., N.D. and P.C. Formal analysis: A.C.-T., P.I., A.B., B.R., N.G., F.R., M.G.-T. and A.R. Investigation and validation: A.C.-T., P.I., A.B., B.R., L.T., F.R., M.G.-T., A.R. and M.B. Resources: N.D., P.C., E.M., M.F., B.V. and F.T. Data curation: A.C.-T. and N.D. Writing (original draft): A.C.-T., N.D. and P.C.; Writing (comments): E.M., F.T., B.V., M.P., and N.K. Visualization: A.C.-T., P.I., A.B., F.R. and A.R. Supervision: N.D. F.T. and P.C.

## Competing interests

The authors declare no competing interests.
