## [Peer Review File · Nature Communications]

The AMPK-Sirtuin 1-YAP axis is regulated by fluid flow intensity and controls autophagy flux in kidney epithelial cellsREVIEWER COMMENTS

Reviewer #1 (Remarks to the Author):

This is a manuscript by Claude-Taupin et al. investigating a relationship between YAP-dependent autophagy, primary cilium and fluid flow, and its potential role in kidney disease. While the topic is interesting, it is felt that claims are not supported by the experimental data and that the physiological relevance is not sufficiently proven. Also, there is a heavy reliance on immunofluorescence for quantitative analysis. Confirmation of critical data by other more quantitative techniques is felt missing. Specific details are below:

1. All immunofluorescence figures: Lower magnification with larger field of view pictures are needed to assess changes occur in the majority of the cells. This is especially relevant in those experiments in which more quantitative techniques (i.e. cell fractioning) are not possible. Confirmation by WB needs to be included in key data.
2. Figure 1: Data shows YAP-TAZ is constitutively active located in the nucleus in KECs. Any comment on this? Is this seen in other cell types?
3. Extended Figure 1: No increased in LC3-II is observed in D1 cells treated with CQ. Any explanation for this? YAP and LC3 levels in static conditions at D4 should be included.
4. Extended Figure 1: YAP protein levels are affected by CQ treatment (also in observed in extended Fig 2) and shear stress itself. Any comment on that?
5. Cell density was not evaluated; therefore, the claim that YAP is mainly located in the cytosol in cell at high density is not accurate. There is no information in methods whether cells are cultivated and high or low density conditions. The discussion about this is a little distracting and it is not clear how it does support the main topic.
6. Have expression levels of Ptplad2 and Cptp in D0 and D1 been evaluated to confirm that the downregulated expression is specific to fluid flow or in contrast YAP/TAZ regulation based in culture conditions?
7. On Page 6, authors should clarify what exactly “autophagy flux is amplified”. Is autophagy activated? Is autophagy flux increased under same autophagy activation?
8. On Page 7, the claim “even at high density, condition known to block YAP-dependent autophagy in static conditions (Fig 2a-d)” needs clarification. What is YAP-dependent autophagy? Which figures do exactly back up this claim?
9. Figure 2e-f: CA YAP seems to change LC3 cellular localization. How were puncta counted in these figures, they do not seem very defined. LC3-II blots are needed.

10. On Page 7, “indicating that YAP/TAZ inhibition by fluid flow is necessary to allow autophagy induction in KEC” is not supported by accompanying figures; autophagy activation is not seen.
11. Figure 2g- CA YAP is located in the cytoplasm. Is this the expected location?
12. Figure 2i- have expression of Rubicon in cells with CA and inactive YAP been quantified?
13. On Page 8: The claims that “shear stress downregulates the expression of YAP/TAZ target genes, including several autophagy inhibitors and allows for autophagy activation” is not supported by the data.
14. Figure 3a-d: All the data included in these figures need to be confirmed by WB. LC3 puncta does not seem reliable based on the poorly defined and overexposed figures.
15. Figure 3e: This should be confirmed by cell fractioning- if not possible based on cell quantity in fluidic chambers, lower magnification pictures must be shown.
16. Extended Figure 4- any comments on why YAP, TAZ and LATS1 protein levels are elevated with shear stress?
17. Figure 4g,h: Lipophagy was not monitored; therefore, the claims at the end of Page 9 are not accurate.
18. Figure 5: Low magnification pictures are needed. Figure 5e shows more cytosolic YAP in EX527-treated cells, which is not shown in the graph.
19. Extended Figure 5a- The decreased in P-YAP with Dorso is not convincing. Quantification is needed.
20. Extended Figure 5b- Statistics in DMSO and EX527 under shear stress are not convincing.
21. Figure 6 c,d: How machine learning was authenticated?
22. Figure 6e: YAP staining needs to be shown in red instead of b/w. As shown, the increased nuclear translocation is not appreciated.
23. Claim on Page 12: “using two independent in vivo models, we confirmed the relevance of the interplay between YAP and autophagy in the functional kidney proximal tubules” is an stretch and not supported by the included data.
24. Figure 7f needs some more detail explanation- Is P-AMPK staining restricted to PC. Two gamma-tub dots in the same cell are observed. Do they represent two PC?

Reviewer #2 (Remarks to the Author):

In this study, Claude-Taupin et al. investigate the crosstalk between autophagy and YAP/TAZ signaling in renal physiology and in the context of shear stress. Physiological shear stress leads to re-localization of YAP/TAZ into the cytoplasm. YAP/TAZ inactivation by flow appears to be required to activate autophagy in PTCs. Cilium-dependent activation of AMPK induces the nuclear export of YAP, dependent on SIRT1, and

inhibits the expression of autophagy repressors. In addition, the authors describe AMPK-induced phosphorylation of YAP, leading to cytoplasmic retention. It is exciting to see that under pathophysiological high shear stress, nuclear YAP increases as compared to normal shear stress. Moreover, the authors use both zebrafish and the UO model in mice to underline the *in vivo* relevance of their findings.

I really like the at hand study. The concept has a high level of novelty and could be highly relevant for the broad cilia community as well as for nephrologists. Data quality is very good and the presentation is very clear! I have only a few minor issues that should be addressed by the authors:

1. The authors show nicely that the KD of KIF3a decreases the effects of shear stress on YAP/TAZ localization. However, given that cilia are still present in KIF3a KD cells, although shorter in length, this could also be caused by extraciliary functions of Kif3a. Therefore, additional models without cilia should be used at this point. What happens if cells entirely lack primary cilia?

2. When applying pathological shear stress on the cells (Fig 7), the role of cilia remains unclear. Is this effect modulated by cilia as well?

3. The constitutive active/inactive mutants of YAP/TAZ should be described in more detail in the methods.

4. Efficiency of YAP/TAZ siRNAs should be shown.

5. Taken together, I feel that the study does not distinguish clearly enough between „normal“ shear stress and the role of primary cilia. This should also be clearer in the Ext. Fig 7. If the authors include the cilium in their model, this requires more data. Currently, the conclusion that cilia play a role in response to shear stress is based on rather weak evidence.

Reviewer #3 (Remarks to the Author):

In this manuscript, Claude-Taupin and colleagues investigated connections among fluid flow, YAP/TAZ, and autophagy. The authors demonstrated that 1) shear stress attenuated YAP/TAZ signaling, 2) shear stress-induced autophagy is further enhanced upon YAP/TAZ gene silencing, and 3) AMPK and SIRT1 are likely involved in YAP/TAZ regulation under the fluid flow in mouse kidney epithelial cells (KECs). While

this manuscript has certain degree of merit, some of the major concerns (e.g., preliminary nature of data, missing controls, overinterpretation of findings, choice of cell line, limitations in experimental approaches and lack of convincing proof in animal models of acute or chronic kidney disease, diminish the enthusiasm for the study. The following experimental concerns needs to be addressed comprehensively.

1. Virtually all in vitro studies are performed in mouse KECs which questions the physiological or pathological relationship of this study to humans. Most of the key experiments need be repeated in human kidney epithelial cells linking shear stress, YAP/TAZ and autophagy.
2. Throughout this study, genetic manipulations are done by transient transfections which raise concerns about gene knockdown or expression efficiency. Moreover, expression of YAP 5SA and YAP 5SA/S94A constructs were not validated and distinguished from endogenous YAP by western blotting in Figure 2f-i. Similar concerns can also be raised for indistinguishability for YAP (WT) and YAP (S61A) expression constructs by immunofluorescence microscopy in Fig 4c&d. The authors should consider stable cell line generation to bypass low efficiency issues associated with transient approaches.
3. Most of the interpretation on YAP, TAZ and SIRT1 nuclear levels in Figure 1, 3, 4, 5 and some extended figures were based only on immunofluorescence (IF) microscopy. Given that some experimental differences are marginal (e.g., 75% vs 81% difference in SIRT1 nuclear levels by shear in Fig 5a and Fig 4f)-both of which barely make the statistical significance-the authors should use nuclear fractionation experiments to corroborate IF data in a comprehensive manner.
4. Implication of key pathways such as AMPK in YAP and autophagy were based on inhibitor studies, but not genetic manipulation (Figure 4 and extended data). Non-specific effects of inhibitors are not taken into consideration or ruled out.
5. Similarly, SIRT1 involvement was not confirmed by gene silencing studies (Figure 5 and extended data). Data in Figure 5a were not representative of histogram quantitation in figure 5b. Western blot in Fig 5c is highly overexposed. Also, immune blots in the Extended data Figure 4a are also overexposed.
6. Some in vitro data were overinterpreted. For example, “YAP/TAZ inactivation during shear stress is Hippo independent” were not supported by genetic silencing or overexpression of core-Hippo kinases (Extended data Fig 4). Statements in lines (218-220) were not also supported by experiments.
7. In vivo relationship between YAP and autophagy using tissue sections was correlative at best without definitive implication of YAP in renal autophagy during UUO progression. Genetic manipulation of YAP in mice (e.g., tubular specific YAP ablation in mice) is necessary to establish causative role of this protein in

renal autophagy and disease progression. Figures 6e and 6f are not convincing. Moreover, published studies indicate that YAP total protein levels are increased during progressive UUO, which complicates interpretation of YAP nuclear localization in both of these figures.

8. Role of YAP and autophagy in AKI and CKD are disease-type or context specific (PMID: 32704047,30072422, 32843569). These findings are not discussed adequately. Moreover, authors may consider inclusion of a second renal injury model to explore relationship between YAP activation/upregulation and autophagy in CKD progression.

Reviewer #4 (Remarks to the Author):

Aurore Claude-Taupin et al show that nuclear YAP/TAZ negatively regulates autophagy machinery in kidney epithelial cells subjected to fluid flow. And reported that this crosstalk is supported by a primary cilium-dependent activation of AMPK and SIRT1, independently of the Hippo pathway. Then, they made a conclusion that dysregulation of this pathway is associated with the early onset of CKD. Although the crosstalk between autophagy and YAP/TAZ pathway has not been reported in the kidney field, in other organs and cell types, a couple of studies have demonstrated the association between these two events. Several issues should be further addressed.

1) In this study, the author used UUO model, several other models such as IRI induced CKD and DKD model should be employed to test the interaction of crosstalk between autophagy and YAP/TAZ pathway.

2) In addition to zebrafish and mouse model, the association should be further examined in the CKD patient kidney biopsies.

3) In addition to YAP/TAZ, whether the other components are modulated by flow Intensity in tubular cells should be deciphered.

4) The role for Teads in regulating autophagy should be tested in this model.

5) In Fig 2e and 2g, the data for the colocation of TAZ and LC3 should be represented.

Reviewer #1 (Remarks to the Author):

This is a manuscript by Claude-Taupin et al. investigating a relationship between YAP-dependent autophagy, primary cilium and fluid flow, and its potential role in kidney disease. While the topic is interesting, it is felt that claims are not supported by the experimental data and that the physiological relevance is not sufficiently proven. Also, there is a heavy reliance on immunofluorescence for quantitative analysis. Confirmation of critical data by other more quantitative techniques is felt missing. Specific details are below:

We would like to thank the reviewer for highlighting that the topic of this study is interesting.

1. All immunofluorescence figures: Lower magnification with larger field of view pictures are needed to assess changes occur in the majority of the cells. This is especially relevant in those experiments in which more quantitative techniques (i.e. cell fractioning) are not possible. Confirmation by WB needs to be included in key data.

We now provide lower magnification pictures (Main Fig. 1a,c; 4a,c,f; 6a,e and Extended Data Figure 1a,c) and we believe that the reviewer and readers will now better appreciate the changes in cells submitted to shear. As mentioned by the reviewer (here and also in point 15), cell fractionation is not possible on cells submitted to shear as we cannot harvest enough cells for this technique. However, as suggested by the reviewer, we have now performed Western Blotting (WB) experiments to confirm our immunofluorescence data on key figures (see Fig. 2,3,5,6 and Extended data Fig. 1,2,6), as described below:

First, we have confirmed by WB the effect of YAP/TAZ on autophagy in mouse and human kidney epithelial cells (KECs and HK2, respectively) by using siRNA, CRISPR-Cas9 KO of YAP (see also point 2 reviewer #3) and stable cells expressing mutant forms of YAP (see also point 2 reviewer #3 and point 1 reviewer #3), (Figures 2, 3 and Extended Data Figure 2e,g). Second, we now also provide WB experiments on in vivo samples to confirm the phosphorylation of YAP (serine 61) by AMPK (see also point 20 reviewer #1 and Figure 5k-m). Third, we have now performed immunoprecipitation and WB experiments to complete the findings shown in former figure 5. As illustrated in Figure 6l, we show the decrease of acetylated YAP protein levels in cells submitted to shear compared to static conditions. These data confirm that YAP is deacetylated upon physiological shear stress and highly suggest that the deacetylase SIRT1 is indeed involved in the regulation of YAP/TAZ activity. Lastly, we also provide WB experiments on another phosphorylated YAP (S397), previously described to inhibit YAP/TAZ activity in a LATS-dependent manner. As shown in Extended Data Fig. 1e,f, we did not detect any effect of shear stress on the levels of phosphorylated YAP (S397) compared to static conditions (1 day). This effect is in line with our previous data (former Extended Data Figure 4a,b, now in Extended Data Figure 5a,b) showing no effect of shear stress on phospho S127 YAP.

2. Figure 1: Data shows YAP-TAZ is constitutively active located in the nucleus in KECs. Any comment on this? Is this seen in other cell types?

We thank the reviewer for this remark and now provide new images (Figure 1a and 1c), which better reflect the quantification illustrated on Figure 1b and 1d. One can now

better appreciate that YAP/TAZ is mostly inactive (around 60% of the staining is located to the cytoplasm) in static conditions, while the cytoplasmic location of YAP/TAZ is further increased in shear stress conditions.

To answer the second question, we have performed new experiments on human kidney epithelial cells (HK2). This also refers to point 1 of reviewer #3. As shown in (Extended Data Figure 1g-j), we observed that YAP/TAZ is mostly in the cytoplasm in static conditions; however, like for KECs, the nuclear exclusion of YAP/TAZ is further induced in cells submitted to shear stress. This inhibition of YAP/TAZ activity is also shown using the TEAD reporter (Extended Data Figure 1k). Thus, the same effect on YAP/TAZ localization can also be seen in other cell types.

3. Extended Figure 1: No increased in LC3-II is observed in D1 cells treated with CQ. Any explanation for this?

We thank the reviewer for raising this important point, and we have repeated the LC3 Western Blotting experiments with a higher concentration of CQ (10 μ M instead of 5 μ M shown on the former Extended Data Figure 2) in KECs knocked down for YAP (or TAZ) submitted to shear stress for 1 day. As illustrated in the new Figure 2e-h, we observe a striking increase of LC3-II in cells treated with CQ (not only in shear but also in static), which was also observed in HK2 (Extended Data Fig. 2e-h). On the previous version of the figure (former Extended Data Figure 2), we were using a lower concentration of CQ because the concentration of 10 μ M is toxic for KECs after 4 days (D4) of shear stress. The data at D4 (using 5 μ M) are now shown on separate blots in the new Extended Data Figure 2a and 2c.

YAP and LC3 levels in static conditions at D4 should be included.

We now provide YAP and LC3 blots in static conditions at D4 (shown in new Extended Data Figure 2a). As illustrated, we did not observe any significant difference on YAP and LC3 levels in static condition at D4.

4. Extended Figure 1: YAP protein levels are affected by CQ treatment (also in observed in extended Fig 2) and shear stress itself. Any comment on that?

We thank the reviewer for giving us the opportunity to comment on our data showing that YAP protein levels are affected by chloroquine (CQ) treatment and shear stress itself.

In regards to the effect of shear stress on YAP protein level, we first checked the mRNA levels of YAP by RT-qPCR. As illustrated in Figure R1 (only for the reviewer) we observed that shear stress is able to upregulate the transcription of YAP. The increase in YAP protein levels (shown in the former Extended Data Fig. 2) is therefore at least partially due to the upregulation of gene expression upon shear. To go further, we have also investigated the effect of shear stress on the stability of the YAP protein. To that aim, we have measured proteasomal activity (using a fluorescent proteasomal kit) of cells submitted to shear. As illustrated below (Figure R2, only for the reviewer), we observed an inhibition of proteasomal activity suggesting that the effect of shear stress on YAP protein levels could also be partially due to an inhibition of proteasomal activity.

Figure R1: Shear stress induces the transcription of YAP mRNA in KECs.

Expression of Yap in KECs subjected to flow (shear) or not (static) during 24h (d1). mRNA levels were quantified by real-time RT-qPCR, normalized to β -actin and are presented as fold increases. Data show the mean \pm s.e.m.; $n = 3$ independent experiments, t-test.

Figure R2: Effect of shear and chloroquine (CQ) treatment on the proteasomal activity in KECs.

Proteasomal activity (measured using the 20S Proteasome activity assay, Sigma, #APT280) in KECs subjected to flow (shear) or not (static) during 24h (D1) in the presence or absence of chloroquine (CQ, 10 μ M). Data show the mean \pm s.e.m.; $n = 3$ independent experiments, t-test.

In regards to the putative effect of CQ on YAP protein levels, as illustrated in former Extended Data Fig. 2, we could not observe a significant decrease in YAP protein levels in CQ-treated KECs subjected to shear, as shown in new Figure 2e and Extended Data Fig.2a. While previous reports have shown that YAP is a substrate for autophagic degradation in static conditions (Lee et al., Nat Commun, 9, 4962- 2018; Wang et al., Cell Death Dis, 10, 432, 2019), we did not observe here a significant relationship between autophagy activity and endogenous YAP protein levels. In order to explain this, we have measured proteasomal activity in cells treated with or without CQ upon shear stress. As shown in Figure R2 (only for the reviewer), we could observe a slight but significant upregulation of proteasomal activity in CQ-treated cells (compared to untreated), suggesting that the effect of CQ on YAP protein levels during shear stress is due to an induction of YAP proteasomal degradation.

5. Cell density was not evaluated; therefore, the claim that YAP is mainly located in the cytosol in cell at high density is not accurate. There is no information in methods whether cells are cultivated and high or low density conditions. The discussion about this is a little distracting and it is not clear how it does support the main topic.

We apologize for being unclear in the manuscript. First, it is important to mention that the cells used in this study are all cultivated at high density. We seed the cells on microslides at high density to reach 100% confluency on the same day and then wait one more day to begin experiments. This time period allows cells to induce proper primary cilium (PC) formation, since we know that the PC is one of the mechanosensors involved in sensing shear in kidney epithelial cells. This standard protocol is used for

every single experiment (static and shear) done in this study. To go further, we decided to quantify the cell density. As illustrated in figure R3 (only for the reviewer), we could not observe any difference between static D1 and shear D1: cells are 100% confluent. As suggested by the reviewer, we now better describe this protocol in the methods section. Knowing the fact that cell density drastically impacts the activation of YAP, we think that we should not overuse the term “high density” in our manuscript since it could create a confusion for readers. We therefore removed these words where suitable. The results section of the manuscript (page 5) and the legend of the Extended Data figure 1 have been modified accordingly.

Figure R3: Shear stress does not impair the cell density. Cell density (measured using Icy software and a macro to evaluate cell confluency in immunofluorescence pictures) in KECs subjected to flow (shear) or not (static) during 24h (d1). Data show the mean \pm s.e.m.; $n = 3$ independent experiments, t-test.

6. Have expression levels of Ptplad2 and Ctp in D0 and D1 been evaluated to confirm that the downregulated expression is specific to fluid flow or in contrast YAP/TAZ regulation based in culture conditions?

To answer this point, we performed RT-qPCR to evaluate the expression levels of Ptplad2 and Ctp at D0 (static D0) and at D1 (static D1) and compared to shear condition (Shear D1) (previously shown in former Extended Data Figure 1e,f). As shown in Figure R4 (only for the reviewer), we did not observe any effect between static D0 and static D1. However, we could confirm the effect of shear stress on the expression of Ptplad2 and Ctp. Altogether, these data show that this effect is specific to fluid flow.

Figure R4: The inhibition of Ptplad2 and Ctp mRNA expression is specific to shear stress

Expression of Ptplad2 (a) and Ctp (b) in KECs subjected to flow (shear) or not (static) during 24h. mRNA levels were quantified by real-time RT-qPCR, normalized to β -actin and are presented as fold increases. Data show the mean \pm s.e.m.; $n = 3$ independent experiments, t-test.

7. On Page 6, authors should clarify what exactly “autophagy flux is amplified”. Is autophagy activated? Is autophagy flux increased under same autophagy activation?

We apologize for being unclear concerning our data on the effect of YAP/TAZ on the autophagic pathway (autophagy initiation and autophagic flux). To better analyze this effect and to strengthen the data shown in the former Figure 2 and Extended Data Figure 2, we have performed two different sets of new experiments:

First, we used the tandem RFP-GFP-LC3 probe to quantify the number of autophagosomes (GFP⁺RFP⁺) and autolysosomes (GFP-RFP⁺) in cells submitted to shear. As illustrated in new Figure 2i,j, we observed an increase in autolysosome numbers after the knockdown of YAP (or TAZ) compared to control cells (Scr: scramble). However, the loss of YAP (or TAZ) did not impair the number of autophagosomes in cells subjected to shear, highly suggesting that YAP and TAZ negatively regulate autophagic flux (fusion of autophagosomes with lysosomes) in cells submitted to fluid flow. Since we did not observe any decrease in autophagosome numbers in cells knocked down for YAP (or TAZ) compared to scramble cells, we wanted to go further to make sure that the loss of YAP and TAZ only play a role in stimulating autophagic flux (and not in inducing the initiation of autophagosome formation as well). To do so, we have quantified the colocalization between two early markers of the autophagic pathway (ATG16L1 and WIPI2) in YAP (or TAZ) deficient cells (new Extended Fig. 2i,j). We did not observe any difference in the number of ATG16L1⁺WIPI2⁺ puncta in cells knocked down for YAP (or TAZ) compared to the scramble condition. Altogether, we conclude that YAP (and TAZ) only play a role in inhibiting autophagic flux in cells subjected to shear. This result is in line with the LC3-Western blot data shown in Figure 2e-h.

8. On Page 7, the claim “even at high density, condition known to block YAP-dependent autophagy in static conditions (Fig 2a-d)” needs clarification. What is YAP-dependent autophagy? Which figures do exactly back up this claim?

We are sorry for being unclear on page 7. YAP-dependent autophagy means a regulation of autophagic activity by YAP. We refer here to two different molecular pathways. The first is a positive effect of YAP on autophagy as previously demonstrated by Pavel et al., Nat Commun 2018, PMID: 30054475 and Totaro et al., PNAS 2019, PMID: 31416916. This regulation occurs in cells cultured on soft/stiff extracellular matrix and in cells cultivated in high/low cell density conditions. The second one is a negative effect of YAP on autophagic flux, reported here in cells submitted to shear stress. Indeed, we now demonstrate using chromatin immunoprecipitation (ChIP) assays (new experiment in Fig. 3j, see also point 13 below), that YAP can bind to the Rubicon gene, a negative regulator of autophagy (Rubicon) and regulates its expression in static conditions (Figure 3g,h). Thus, the effect of YAP on autophagy is dual and differs between different mechanical cues.

To remove possible confusion in our previous text, we have deleted the term “YAP-dependent autophagy” (see results section on page 7). Additionally, we noticed that we have also used this term in the title of this manuscript and have slightly changed it as well.

9. Figure 2e-f: CA YAP seems to change LC3 cellular localization. How were puncta counted in these figures, they do not seem very defined. LC3-II blots are needed.

We thank the reviewer for this comment. Regarding the protocol used to quantify LC3 puncta, we used ICY software and the “spot detector” tool. It is important to mention that only the cytoplasmic LC3 puncta were quantified. In addition, since the LC3 puncta were not very well defined, we have decided to replace the pictures that were in former Fig.2e (transient transfection of YAP 5SA or YAP 5SA/S94A) with pictures where LC3 puncta are more defined from KEC-YAP^{KO} stably expressing YAP-5SA or YAP 5SA/S94A (New Fig.3c,d; designed for this study, see also point 2 for reviewer #3), as the transient

overexpression of YAP 5SA in cells expressing endogenous YAP could have led to some deleterious effects on LC3. We hope that the reviewer can now appreciate that the expression of YAP 5SA does not change LC3 localization. Lastly, and as suggested by the reviewer, we now provide LC3 blots on cells expressing CA-YAP or the inactive (5SA/S94A) form of YAP. As illustrated in Figure 3a,b, we now observe an induction of autophagic flux in cells expressing the 5SA/S94A mutant (compared to the CA-YAP form). Altogether, these data confirm our previous immunofluorescence findings shown in former Figure 2e-f.

10. On Page 7, “indicating that YAP/TAZ inhibition by fluid flow is necessary to allow autophagy induction in KEC” is not supported by accompanying figures; autophagy activation is not seen.

We thank the reviewer for raising this important point. We first generated stable KEC lines stably expressing CA-YAP or 5SA/S94A inactive form of YAP (see previous point 9). This generation was done in our KO CRISPR clone of YAP. Then, we performed LC3 Western blots. As illustrated in the new Figure 3a,b, we now observe an induction of autophagic flux only in cells expressing the 5SA/S94A mutant while cells expressing CA-YAP showed an inhibition of autophagy flux upon shear stress, as CQ treatment did not impact LC3-II protein levels. These data now better support our statement on page 8. We modified it accordingly since now we know that autophagy flux is inhibited by the CA-YAP mutant: “indicating that YAP/TAZ inhibition by fluid flow is necessary to increase autophagy flux in kidney epithelial cells.”

11. Figure 2g- CA YAP is located in the cytoplasm. Is this the expected location?

It is important to mention that, as expected, we could observe that CA-YAP is located in the nucleus in static cells (Figure R5, only for the reviewer). However, the reviewer is correct as we could observe some (but not all) cytoplasmic staining of CA-YAP (former Figure 2g, now in Figure 3c) in cells submitted to shear stress. The unexpected localization of CA-YAP in cells submitted to shear could be explained by the activation of SIRT1, which induces YAP nuclear exclusion in cells submitted to shear (see point 20).

Figure R5: CA-YAP is located in the nucleus in cells cultivated in static condition.

Representative images of CA-YAP localization in the nucleus, labeled with DAPI, in KECs subjected to flow (shear) or not (static) during 24h. Scale bars, 10 μ m

12. Figure 2i- have expression of Rubicon in cells with CA and inactive YAP been quantified?

We thank the reviewer for this question. To answer this point, we have used our KEC-YAP^{KO} cells stably expressing CA and inactive form of YAP (see point 9 for details for the generation of these cell lines). We have checked the expression of Rubicon at the mRNA level and observed low levels of Rubicon in cells expressing the 5SA/S94A form of YAP (in static or shear conditions) compared to the CA-YAP (5SA) mutant (see new

Figure 3h). This second set of data confirms that the regulation of Rubicon transcription is dependent on the activity of YAP.

13. On Page 8: The claims that “shear stress downregulates the expression of YAP/TAZ target genes, including several autophagy inhibitors and allows for autophagy activation” is not supported by the data.

We agree with the reviewer. First, it is worth mentioning that chromatin immunoprecipitation (ChIP) was not possible on cells cultured in microslides (for static or shear condition) as we cannot harvest enough cells for this technique. Cells are seeded (at 5×10^5 cells per microslide) in a very small and confined environment in which we cannot harvest them by scraping. Nevertheless, we have decided to perform ChIP using anti-GFP antibody in our KEC-YAP^{KO} cells stably expressing GFP-CA or inactive form of YAP (as a negative control) cultured in petri dishes, as Rubicon mRNA levels were similar in these cells in static and shear conditions (new Figure 3h). As illustrated in the new Figure 3i,j, we demonstrate that only the CA form of YAP, and not the constitutively inactive one, is able to bind to the Rubicon gene. As a positive control, we checked the binding of GFP-YAP 5SA on CTGF (Fig. 3i). These new data now better support our claim on page 8 and is also in agreement with the literature showing a YAP (or TEAD2) ChIP-seq peak in the mouse Rubcn gene (Tremblay et al., Cancer Cell 2014, PMID: 25087979 and Diepenbruck et al., J Cell Sci 2014, PMID: 24554433), available online in the cistrome database (<http://cistrome.org/>).

14. Figure 3a-d: All the data included in these figures need to be confirmed by WB. LC3 puncta does not seem reliable based on the poorly defined and overexposed figures.

We assume here that the reviewer is talking about the former figure 2a-d (and not 3a-d). To answer this point, we now provide LC3 blots. As illustrated in figure 2e-h; we now observe the effect of the loss of YAP (or TAZ) on LC3 lipidation.

15. Figure 3e: This should be confirmed by cell fractioning- if not possible based on cell quantity in fluidic chambers, lower magnification pictures must be shown.

We now provide lower magnification pictures on Figure 3 (now in Figure 4). We hope that the reviewer and readers will now better appreciate the changes in YAP/TAZ localization in cells submitted to shear after a knockdown of KIF3A (Now in Fig.4), which shortens primary cilia. As mentioned by the reviewer, cell fractionation is indeed not possible on cells submitted to shear as we cannot harvest enough cells for this technique. We now also confirm the role of primary cilium in YAP activity by knocking down CEP164, to obtain cells lacking primary cilia (Figure 4f-g and Extended Data Fig. 4d-f; as suggested by reviewer 2, point 1).

16. Extended Figure 4- any comments on why YAP, TAZ and LATS1 protein levels are elevated with shear stress?

We thank the reviewer for giving us the opportunity to comment on our data showing that YAP, TAZ and LATS1 protein levels are elevated with shear. This point also refers to the point 4 above.

First, we show by RT-qPCR an increase of YAP and TAZ at mRNA levels (Figure R6, only for the reviewer) in cells submitted to shear stress (compared to the static condition). Second, and as discussed above (see point 4), we observed an inhibition of proteasomal activity in cells subjected to shear (compared to the static condition)

(Figure R7, only for the reviewer). Overall, the transcriptional regulation of YAP/TAZ as well as the possible decrease in YAP/TAZ proteasomal degradation could explain why YAP and TAZ protein levels are elevated with shear stress.

Figure R6: Shear stress induces the transcription of YAP and TAZ in KECs.

Expression of Yap (a) and Taz (b) in KECs subjected to flow (shear) or not (static) during 24h. mRNA levels were quantified by real-time RT-qPCR, normalized to β -actin and are presented as fold increases. Data show the mean \pm s.e.m.; $n = 4$ or 5 independent experiments for Yap and Taz, respectively, t-test.

Figure R7: Shear stress inhibits the proteasomal activity in KECs.

Proteasomal activity (measured using the 20S Proteasome activity assay, Sigma, #APT280) in KECs subjected to flow (shear) or not (static) during 24h (d1). Data show the mean \pm s.e.m.; $n = 3$ independent experiments, t-test.

Regarding the effect of shear on LATS1 protein levels, only a post-transcriptional regulation seems to explain why LATS1 is increased with shear. We indeed observed a decrease (and not an increase) of LATS1 mRNA levels in cells subjected to shear (Figure R8 only for the reviewer).

Figure R8: Shear stress decreases the mRNA levels of LATS1 in KECs.

Expression of LATS1 in KECs subjected to flow (shear) or not (static) during 24h (d1). mRNA levels were quantified by real-time RT-qPCR, normalized to β -actin and are presented as fold increases. Data show the mean \pm s.e.m.; $n = 3$ independent experiments, t-test.

17. Figure 4g,h: Lipophagy was not monitored; therefore, the claims at the end of Page 9 are not accurate.

We agree with the reviewer that lipophagy was not monitored. To answer this point, we have now performed loss of function experiments of Atg5 and have repeated the experiments shown in the former Figure 4g,h. As illustrated in new Fig. 5g,h, we observe

a degradation of lipid droplets upon shear in scramble conditions, but this degradation is impaired in autophagy-deficient cells (siATG5), demonstrating that lipophagy is indeed implicated in LD removal in cells subjected to shear stress. These data are in line with our previous published findings (Miceli et al, Nat Cell Biol 2020; PMID: 32868900). We also performed similar experiments in cells treated with dorsomorphin and as expected, we were not able to see any lipophagic activity in cells treated with dorsomorphin (Fig. 5g,h), with no additive effect in siATG5 cells.

18. Figure 5: Low magnification pictures are needed. Figure 5e shows more cytosolic YAP in EX527-treated cells, which is not shown in the graph.

We now provide low magnification pictures in former Figure 5 (now Figure 6e). We hope that the reviewer and readers will now better appreciate the effect of shear on SIRT1 localization and the effect of EX527 on YAP localization, which now are better reflecting the quantification shown in Figure 6f.

19. Extended Figure 5a- The decreased in P-YAP with Dorso is not convincing. Quantification is needed.

We now provide the quantification of the blot shown in Extended Data Figure 6c (former Extended Data Figure 5a) in Extended Data Fig. 6d. To go further, we have used two complimentary strategies to confirm the effect of dorsomorphin on YAP phosphorylation at S61. First, we performed loss of function experiments of LKB1, a key kinase upstream of AMPK. LKB1 is known to be activated in kidney epithelial cells submitted to shear stress (Boehlke C et al., Nat Cell Biol 2010; PMID: 20972424). As illustrated in Fig. 5i,j, we observed a decrease in the phosphorylated form of YAP (S61) in cells knocked down for LKB1. Second, we have analyzed the phosphorylation status of YAP in in vivo samples by using kidney samples from mice knocked out for the alpha1 subunit of AMPK. In contrast to the WT control samples, we could confirm the effect of the loss of AMPK alpha 1 on the phosphorylation status of YAP (S61) in the kidney (new Figure 5k-m). Altogether, these new data confirm our previous findings obtained with the inhibitor of AMPK (dorsomorphin), showing an AMPK-dependent phosphorylation of YAP at S61 (New Fig. 5i-m).

20. Extended Figure 5b- Statistics in DMSO and EX527 under shear stress are not convincing.

We agree with the reviewer on this point. Our previous data shown in Extended Data Figure 5b (now in Extended Data Fig.7c) were not convincing enough even though we could observe a significant increase of YAP/TAZ activity upon EX527 treatment, measured by luciferase assay. To strengthen our conclusion, we repeated this experiment in cells overexpressing a mutant form of SIRT1 unable to enter the nucleus (SIRT1 harboring a mutated nuclear localization signal: NLS^{mut}; Tanno M et al., JBC 2007; PMID: 17197703). As illustrated in Fig. 6k, we observe more YAP/TAZ activity upon shear in cells overexpressing the NLS^{mut}-SIRT1, compared to cells overexpressing the WT form of SIRT1. These data confirm that nuclear SIRT1 is able to control the activity of YAP/TAZ in cells submitted to shear stress. Since this effect is not complete, we believe that other proteins such as XPO1, already known to regulate the nucleo-cytoplasmic shuttling of YAP (Martin AP et al. EMBO Rep 2019; PMID: 31544310), could participate in the induction of YAP/TAZ nuclear exclusion. To go further, we repeated this experiment in cells overexpressing another mutant form of SIRT1, unable to exit the nucleus (SIRT1 harboring a mutated nuclear export signal (NES^{mut}; Tanno M et al., JBC 2007; PMID: 17197703). As illustrated in Fig. 6k, we observed decreased levels of YAP/TAZ activity upon shear in cells overexpressing the NES^{mut}-SIRT1, compared to

cells overexpressing the NLS^{mut} form of SIRT1. This second piece of data confirms that nuclear SIRT1 induces the nuclear exclusion of YAP/TAZ upon shear in kidney epithelial cells.

21. Figure 6 c,d: How machine learning was authenticated?

For each channel (i.e Hoechst (Blue) and Wt1b-GFP (tubules, green)), Ilastik Pixel classification (v1.3.3post3) models were created with live prediction using at least 3 representative images of all data analyzed. The model was then validated when all predictions on representative images were visually correct. Using Fiji (v2.3.0/1.53f51), we then checked each image (one by one) and corrected, if necessary, the tubules (Wt1b-GFP) and nuclei masks created with ilastik, in order to finally reach the ground truth. Finally, we measured different local backgrounds to estimate the mean background and measured the ratio of mean YAP intensity with background subtraction into corrected 3D ROIs, inside and outside the nucleus. We now better describe this authentication in the methods section (page 51).

22. Figure 6e: YAP staining needs to be shown in red instead of b/w. As shown, the increased nuclear translocation is not appreciated.

As suggested by the reviewer, we now provide pictures in colors (red) and not in black and white. We hope that the reviewer and readers will now better appreciate the increase of nuclear translocation shown in the former Figure 6e (now in Figure 7e).

23. Claim on Page 12: “using two independent in vivo models, we confirmed the relevance of the interplay between YAP and autophagy in the functional kidney proximal tubules” is an stretch and not supported by the included data.

We agree that our claim on page 12 was not supported by the previous data. As requested by reviewer #3 (see point 7) and #4 (see point 1), new sets of in vivo data are now shown and support this claim (new Figures 9 and 10).

24. Figure 7f needs some more detail explanation- Is P-AMPK staining restricted to PC. Two gamma-tub dots in the same cell are observed. Do they represent two PC?

We thank the reviewer for giving us the opportunity to better explain the data shown in the former figure 7f (now in Figure 8f). The two gamma-tubulin positive dots do not represent two different primary cilia (PC) but define the daughter centriole and the basal body, which is derived from the mother centriole of the centrosome. The axoneme of the solitary PC extends from the basal body. As shown in Figure 9f, P-AMPK labeling seems to not be restricted only to the basal body (i.e. the base of PC), even if the fluorescence intensity signal seems to be higher in the basal body compared to the daughter centriole. This observation could be explained by the fact that the basal body and the daughter centriole are considered to exist as pairs with only the mother one capable of nucleating cilia. So far, this point, beyond the scope of this study, would warrant further investigations.

Reviewer #2 (Remarks to the Author):

In this study, Claude-Taupin et al. investigate the crosstalk between autophagy and YAP/TAZ signaling in renal physiology and in the context of shear stress. Physiological shear stress leads to re-localization of YAP/TAZ into the cytoplasm. YAP/TAZ inactivation by flow

appears to be required to activate autophagy in PTCs. Cilium-dependent activation of AMPK induces the nuclear export of YAP, dependent on SIRT1, and inhibits the expression of autophagy repressors. In addition, the authors describe AMPK-induced phosphorylation of YAP, leading to cytoplasmic retention. It is exciting to see that under pathophysiological high shear stress, nuclear YAP increases as compared to normal shear stress. Moreover, the authors use both zebrafish and the UUO model in mice to underline the in vivo relevance of their findings.

I really like the at hand study. The concept has a high level of novelty and could be highly relevant for the broad cilia community as well as for nephrologists. Data quality is very good and the presentation is very clear! I have only a few minor issues that should be addressed by the authors:

We would like to thank the reviewer for highlighting the relevance of our study. We answered below his/her few minor concerns.

1. The authors show nicely that the KD of KIF3a decreases the effects of shear stress on YAP/TAZ localization. However, given that cilia are still present in KIF3a KD cells, although shorter in length, this could also be caused by extraciliary functions of Kif3a. Therefore, additional models without cilia should be used at this point. What happens if cells entirely lack primary cilia?

We thank the reviewer for raising this important point, which we have addressed by performing new experiments by knocking down CEP164, a centriole appendage protein required for primary cilium formation. We now provide new panels (see new Figure 4f-h and Extended Data Figure 4d-f) showing (i) an inhibition of primary cilium formation in CEP164-deficient cells (ii) an inhibition of YAP nuclear translocation in CEP164-deficient cells subjected to shear and (iii) increased YAP/TAZ activity (measured by luciferase assay) in CEP164-deficient cells. Altogether, these data strongly confirm that primary cilia play a role in controlling YAP/TAZ activity in kidney epithelial cells subjected to physiological shear stress.

2. When applying pathological shear stress on the cells (Fig 7), the role of cilia remains unclear. Is this effect modulated by cilia as well?

We thank the reviewer for giving us the opportunity to better describe the role of cilia when cells are submitted to pathological shear stress. To that aim, we have downregulated Cep164 to obtain cells that entirely lack PC, as suggested by the reviewer above, to study YAP activity using the TEAD luciferase reporter. As illustrated in Extended Data Figure 8g, we observed a reduction of YAP/TAZ activity in PC-deficient cells subjected to pathological shear (compared to the Scr cells). PC have therefore a dual role. In physiological conditions, PC are important to stimulate YAP/TAZ nuclear exclusion, which is important to induce autophagy flux. However, PC play a reverse role in pathological conditions by promoting YAP/TAZ nuclear translocation and therefore downregulating the autophagic pathway. As YAP/TAZ activity is not efficiently reactivated in cells lacking PC upon pathological shear stress and AMPK activation is impaired at the base of PC in pathological conditions (versus physiological shear stress), we believe here that the possible increased bending of the PC between physiological and pathological conditions could lead to a different ciliary response. This could explain this dual role of PC in regulating YAP/TAZ activity and the autophagic pathway, depending on fluid flow intensity. This point, beyond the scope of this study, would warrant further investigations.

3. The constitutive active/inactive mutants of YAP/TAZ should be described in more detail in the methods.

We apologize for being unclear and we now better describe these mutants in the methods section (page 43) in this study. We also provide more information in Supplementary Table 2.

4. Efficiency of YAP/TAZ siRNAs should be shown.

Figures 2e,g and Extended Data Fig.2a,c,e,g are now showing the efficiency of YAP/TAZ siRNAs in KECs and HK2 cells.

5. Taken together, I feel that the study does not distinguish clearly enough between „normal“ shear stress and the role of primary cilia. This should also be clearer in the Ext. Fig 7. If the authors include the cilium in their model, this requires more data. Currently, the conclusion that cilia play a role in response to shear stress is based on rather weak evidence.

We fully agree with the reviewer that our previous data on the role of cilia in controlling YAP/TAZ activity were not robust enough. As suggested by this reviewer (see point 1), we now provide new experiments using a new cellular model without cilia, by using cells deficient for CEP164. We now provide new panels (see new Figure 4f-h and Extended Data Figure 4d-f) showing an inhibition of YAP nuclear translocation in CEP164-deficient cells subjected to shear correlated with an impairment in the reduction of YAP/TAZ activity (measured by luciferase assay). Altogether, these data confirm our previous findings obtained in KIF3A-depleted cells (Fig. 4a-e and Extended Data Fig.4a-c) and strongly confirm that cilia play a role in controlling YAP/TAZ activity in kidney cells subjected to shear in physiological but also in pathological conditions, as shown in Extended Data Fig.8g and highlighted in the schematic summary shown in Extended Data Fig.10.

Reviewer #3 (Remarks to the Author):

In this manuscript, Claude-Taupin and colleagues investigated connections among fluid flow, YAP/TAZ, and autophagy. The authors demonstrated that 1) shear stress attenuated YAP/TAZ signaling, 2) shear stress-induced autophagy is further enhanced upon YAP/TAZ gene silencing, and 3) AMPK and SIRT1 are likely involved in YAP/TAZ regulation under the fluid flow in mouse kidney epithelial cells (KECs). While this manuscript has certain degree of merit, some of the major concerns (e.g., preliminary nature of data, missing controls, overinterpretation of findings, choice of cell line, limitations in experimental approaches and lack of convincing proof in animal models of acute or chronic kidney disease, diminish the enthusiasm for the study.

The following experimental concerns needs to be addressed comprehensively.

1. Virtually all in vitro studies are performed in mouse KECs which questions the physiological or pathological relationship of this study to humans. Most of the key experiments need be repeated in human kidney epithelial cells linking shear stress, YAP/TAZ and autophagy.

We would like to thank the reviewer for giving us the opportunity to test our hypothesis in human kidney epithelial cells. We now provide several new figures panels (Extended Data Figure 1g-k; 2e-h, 7a,b and 8e-f) using HK-2 cells, which are proximal tubular cells

derived from a normal human adult male kidney. Of note, our lab previously published that shear stress stimulates the autophagic pathway in HK-2 cells (Boukhalfa A et al, Nature Communications 2020; PMID: 31941925). We showed (i) an inhibition of YAP and TAZ nuclear translocation by immunofluorescence in HK-2 cells submitted to physiological fluid flow (Extended Data Figure 1g-j) (ii) an inhibition of YAP/TAZ activity using the TEAD reporter when they were submitted to physiological fluid flow (Extended Data Figure 1k) (iii) an induction of the autophagic flux in cells knocked down for YAP (or TAZ) (Extended Data Figure 2e-h) (iv) an effect of EX527 (a SIRT1 inhibitor) on YAP nuclear translocation in cells submitted to physiological fluid flow (Extended Data Figure 7a,b) and (v) a reactivation of YAP activity in HK-2 cells subjected to pathological shear stress, compared to physiological shear stress (Extended Data Figure 8e,f). In addition, we studied the interplay between YAP and autophagy in renal biopsies from patients suffering of diabetic kidney disease (DKD), a leading cause of CKD and a pathological condition characterized by increased urinary flow. Remarkably, we confirmed the nuclear localization of YAP and the decrease fusion of autophagosomes with acidic compartments (LAMP1⁺) in DKD kidneys compared to control kidneys (Fig. 9 a-d). Altogether, these data show the relevance of our experimental findings in humans.

2. Throughout this study, genetic manipulations are done by transient transfections which raise concerns about gene knockdown or expression efficiency. Moreover, expression of YAP 5SA and YAP 5SA/S94A constructs were not validated and distinguished from endogenous YAP by western blotting in Figure 2f-i. Similar concerns can also be raised for indistinguishability for YAP (WT) and YAP (S61A) expression constructs by immunofluorescence microscopy in Fig 4c&d. The authors should consider stable cell line generation to bypass low efficiency issues associated with transient approaches.

We thank the reviewer for raising these important points, which we have addressed by providing new sets of experiments using stable cell lines. To do so, we have generated KECs knocked out (KO) for YAP (KEC-YAP^{KO}) using the CRISPR/Cas9 technology. We have first checked by western blot the absence of YAP signals in KO compared to a WT clone for YAP (New Figure 2k,l). Then, we also analyzed LC3 lipidation in cells submitted to shear (Figure 2k,l) and observed, as expected, an increase of the autophagic flux in YAP KO cells (compared to WT clone), confirming our experiments done with transient siRNA transfections.

In addition to this, we have also generated stable cell lines expressing GFP-YAP or mutants form of YAP (GFP-YAP 5SA, GFP-YAP 5SA/S94A or GFP-YAP S61A) using lentiviral vectors. This generation was done in our KEC-YAP^{KO} clone, to avoid any effect of endogenous YAP. We checked the LC3 lipidation by western blotting in cells expressing constitutively active (5SA)/inactive (5SA/S94A) mutants forms of YAP. As illustrated in Figure 3a,b, we could only observe a stimulation of the autophagic flux in cells expressing the inactive form of YAP (5SA/S94A compared to YAP 5SA). We confirmed these findings by immunofluorescence (New Figure 3c,d). Interestingly, the opposite effect was observed in cells expressing the mutant form of YAP (YAP S61A, compared to YAP WT), unable to be phosphorylated by AMPK, as observed by WB and IF analysis of LC3 (New Figure 5c,d and Extended Data Fig.6a,b). These data confirm and strengthen the conclusion we made with our previous experiments done with transient transfections (Former Figure 2e,f for constitutively active/inactive YAP mutants and Former Figure 4c,d for YAP S61A mutant).

3. Most of the interpretation on YAP, TAZ and SIRT1 nuclear levels in Figure 1, 3, 4, 5 and some extended figures were based only on immunofluorescence (IF) microscopy. Given that some experimental differences are marginal (e.g., 75% vs 81% difference in SIRT1 nuclear levels by shear in Fig 5a and Fig 4f)-both of which barely make the statistical significance-the authors should use nuclear fractionation experiments to corroborate IF data in a comprehensive manner.

This point also refers to point 1 of reviewer #1. As mentioned by reviewer #1, cell fractionation is not possible on cells submitted to shear. Unfortunately, we do not harvest enough cells for this technique. However, we have now performed Western Blotting (WB) and luciferase experiments to confirm our immunofluorescence data on key figures:

First, we performed WB experiments to complete the findings shown in the former Figure 5 on in vivo samples, to confirm the phosphorylation of YAP at S61 by AMPK (see point 19 for reviewer #1, point 4 for reviewer #3 and new Figure 5k-m). Second, we now provide new luciferase experiments with the TEAD reporter, showing the importance of the subcellular localization of SIRT1 in regulating YAP/TAZ activity. For this, we used a SIRT1 mutant unable to enter the nucleus (mutated NLS, NLS^{mut} from Tanno M et al., JBC 2007; PMID: 17197703), which led to increased TEAD activity, compared to the expression of WT SIRT1 (Figure 6k). These data were confirmed by the use of another mutant, unable to reach the cytoplasm because its nuclear export signal is mutated (NES^{mut}; Tanno M et al., JBC 2007; PMID: 17197703). Indeed, cells transfected with SIRT1-NES^{mut} showed decreased TEAD activity compared to SIRT1-NLS^{mut} transfected cells (new Figure 6k, see also point 20 for reviewer 1). Third, as illustrated in Figure 6l, we now show the decrease of acetylated YAP protein levels in cells submitted to shear compared to the static condition. These data highly suggest a role for the deacetylase SIRT1 in regulating YAP/TAZ activity.

4. Implication of key pathways such as AMPK in YAP and autophagy were based on inhibitor studies, but not genetic manipulation (Figure 4 and extended data). Non-specific effects of inhibitors are not taken into consideration or ruled out.

We agree with the reviewer. To rule out the non-specific effects of inhibitors, we have decided to use two different strategies. First, we have now performed loss of function experiments on LKB1, the main kinase upstream of AMPK known to be activated in kidney epithelial cells submitted to shear (Boehlke C et al, Nat Cell Biol 2010; PMID: 20972424). As illustrated in new Fig. 5i,j, we observed decreased levels of the S61 phosphorylated form of YAP in cells knocked down for LKB1. Second, we have also analyzed the phosphorylation status of YAP in vivo by analyzing kidney samples from mice knocked out for the alpha1 subunit of AMPK. In contrast to the WT control samples, we could confirm the effect of the loss of AMPK alpha 1 on the phosphorylation of YAP in kidney (new Figure 5k-m). Altogether, these new data confirm our previous findings obtained with the inhibitor of AMPK (dorsomorphin) shown in the former Figure 4e,f (now in Figure 5e,f).

5. Similarly, SIRT1 involvement was not confirmed by gene silencing studies (Figure 5 and extended data). Data in Figure 5a were not representative of histogram quantitation in figure 5b. Western blot in Fig 5c is highly overexposed. Also, immune blots in the Extended data Figure 4a are also overexposed.

We thank the reviewer for this remark. First, it is important to mention that the inhibition of the expression of SIRT1 protein directly impacts the stability of the YAP protein (Figure R9, only for the reviewer). Based on this finding, we decided to use another strategy to strengthen our data regarding the role of SIRT1 in controlling YAP/TAZ activity. As explained in comment #3 above, we overexpressed SIRT1 mutants (NLS^{mut} or NES^{mut}) and observed increased YAP/TAZ activity (TEAD reporter, luciferase assay) upon shear in cells overexpressing SIRT1-NLS^{mut}, a mutant form of SIRT1 that cannot enter the nucleus, compared to cells overexpressing WT SIRT1 (see Figure 6k). Accordingly, we found decreased YAP/TAZ activity upon shear in cells overexpressing SIRT1- NES^{mut}, a SIRT1 mutant that cannot exit the nucleus, compared to cells overexpressing SIRT1-NLS^{mut}. These data confirm that SIRT1 is able to control the activity of YAP/TAZ in cells submitted to shear stress in kidney epithelial cells. Since this effect is not complete, we think that other proteins such as XPO1 (already known to regulate YAP nucleo-cytoplasmic shuttling (Martin, AP et al. EMBO Rep 2019, PMID: 17197703) participate in the induction of YAP/TAZ nuclear exclusion.

Figure R9: The loss of SIRT1 decreases YAP protein levels in KECs during shear stress.

Representative images of YAP, SIRT1 and Actin proteins levels in KECs subjected to physiological shear stress for 1 day.

Lastly, we now provide new pictures (Figure 6a) that better reflect the quantification shown on the histogram in former Figure 5b (now Figure 6b). In addition, blots on former Figure 5c (now in Figure 6c) and former Extended Figure 4a (now in Extended Data Figure 5a) have also been changed.

6. Some in vitro data were overinterpreted. For example, “YAP/TAZ inactivation during shear stress is Hippo independent” were not supported by genetic silencing or overexpression of core-Hippo kinases (Extended data Fig 4). Statements in lines (218-220) were not also supported by experiments.

We agree with the reviewer that these in vitro data were overinterpreted. To address this point, we have now performed loss-of-function experiments of the core-Hippo kinases LATS1 and LATS2. As illustrated in Extended Data Fig. 5h-m, we observed that the loss of LATS1 and LATS2 proteins (by siRNA transfection) is unable to impair YAP/TAZ nuclear exit in KECs submitted to fluid flow, as shown by immunofluorescence. Thus, these data strongly suggest that YAP/TAZ inactivation during shear stress is independent of the canonical Hippo pathway.

7. In vivo relationship between YAP and autophagy using tissue sections was correlative at best without definitive implication of YAP in renal autophagy during UUO progression. Genetic manipulation of YAP in mice (e.g., tubular specific YAP ablation in mice) is necessary to establish causative role of this protein in renal autophagy and disease progression. Figures 6e and 6f are not convincing. Moreover, published studies indicate that YAP total protein levels are increased during progressive UUO, which complicates interpretation of YAP nuclear localization in both of these figures.

We would like to thank the reviewer for these insightful suggestions which can allow us to reinforce our main conclusions. As suggested by the reviewer, we have now generated kidney conditional knockout mice for YAP (by crossing Yap^{fl/fl} mice with KSP-CreER mice) that we have submitted to UUO in order to investigate the role of YAP in

renal autophagy and disease progression. Towards this goal, we decided to sacrifice mice 14 days after surgery, when lesions develop. Consistently, 14 days after UUU we observed severe tubular injury and interstitial fibrosis in obstructed kidneys of wild-type mice (Fig. 10a-d and Extended Data Fig.9c,d). We also observed increased levels of nuclear YAP in tubular epithelial cells (Figure 10e,f), which correlated with the accumulation of LC3-positive structures in obstructed kidneys (Figure 10g,h). Even though we did not observe any significant difference in the number of LC3-positive structures accumulating in the obstructed kidney of *Yap^{ATub}* as compared to *Yap^{fl/fl}* littermates (Figure 10g,h), we observed a difference in their size. Indeed, LC3-positive structures were smaller in *Yap^{ATub}* mice compared to *Yap^{fl/fl}* mice (Figure 10g,i), suggesting that the autophagy blockade in *Yap* mutant mice is less pronounced than in wild-type mice. Accordingly, we observed a significant accumulation of the autophagy substrate P62/SQSTM1 in wild-type mice but not in *Yap^{ATub}* mice (Figure 10j,k), suggesting that the accumulating autophagosomes observed during UUU could not undergo lysosomal degradation in wild-type mice but it is partially rescued in *Yap^{ATub}* mice. The autophagy blockade observed in wild-type mice 14 days after UUU is in line with previous data showing (i) a blockade of autophagy flux in mice 1 week after UUU (Lopez-Soler RI et al, *Front Immunol* 2023; PMID: 36875088) and (ii) an hyperactivation of YAP upon UUU (Chen et al *Diabetes* 2020, PMID: 32843569; Szeto et al, *JASN* 2016, PMID: 26961347; Liang et al, *JASN* 2017, PMID: 28768710 and Wong et al, *Am J Physiol Renal Physiol* 2016, PMID: 27194720). Also, the fact that the decrease of renal fibrosis paralleled the amelioration of the autophagy flux in *Yap^{ATub}* mice after UUU is in line with the literature showing the protective role of autophagy against UUU (see the recent review by Bhatia and Choi, *Am J Physiol Renal Physiol* 2023; PMID: 37167272). Altogether, our results clearly demonstrate that the maintenance of autophagy flux in tubules lacking *Yap* has a protective role in the subsequent development of renal lesions after UUU.

Lastly, as requested by the reviewer, we also provide new blots (Figure R10, only for the reviewer) showing the increase of YAP protein levels in obstructed kidneys from mice subjected to UUU during 24 hours, in line with the literature (Gui Y et al, *J Biol Chem* 2018, PMID: 30154246; Song J et al, *Cell Reports* 2019, PMID: 31825843 and Bai Y et al, *Inter J of Biol Sc* 2020, PMID: 31929748). However, we would like to emphasize here that even though YAP total levels are increased upon UUU, we always measured the ratio of nuclear/cytoplasmic YAP, which is not impacted by YAP total levels.

Figure R10: Inhibition of urinary fluid flow induces the expression of YAP in kidney. (a) Representative images of YAP and Actin proteins levels in kidneys samples from mice subjected or not to unilateral ureteral obstruction during 24 hours (UUO), by western blot analysis. (b) The ratio of YAP to Actin was determined by densitometry, relative to panel a. Data show the mean ± s.e.m.; n = 5 Sham kidneys and n = 6 UUO kidneys, t-test.

8. Role of YAP and autophagy in AKI and CKD are disease-type or context specific (PMID: 32704047,30072422, 32843569). These findings are not discussed adequately. Moreover, authors may consider inclusion of a second renal injury model to explore relationship between YAP activation/upregulation and autophagy in CKD progression.

We thank the reviewer for raising this point. As suggested by the reviewer, we have now added a new paragraph discussing the complex and dual role of YAP and autophagy in AKI and CKD in the discussion section of the manuscript (see pages 19-20). In addition, as indicated above, we have now investigated the role of the YAP/autophagy interplay in vivo by applying the UUO model to mice in which Yap was selectively inactivated in renal tubular cells. Notably, inactivation of Yap significantly decreased the severity of renal lesions in this experimental model, further reinforcing the deleterious role of YAP in chronic kidney disease. On the other hand, as discussed below (see answer point 1 of reviewer 4), we studied the impact of the ischemia-reperfusion injury on YAP/autophagy interplay. However, we could not detect any difference between Yap^{ΔTub} and Yap^{fl/fl} littermates 2 weeks after renal ischemia. Whether the experimental time point (a period that follows the recovery phase, but proceeds the development of renal lesions) or the experimental model (a mix of acute and chronic kidney disease) account for this observation would deserve further investigations.

Reviewer #4 (Remarks to the Author):

Aurore Claude-Taupin et al show that nuclear YAP/TAZ negatively regulates autophagy machinery in kidney epithelial cells subjected to fluid flow. And reported that this crosstalk is supported by a primary cilium-dependent activation of AMPK and SIRT1, independently of the Hippo pathway. Then, they made a conclusion that dysregulation of this pathway is associated with the early onset of CKD. Although the crosstalk between autophagy and YAP/TAZ pathway has not been reported in the kidney field, in other organs and cell types, a couple of studies have demonstrated the association between these two events. Several issues should be further addressed.

1) In this study, the author used UUO model, several other models such as IRI induced CKD and DKD model should be employed to test the interaction of crosstalk between autophagy and YAP/TAZ pathway.

We would like to thank the reviewer for these insightful suggestions. As requested, we investigated the impact of ischemia-reperfusion model (unilateral ischemia during 30 minutes, followed by 2 weeks of reperfusion). As illustrated in new figure (figure R11, only for the reviewer), we could not detect any effect on YAP subcellular localization (Fig. R11a,b) and autophagy measured by counting LC3-positive structures (Fig. R11c,d). Even though few reports have shown that YAP/TAZ activity and autophagy are upregulated at early time points of reperfusion after ischemia, it is likely that at late time points these adaptive phenomena have been overcome.

Figure R11: The ischemia/reperfusion (IR) injury does not affect the renal tubular localization of YAP nor does alter the formation of LC3 puncta in kidney.

(a-d) Mice subjected to Ischemia Reperfusion (IR-30 minutes) or sham operation were euthanized at 14 days after surgery. Representative images (a) and quantification (b) of YAP nuclear levels in renal tubules. Scale bars, 10 μ m. Data show the mean \pm s.e.m.; n = 5 (sham) and n = 4 (IR) different mice. t-test. Representative images (c) and quantification (d) of LC3 puncta in renal tubules. Scale bars, 10 μ m. Data show the mean \pm s.e.m.; n = 5 (sham) and n = 4 (IR) different mice. t-test.

Moreover, we now add new data to complete the ones obtained 24h post UUO (Figure 7e,f) with mice sacrificed 14 days after surgery, when lesions develop (Figure 10 and Extended Data Figure 9), in order to study the role of Yap and autophagy in disease progression (See also point 7 for reviewer 3). In this setting, we found that Yap deletion in kidney tubular cells (using the Ksp-Cre driver) ameliorated autophagy flux 14 days after UUO, correlated with decreased renal fibrosis. Thus, our data demonstrate that the maintenance of autophagy flux in tubules lacking Yap has a protective role in the development of renal lesions after UUO.

2) In addition to zebrafish and mouse model, the association should be further examined in the CKD patient kidney biopsies.

As recommended by the reviewer, we have analyzed kidney biopsies from 5 CKD patients (diabetic nephropathy, DKD) and 5 control kidney samples. We chose DKD because this is the leading cause of CKD. As illustrated in New Figure 9a-d, we observed that the increased nuclear localization of YAP correlated with decreased levels of colocalization between autophagosomes (LC3-positive structures) and acidic compartments (LAMP1-positive structures), suggesting a blockade of autophagic flux. These results are line with previous data showing (i) a hyperactivation of YAP in a diabetic mouse model (Chen J et al, J Am Soc Nephrol 2016, PMID: 26453611) and in kidney biopsies from DKD patients (Chen J et al, Diabetes 2020, PMID: 32843569) and (ii) a blockade in autophagy flux in DKD samples (Liu WJ et al, JBC 2015, PMID:

26100632). Moreover, these results and others reported in this manuscript offer a cellular and molecular explanation to the recent observation that the expression of Rubicon is increased in the plasma of DKD patients (Watany et al, Mol Med 2022, PMID: 36476132).

3) In addition to YAP/TAZ, whether the other components are modulated by flow Intensity in tubular cells should be deciphered.

We thank the reviewer for raising this point. In addition to YAP/TAZ, we have shown a decrease of a pool of activated AMPK localized at the base of the primary cilia in cells submitted to high shear stress compared to physiological shear stress (see former Figure 7f,g, now in Figure 8f,g). To go further, we have also performed new experiments and observed that SIRT1 protein levels decrease upon pathological shear stress (versus the physiological shear condition, see Extended Data Figure 8a,b), while AMPK levels are not affected by pathological shear stress (Fig. R12, only for the reviewer). Altogether, these data show that several components are modulated by flow intensity in tubular cells.

Figure R12: AMPK protein levels are not affected by pathological shear stress.

Representative images of AMPK and Actin protein levels in KECs subjected (or not, static) to physiological shear stress for 1 day, then kept for 1 more day with physiological shear stress (Shear 1 dyn) or subjected to pathological flow (Shear 4 dyn), by western blot analysis, representative of 3 independent experiments.

4) The role for Teads in regulating autophagy should be tested in this model.

We thank the reviewer for this suggestion. To answer this point, we have repeated autophagic read out experiments using cells overexpressing a genetically encoded fluorescently-tagged inhibitor of the interaction between YAP/TAZ and TEAD transcription factors (GFP-TEADi, Yuan Y et al., Nat Commun 2020, PMID: 32193376). As illustrated in Extended Data Figure 3b,c, we observed an induction of LC3 puncta accumulation in cells overexpressing GFP-TEADi treated with chloroquine (compared to cells overexpressing GFP alone treated with chloroquine). These data show that TEAD proteins are able to inhibit autophagy in kidney epithelial cells submitted to shear.

5) In Fig 2e and 2g, the data for the colocalization of TAZ and LC3 should be represented.

We now provide data (See Figure R13, only for the reviewer) showing the localization of TAZ and LC3 in cells overexpressing the constitutively active form of YAP (YAP 5SA) or inactive form (YAP 5SA/S94A). As the reviewer can appreciate, TAZ can exit the nucleus upon shear, even in the presence of YAP 5SA or 5SA/S94A.

Figure R13: Colocalization of TAZ and LC3 in KECs expressing a constitutively active or inactive form of YAP during shear stress.

Representative images of TAZ and LC3 in KECs subjected to shear stress during 24h in the presence or absence of chloroquine (CQ). Scale bars, 10 μ m.

REVIEWER COMMENTS

Reviewer #1 (Remarks to the Author):

This is a revised version of the manuscript by Claude-Taupin et al. The authors postulate that PC senses shear stress in KEC resulting AMPK activation and YAP phosphorylation thus promoting its cytoplasmic retention/nuclear exit. Inactivation of YAP/TAZ leads to increased in autophagy flux by inhibiting the expression of autophagy repressors (i.e. rubicon) mediated by Sirt1. In this revised version, authors have made an effort in addressing concerns arose during prior review and incorporated novel data. However, a number of concerns remain unresolved and contradicting findings, some of which null the hypothesis were found, as detailed below, which considerably lower enthusiasm.

- Follow up R1, Comment 1- Still lower magnification pictures have not been provided in many key Figures (i.e. Figure 3c, 3e, ext Fig 3a,b)
- Extended Figure 1: Follow up R1, Comment 2: Decreased nuclear YAP/TAZ and increased p-YAP are seen in static D1, without the need of shear stress. This needs to be explain. YAP/TAZ activity in Static D0, Static D1/D2, Shear D1/D2 needs to be evaluated to support the claim that “the decrease in YAP/TAZ activity was more important when KECs were subjected to fluid flow than under static conditions (lane 128)”.
- Follow up R1, Comment 3: Still many figures show no increase in LC3-II or puncta with CQ- Why CQ does not induce puncta (Fig 1 b, d) under static conditions in siTAZ and siYAP transfected cells? (Fig 1 b, d) or LC3-II in Ext Figure 2a, b, c, d? That should be expected.
- Follow up R1, Comment 3: Extend Fig 2a does not show increased LC3-II with shear stress (!!).
- In some, but not all, YAP/TAZ silencing increases basal LC3 levels (fig 2e, f, g, h, ext Fig 2 e, g). This somehow contradicts data in Figure 2i and the claim on Lane 158 that YAP and TAZ individually play a role in inhibiting autophagic flux in cells subjected to shear stress. Shear stress might not be needed to observe that effect. This must be clarified since it invalidates the hypothesis.
- Follow up R1, Comment 5: Lane 160-161- The concept of high density needs to be defined. Same in Extended Figure 1. D0, D1 and D2 are not explained in the text. Which is the relevance for the study?

- Figure legend for Fig 2k and l should indicate that these are cell lines with CRISPR/Cas9 inactivation of YAP.

- Follow up R1, Comment 9: Figure 2 and C seems to be at different magnification. Authors should indicate what is considered or identified as puncta. For instance, as small spots in DMSO-Static siTAZ considered puncta? What does the “spot detector” tool select?

- Figure 3, Follow up R1, Comment 9 & 10: How were cell lines validate? YAP/TAZ inhibition or activation must be confirmed. There are many problems with this figure. First, no accumulation of LC3-II with CQ is seen without shear stress. Second, no further increase of LC3-II with CQ and shear stress is seen. Most importantly, no increase in LC3-II with shear stress is observed in the cells with constitutively inactive YAP. This completely contradicts the hypothesis/claims that inactivation of YAP/TAZ is required to activate autophagy flux in response to shear stress (!).

- Figure 3,d and extended figure 3c puncta graphs and statistics are plotted differently than in prior figures (data points versus average). The authors must detail how counting and statistical analysis are conducted. How many cells or fields per replicate? How areas are selected. This is crucial to evaluate rigor of the data, especially in those experiments in which results were not confirmed using biochemical approaches. This needs to be revise throughout the manuscript.

- Ext Fig 3 is not convincing. Hardly any puncta is observed. Flux needs to be quantified by WB. No experiments have been conducted to claim that YAP/TAZ regulate autophagy in a manner dependent on their co-transcriptional activity (Lane 189).

- Follow up R1, Comment 7 & 13: The claim on Lane 201 – “several autophagy inhibitors” still needs to be soften up. Just one has been shown. To claim statement on Lane 203, author must show that downregulation of rubicon activates autophagy and confirm downregulated protein levels of rubicon in stress cells and validated 5SA and 5SA/S94A cells (still not shown, Follow up R1, Comment 12. Also, on Lane 156 authors claim that inactivation of YAP/TAZ do not affect autophagosome formation, thus concluding that the effect is mediated through increased autophagy flux. However, results here suggest activation of autophagy. This needs to be clarified.

- Lane 212- nuclear exit has not been experimentally confirmed. Earlier in the manuscript, cytoplasmic retention was presumed as the culprit for increased cytoplasmic YAP TAZ levels. Authors need to clarify or re-word.

- Fig 4e comparison between shear stress does not look significant. Authors need to clarify “mean \pm s.e.m.; n = 9 from 3 independent experiments, t-test”. Data with siCEP164 is more convincing.
- Figure 5c d, Lane 239-242 Expression of YAP S61A and validation must be shown. Data in Fig 6,b not convincing. No increased in LC3-II with shear stress is observed.
- Follow up R1, Comment 19 - Ext Fig 6: No decrease in P-YAP is observed in static cells treated with Dorso. Why? That should be expected.
- Figure 6g: Follow up R1, Comment 17 -Why expression of Atg5 was silenced? This needs to be stated in the manuscript. Also, the effect of Dorso treatment in autophagy should be evaluated.
- Lanes 251-253: Cytoplasmic retention and autophagy was not tested in figures 5i-m. Conclusions are not supported by data.
- Follow up R1, Comment 20 - Extend Fig 7c, the significant increase in transactivation activity in DMSO vs EX527 in shear is still not convincing. Statistical details are needed.
- Lanes 272, the claim that EX527 inhibits autophagy flux needs to be supported by quantitative data (WB).
- Fig 6k. SIRT1-NLS mut expression and sirt localization must be validated.
- Lanes 305-306 & 314-315 - Causation has not been proven. Autophagy was not evaluated in figure e.
- Lane 335-336: WB are necessary to claim autophagy flux inhibition here as 4 dyn shear stress has not been applied in earlier experiments.
- Figure 9a: Channels need to be split. General increase in YAP levels is observed in DKD. Also, no correlative analysis between nuclear yap and LC3-LAMP have been performed to support claim on Lane 350. Similarly, fusion of autophagosomes with lysosomes has not been investigated. Conclusions and title of figure legend 9 is not supported by the data.

- Figure 10 (g-i): Pictures show a decrease in LC3 staining in UUO. Reduced puncta size in YAP ko mice is not convincing. Statement in text re- autophagosomes accumulation is not supported by data. No lysosomal marker has been evaluated. P62 levels by itself cannot be used for flux studies. WB quantifications are needed.

- Ext Fig 10- Many aspects of that hypothesis need to be yet tested.

Reviewer #2 (Remarks to the Author):

All my concerns have been addressed by the authors in this revised MS. I fully support the publication of this MS.

Reviewer #3 (Remarks to the Author):

In the revised manuscript, the authors have performed several additional studies to address some of the concerns of this reviewer. Few major concerns, however, still remain unanswered.

- It is quite puzzling that renal YAP involvement on autophagy is not consistent between UUO and IR-induced renal injury according to new data. It is evident from previous studies that YAP upregulation plays a protective role in the context of acute kidney injury (PMID 6115662) while YAP induction plays a causative role in the context of UUO-driven fibrosis. These observations raise concerns regarding broad in vivo relevance of YAP involvement in renal autophagy during CKD or YAP involvement in autophagy is dependent on kidney disease etiology. Although authors provide extensive human studies linking YAP upregulation to autophagy in diabetic kidney disease, pertinent animal studies in diabetic nephropathy (most dominant cause of CKD) are still lacking which could clarify above confusion.

- IF data in panels a and c of Figure 2 showed the cells with nuclei of different sizes. Are these images taken at same magnification? Autophagic flux determination is mostly based on the punctated expression of LC3 protein. The quantification of puncta is not shown for these figures.

- YAP involvement in shear stress is well studied in the context of various diseases, but these studies are not cited. For example.

1: Lee, H., Diaz, M., Price, K. et al. Fluid shear stress activates YAP1 to promote cancer cell motility. *Nat Commun* 8, 14122 (2017). <https://doi.org/10.1038/ncomms14122>

2. Lee, H. J., Ewere, A., Diaz, M. F., & Wenzel, P. L. (2018). TAZ responds to fluid shear stress to regulate the cell cycle.

3. Yu, H., He, J., Su, G., Wang, Y., Fang, F., Yang, W., Gu, K., Fu, N., Wang, Y., Shen, Y., & Liu, X. (2021). Fluid shear stress activates YAP to promote epithelial-mesenchymal transition in hepatocellular carcinoma. *Molecular oncology*, 15(11), 3164–3183. <https://doi.org/10.1002/1878-0261.13061>

4. Kim KM, Choi YJ, Hwang J-H, Kim AR, Cho HJ, Hwang ES, et al. (2014) Shear Stress Induced by an Interstitial Level of Slow Flow Increases the Osteogenic Differentiation of Mesenchymal Stem Cells through TAZ Activation. *PLoS ONE* 9(3): e92427. <https://doi.org/10.1371/journal.pone.0092427>

5. Wang KC, Yeh YT, Nguyen P, Limqueco E, Lopez J, Thorossian S, Guan KL, Li YJ, Chien S. Flow-dependent YAP/TAZ activities regulate endothelial phenotypes and atherosclerosis. *Proc Natl Acad Sci U S A*. 2016 Oct 11;113(41):11525-11530. doi: 10.1073/pnas.1613121113.

Reviewer #4 (Remarks to the Author):

The author have fully addressed my concerns.

REVIEWER COMMENTS

Reviewer #1 (Remarks to the Author):

This is a revised version of the manuscript by Claude-Taupin et al. The authors postulate that PC senses shear stress in KEC resulting AMPK activation and YAP phosphorylation thus promoting its cytoplasmic retention/nuclear exit. Inactivation of YAP/TAZ leads to increased in autophagy flux by inhibiting the expression of autophagy repressors (i.e. rubicon) mediated by Sirt1. In this revised version, authors have made an effort in addressing concerns arose during prior review and incorporated novel data. However, a number of concerns remain unresolved and contradicting findings, some of which null the hypothesis were found, as detailed below, which considerably lower enthusiasm.

- Follow up R1, Comment 1- Still lower magnification pictures have not been provided in many key Figures (i.e. Figure 3c, 3e, ext Fig 3a,b)

We now provide lower magnification pictures (Figure 3c, 3e, ext Fig 3a,b). The reviewer and readers should now better appreciate the changes in cells submitted to shear.

- Extended Figure 1: Follow up R1, Comment 2: Decreased nuclear YAP/TAZ and increased p-YAP are seen in static D1, without the need of shear stress. This needs to be explain.

The reviewer is right. Decreased nuclear YAP/TAZ and increased p (S397)YAP are seen in static D1 (versus static D0) in the absence of shear stress. Sixty percent of YAP proteins are indeed nuclear in static D0 while only 40% are nuclear in static D1 (Extended Data Figure 1a,b). Here, it is important to mention that this effect is consistent with the literature (Pavel et al., Nat Commun 2018, PMID: 30054475) showing a Hippo-dependent nuclear exclusion of YAP in cells cultivated at high confluency.

However, shear stress significantly stimulates YAP nuclear exclusion (from 40% in static D1 to 25% in shear D1), as illustrated in Figure 1a, b) and this stimulation is Hippo independent (as shown in Extended Data Figure 5), thus explaining why the levels of phosphorylated YAP (S397) are not further increased by shear stress (Extended Data Fig.1e).

- YAP/TAZ activity in Static D0, Static D1/D2, Shear D1/D2 needs to be evaluated to support the claim that “the decrease in YAP/TAZ activity was more important when KECs were subjected to fluid flow than under static conditions (lane 128)”.

First, it is worth mentioning that YAP/TAZ activity (using luciferase assay) has already been evaluated in static D1 versus shear D1 (Figure 1) and in static D2 versus shear D2 (Figure 8). Unfortunately, the luciferase assay is not technically possible on static cells (D0 versus D1) (or static D0 versus shear D1) as we have to use fresh cell lysates for this assay and here, we are not harvesting cells the same day. However, we have checked the YAP/TAZ nuclear translocation for static D0, static D1/D2, Shear D1/D2 (see Extended Data Figure 1).

- Follow up R1, Comment 3: Still many figures show no increase in LC3-II or puncta with CQ- Why CQ does not induce puncta (Fig 1 b, d) under static conditions in siTAZ and siYAP transfected cells? (Fig 1 b, d) or LC3-II in Ext Figure 2a, b, c, d? That should be expected.

First, we assume here that the reviewer is referring to the former Figure 2b,d (and not figure 1b,d). The reviewer is right, there is no increase of LC3 puncta (Figure 2b,d) or LC3-II (Extended Data Figure 2a-d) under static conditions in cells knocked down for YAP or TAZ in the presence of CQ. These data are in agreement with the literature (Pavel et al., Nat Commun 2018, PMID: 30054475 and Totaro et al., PNAS 2019, PMID:31416916) showing a positive effect of YAP/TAZ on autophagy in cells cultivated without mechanical stress. It should also be mentioned here that in static conditions, cells are cultivated at high confluency and thus decreased autophagy flux is due to hippo-dependent YAP nuclear exclusion (as shown in Pavel et al., Nat Commun 2018, PMID: 30054475). However, YAP/TAZ play an opposite role on autophagy in cells submitted to shear stress (i.e. an inhibitory effect on the autophagic flux). Thus YAP/TAZ have a positive or a negative role on autophagy depending on the conditions (i.e. static vs shear stress). This point was discussed in the previous rebuttal letter in Reviewer 1's point 8.

- Follow up R1, Comment 3: Extend Fig 2a does not show increased LC3-II with shear stress (!!).

This is mainly due to the high impact of YAP KD on LC3-II levels during shear stress. Indeed, to avoid saturated bands, we had to use a shorter exposure times. We now provide a higher exposed LC3 blot in Extended Data Fig. 2a. We believe that the reviewer and readers should now better appreciate the increase of LC3-II in cells submitted to shear and treated with chloroquine (compared to static cells treated with chloroquine).

- In some, but not all, YAP/TAZ silencing increases basal LC3 levels (fig 2e, f, g, h, ext Fig 2 e, g). This somehow contradicts data in Figure 2i and the claim on Lane 158 that YAP and TAZ individually play a role in inhibiting autophagic flux in cells subjected to shear stress. Shear stress might not be needed to observe that effect. This must be clarified since it invalidates the hypothesis.

We respectfully disagree with the reviewer's analysis of LC3-II levels as "basal levels" in the absence of chloroquine. Indeed, LC3-II levels in cells knocked down for YAP/TAZ cannot lead to any conclusion without taking into account the LC3-II levels in cells treated with chloroquine (CQ). To accurately analyze autophagy activity, it is essential to evaluate the dynamics of autophagy, as the quantity of LC3-II in cells results from autophagosomes that have been formed in the cells, which have not yet been degraded after their fusion with a lysosome. This is called "autophagic flux" (see Mizushima et al. 2010 Cell and many other methodological reviews, including "Guidelines for the use and interpretation of assays for monitoring autophagy" published in Autophagy). To analyze autophagic flux, it is important to compare the difference of LC3-II levels in CQ-treated and untreated cells knocked down for YAP/TAZ with the difference of CQ treated and untreated cells in scramble condition. Thus, an increase in LC3-II levels in the absence of chloroquine does not necessarily mean an increase of autophagosome formation as this could reflect an inhibition of autophagosome fusion with lysosomes. As we always observed increased accumulation of LC3-II in the presence of chloroquine in siYAP/TAZ samples compared to Scr

cells, we concluded that autophagy flux is increased after YAP/TAZ KD/KO. In conclusion, the increase of LC3-II levels observed in some WB in the absence of chloroquine do not invalidate our findings.

- Follow up R1, Comment 5: Lane 160-161- The concept of high density needs to be defined.

This point was discussed in the previous rebuttal letter (point 5 Reviewer 1). Cells used in this study are all cultivated at high density. As illustrated in Figure R3 (previous rebuttal letter, only for the reviewer), we could not observe any difference between static D1 and shear D1: cells are 100% confluent. We have already better defined and described the concept of high density and the protocol in the methods. We propose now to completely remove the term “high density” in our manuscript if the reviewer still thinks that it could create a confusion for readers.

- Same in Extended Figure 1. D0, D1 and D2 are not explained in the text. Which is the relevance for the study?

In this study, we have used 3 different time points for static conditions (D0, D1, D2). For experiments using KECs submitted to 24 hours (D1) of physiological shear stress, we have used static D1 as control. Static D0 was only shown (in Extended Data Figure 1a-d) to show the effect of static D1 (versus static D0) on YAP/TAZ nuclear exclusion. For experiments using KECs submitted to 48 hours (D2) of shear stress (either 48 hours of physiological or 24 hours of physiological followed by another additional 24 hours of pathological shear stress), we have used static D2 as control (Figure 8a,b; Extended Data Figure 8c,d). Regarding experiments using HK-2, we have always used static D2 as control since HK-2 were always submitted to 48h of shear (either 48 hours of physiological one or 24 hours of physiological followed by another additional 24 hours of pathological shear stress) (Extended Data Figure 1g-j and 8e,f). We have now reworded the Methods to better explain the relevance of these different controls.

- Figure legend for Fig 2k and l should indicate that these are cell lines with CRISPR/Cas9 inactivation of YAP.

We have now modified the legend for Fig. 2k and l.

- Follow up R1, Comment 9: Figure 2 and C seems to be at different magnification.

We have now adjusted this (see Figure 2a and c).

- Authors should indicate what is considered or identified as puncta. For instance, as small spots in DMSO-Static siTAZ considered puncta? What does the “spot detector” tool select?

We now provide a screen shot (Figure R1, only for the reviewer) showing how the “spot detector” tool select the puncta. For all experiments, we have only selected the scale 2 corresponding to spots of about 4 to 7 pixels diameter. For more details, see the following link <https://icy.bioimageanalysis.org/plugin/spot-detector>. We have now reworded the Methods to better explain the quantification of puncta using the Plugin spot detector.

Figure R1: LC3 puncta quantification using the spot detector plugin in Icy. In picture A, the original image before using the spot detector plugin. After setting up the plugin to “Detect bright spot over dark background” in the red channel, “Scale 2” was selected to detect spots of around 3 pixels, using a sensitivity of 70. In Image B, a screenshot of all the spots detected by the plugin. Their number can be saved by the plugin in a new excel file. If needed, some ROIs can be drawn before launching the plugin, in order to analyze the spots in specific regions/cells.

- Figure 3, Follow up R1, Comment 9 & 10: How were cell lines validated?

We have now better described the validation of our cell lines in the methods section (see page 71). The expression of GFP-YAP 5SA and 5SA/S94A are shown in Figure 3a by western blot using GFP antibodies. Both mutants migrate at around 100 kDa, as expected and according to the molecular weight of GFP (27 kDa). We also observed the expression of GFP-YAP 5SA and GFP-YAP 5SA/S94A by immunofluorescence (Figure 3c).

- YAP/TAZ inhibition or activation must be confirmed.

We now provide new figures (Figure R2, only for the reviewer) showing increased YAP/TAZ activity (measured by luciferase assay) in cells expressing GFP-YAP 5SA (compared to cells expressing 5SA/S94A).

Figure R2: YAP/TAZ activity is increased in KECs expressing a constitutively active (5SA) form of YAP. Luciferase assay for YAP/TAZ activity in KECs expressing a constitutively active (5SA) or inactive (5SA/S94A) form of GFP-YAP. Data show the mean \pm s.e.m.; $n = 6$ individual data points from 3 independent experiments (duplicate), t -test.

- There are many problems with this figure. First, no accumulation of LC3-II with CQ is seen without shear stress.

We now provide a higher exposed LC3 blot (Fig. 3a). We believe that the reviewer and readers should now better appreciate the increase of LC3-II in cells expressing GFP-YAP 5SA and treated with chloroquine (CQ) (compared to cells without CQ in static condition). It is important to mention that no accumulation of LC3-II is observed with CQ in cells expressing GFP-YAP 5SA/S94A since GFP-YAP 5SA/S94A inhibits autophagic flux in static conditions in agreement with the literature (Pavel et al., Nat Commun 2018, PMID: 30054475 and Totaro et al., PNAS 2019, PMID: 31416916).

- Second, no further increase of LC3-II with CQ and shear stress is seen.

We assume here that the reviewer is referring to the effect of CQ and shear stress on cells expressing GFP-YAP 5SA. Indeed, we did not observe any effect, which was expected and validate our main hypothesis. When we genetically force YAP to localize in the nucleus (using cells expressing GFP-YAP 5SA) in shear stress condition, we induce the expression of Rubicon, which inhibits autophagic flux.

- Most importantly, no increase in LC3-II with shear stress is observed in the cells with constitutively inactive YAP. This completely contradicts the hypothesis/claims that inactivation of YAP/TAZ is required to activate autophagy flux in response to shear stress (!).

We assume here that the reviewer is referring to the effect of shear stress in cells expressing GFP-YAP 5SA-SS94A in the condition without CQ (compared to static) (Figure 3a). As explained earlier, one must take into account the LC3-II levels in cells treated with chloroquine (CQ). To accurately analyze autophagy activity, it is essential to evaluate the dynamics of autophagy, which is called “autophagic flux”, by analyzing the difference of LC3-II levels in the presence and absence of CQ.

- Figure 3,d and extended figure 3c puncta graphs and statistics are plotted differently than in prior figures (data points versus average).

In all graphs, individual data points were shown with average. For graphs showing a WB quantification, columns were shown (with average and with individual data points) for clarity. We have now harmonized our graphs by removing the columns in graphs showing WB quantification.

- The authors must detail how counting and statistical analysis are conducted. How many cells or fields per replicate? How areas are selected. This is crucial to evaluate rigor of the data, especially in those experiments in which results were not confirmed using biochemical approaches. This needs to be revise throughout the manuscript.

We are surprised by these new concerns raised by the referee since he/she did not raise these points during the first round of reviewing despite his/her very careful analysis of the manuscript. We have now better described the protocol of quantification in the figure legends and in the Methods. For each figure showing quantification, we analyzed at least 30 cells in 10 different (selected randomly) areas. Each individual experiment was then independently repeated and analyzed at least three times.

For Statistical analysis, as mentioned in the methods, they were carried out using Prism 7.0a (GraphPad). P values were calculated using the unpaired t-test or analysis of variance, as mentioned in figure legends. All values are given as mean \pm SEM. For animal studies, no statistical method was used to predetermine sample size. The experiments were not randomized.

- Ext Fig 3 is not convincing. Hardly any puncta is observed. Flux needs to be quantified by WB.

We now provide new pictures for the Extended Data Fig. 3b. We believe that the reviewer and readers should now better appreciate the LC3 puncta. We were not able to perform an autophagy flux assay by WB for this experiment since the expression of GFP-TEADi was transient. This did not allow us to obtain sufficient amounts of transfected cells to observe the effects of TEADi expression on LC3-II levels in the presence and absence of CQ by WB.

- No experiments have been conducted to claim that YAP/TAZ regulate autophagy in a manner dependent on their co-transcriptional activity (Lane 189).

*We respectfully disagree with the reviewer as our sentence in lines 188-189 did not directly draw this conclusion, as it stated “GFP-TEADi-transfected cells exhibited a high autophagic flux (Extended Data Fig. 3b,c), **suggesting** that YAP/TAZ regulate autophagy in a manner dependent on their co-transcriptional activity”. However, after this statement, we confirmed by chromatin immunoprecipitation (ChIP) that the CA form of YAP is able to bind the Rubicon gene (Fig. 3j) and that the expression of a constitutively inactive mutant of YAP was sufficient to decrease Rubicon mRNA levels (Fig. 3h). Thus, these data support our claim on page 8 and is also in agreement with the literature showing a YAP (or TEAD2) ChIP-seq peak in the mouse Rubcn gene*

(Tremblay et al., *Cancer Cell* 2014, PMID: 25087979 and Diepenbruck et al., *J Cell Sci* 2014, PMID: 24554433), available online in the cistrome database (<http://cistrome.org/>).

- Follow up R1, Comment 7 & 13: The claim on Lane 201 – “several autophagy inhibitors” still needs to be softened up. Just one has been shown.

We agree with the reviewer. We have therefore changed the text (see page 9).

- To claim statement on Lane 203, author must show that downregulation of rubicon activates autophagy and confirm downregulated protein levels of rubicon in stress cells and validated 5SA and 5SA/S94A cells (still not shown,

We now provide LC3 pictures showing an induction of the autophagic flux in cells expressing the 5SA-YAP and knocked down for Rubicon (Figure R3, only for the reviewer) compared to scramble cells expressing the 5SA. These data validate the 5SA genetic tool and our main hypothesis.

Figure R3: The loss of Rubicon stimulates the autophagic flux in KECs cells using a constitutively active (5SA) form of YAP. **(a)** Confirmation by real-time RT-qPCR of Rubicon knockdown in KEC cells stably expressing a constitutively active (5SA) form of YAP. mRNA levels were quantified by real-time RT-qPCR, normalized to β -actin and are presented as fold increases. Data show the mean \pm s.e.m.; $n = 3$ independent experiments, t -test. **(b,c)** Representative images **(b)** and quantification **(c)** of LC3 puncta in KECs stably expressing a constitutively active (5SA) form of YAP after transfection with a control siRNA (si Scr) or siRNA against Rubicon. Cells were then subjected to flow during 24h in the presence or absence of chloroquine (CQ). Data show the mean \pm s.e.m.; $n = 6$ from 3 independent experiments, t -test. Scale bars, 10 μ m. #: number.

- Follow up R1, Comment 12. Also, on Lane 156 authors claim that inactivation of YAP/TAZ do not affect autophagosome formation, thus concluding that the effect is mediated through

increased autophagy flux. However, results here suggest activation of autophagy. This needs to be clarified.

*We respectfully disagree with the reviewer. As mentioned in the previous rebuttal letter (see point 7), we have provided new data showing that inactivation of YAP/TAZ stimulates autophagic flux without affecting autophagosome formation. **First**, we used the tandem RFP-GFP-LC3 probe to quantify the number of autophagosomes (GFP+RFP+) and autolysosomes (GFP-RFP+) in cells submitted to shear. As illustrated in Figure 2i,j, we observed an increase in autolysosome numbers after the knockdown of YAP (or TAZ) compared to control cells (Scr: scramble). However, the loss of YAP (or TAZ) did not impair the number of autophagosomes in cells subjected to shear, highly suggesting that YAP and TAZ negatively regulate autophagic flux (fusion of autophagosomes with lysosomes) in cells submitted to fluid flow. **Second**, to make sure that the loss of YAP/TAZ only plays a role in stimulating autophagic flux, we have quantified the colocalization between two early markers of the autophagy pathway (ATG16L1 and WIPI2) in YAP (or TAZ) deficient cells (new Extended Data Fig. 2i,j). We did not observe any difference in the number of ATG16L1+WIPI2+ puncta in cells knocked down for YAP (or TAZ) compared to the scramble condition. Altogether, we conclude that YAP (and TAZ) only play a role in the fusion between autophagosomes and lysosomes in cells subjected to shear. This result is in line with the LC3-Western blot data shown in Figure 2e-h.*

- Lane 212- nuclear exit has not been experimentally confirmed. Earlier in the manuscript, cytoplasmic retention was presumed as the culprit for increased cytoplasmic YAP TAZ levels. Authors need to clarify or re-word.

We agree with the reviewer. We have therefore changed the text (see page 9).

- Fig 4e comparison between shear stress does not look significant. Authors need to clarify “mean \pm s.e.m.; n = 9 from 3 independent experiments, t-test”. Data with siCEP164 is more convincing.

The expression “does not look significant” used by the referee is not coherent in front of a statistical analysis. Results obtained with siCEP164 are more striking because the KD of CEP164 ablates the primary cilium whereas siKIF3a only shortens the primary cilium in proximal tubule derived epithelial cells (our results in Extended Data Figure 4c and see (Qiu et al, J Cell Sci 2012 PMID:22357948).

- Figure 5c d, Lane 239-242 Expression of YAP S61A and validation must be shown.

The expression of GFP-YAP S61A (point mutation confirmed by sequencing as mentioned in the methods section) and validation are shown in Extended Data Fig. 6a. Indeed, we have checked the expression of GFP-YAP S61A by western blot using GFP antibodies. We also observed the expression of GFP-YAP WT and GFP-YAP S61A by immunofluorescence (Figure 5c).

- Data in Fig 6,b not convincing. No increased in LC3-II with shear stress is observed.

We assume here that the reviewer is referring to the former Extended Data Figure 6b (and not figure 6b). We agree with the reviewer that shear stress does not increase LC3-II in DMSO-treated

cells. However, as explained previously, this finding does not mean anything without taking in account the LC3 levels in cells treated with chloroquine (CQ). Here, we observe a huge increase in cells submitted to shear and treated with CQ compared to CQ-treated static cells.

- Follow up R1, Comment 19 - Ext Fig 6: No decrease in P-YAP is observed in static cells treated with Dorso. Why? That should be expected.

First, it is important to mention that not only AMPK but also LATS is able to phosphorylate YAP at S61. We therefore think that LATS (and not AMPK) is responsible for YAP phosphorylation (at S61) in static conditions. In shear conditions, AMPK, which is known to be activated (Boehlke C et al., Nat Cell Biol 2010; PMID: 20972424 and our previous works Orhon et al 2016 NCB, Miceli et al 2020 NCB) is the main protein responsible for YAP S61 phosphorylation, since we showed that YAP inactivation is independent on the canonical Hippo pathway (Extended Data Fig. 5h-j).

- Figure 6g: Follow up R1, Comment 17 -Why expression of Atg5 was silenced? This needs to be stated in the manuscript.

We assume here that the reviewer is referring to figure 5g. We performed loss of function experiments for Atg5 because the reviewer asked us to monitor lipophagy. Indeed, one way to monitor lipophagy is to knock down ATG genes and then quantify the number of lipid droplets. We opted to use siATG5 since we have previously shown that Atg5 silencing inhibits lipid droplet degradation in cells submitted to shear (Miceli et al, Nat Cell Biol 2020; PMID:32868900). We have now modified the text (see page 11) as requested.

- Also, the effect of Dorso treatment in autophagy should be evaluated.

We are surprised by this new concern raised by the referee since he/she did not raise this point during the first round of reviewing. However, we now provide LC3 pictures showing an inhibition of the autophagic pathway in KECs treated with dorsomorphin (compared to the control) in shear conditions (Figure R4, only for the reviewer).

Figure R4: The dorsomorphin treatment inhibits the autophagic pathway in KECs subjected to shear. (a,b) Representative images (a) and quantification (b) of LC3 puncta in KECs treated or not with dorsomorphin (Dorso.) and subjected to flow (shear) during 24h in the presence or absence of chloroquine (CQ). Data show the mean \pm s.e.m.; $n = 30$ images analyzed from 3 independent experiments, t -test. Scale bars, $10 \mu\text{m}$. #: number.

- Lanes 251-253: Cytoplasmic retention and autophagy was not tested in figures 5i-m. Conclusions are not supported by data.

We now provide new immunofluorescence experiments (Extended Data Figure 6e and 6f, new experiment) showing the nuclear exclusion of YAP in cells knocked down for LKB1. Altogether, these new data confirm our previous findings. Regarding the effect of LKB1 on autophagy, we have previously shown (Orhon et al, Nat Cell Biol, 2016; PMID: 27214279) that LKB1 is important to induce an AMPK-dependent autophagy in cells submitted to shear stress.

- Follow up R1, Comment 20 - Extend Fig 7c, the significant increase in transactivation activity in DMSO vs EX527 in shear is still not convincing. Statistical details are needed.

It is worth mentioning that we cannot change the data (Extended Data Fig. 7c) we obtained (in the first version of the manuscript); however to support the “unconvincing (**but statistically significant**) data”, we provided new data in the first round of revision (see point 20), using cells expressing SIRT1 mutants (Figure 6k). We propose to remove these data (Extended Data Fig. 7c) if needed. Again, as mentioned in the discussion (and in the first rebuttal), we believe that other proteins, such as XPO1, already known to regulate the nucleocytoplasmic shuttling of YAP (Martin AP et al. EMBO Rep 2019; PMID: 31544310), could participate (together with SIRT1) in the induction of YAP/TAZ nuclear exclusion.

- Lanes 272, the claim that EX527 inhibits autophagy flux needs to be supported by quantitative data (WB).

We are again surprised by this new concern raised by the referee since he/she did not raise this point during the first round of reviewing despite his/her very careful analysis of the manuscript. We now provide LC3 pictures (Figure R5, only for the reviewer) showing an inhibition of the autophagic flux in KECs treated with EX527 compared to the control in shear condition.

Figure R5: EX527 inhibits the autophagic pathway in KECs subjected to shear. (a,b) Representative images (a) and quantification (b) of LC3 puncta in KECs treated or not with EX527 and subjected to flow (shear) during 24h in the presence or absence of chloroquine (CQ). Data show the mean \pm s.e.m.; $n = 30$ images analyzed from 3 independent experiments, t -test. Scale bars, $10 \mu\text{m}$. #: number.

- Fig 6k. SIRT1-NLS mut expression and sirt localization must be validated.

We now provide new pictures (Extended Data Figure 7d) showing that the expression and the localization of WT SIRT1 and mutants forms of SIRT1 (NLS^{mut} and NES^{mut}) in KECs. This new data, together with data from the literature ((Tanno et al, J Biol Chem, 2007 PMID 17197703), validate these genetic tools.

- Lanes 305-306 & 314-315 - Causation has not been proven. Autophagy was not evaluated in figure e.

Autophagy was indeed not evaluated in Figure 7e because we have previously reported that interruption of urinary flow in the kidney of mice by unilateral ureteral obstruction (UUO-24 hours) resulted in the downregulation of autophagy (Orhon et al, Nat Cell Biol, 2016; PMID: 27214279). To demonstrate the causal in vivo relationship between YAP and autophagy, we have generated kidney conditional knockout mice for YAP (Figure 10 and Extended Data Figure 9) that we have submitted to UUO (14 days) in order to investigate the role of YAP in renal autophagy and disease progression (see point 7 for reviewer 3 in the previous rebuttal letter). We indeed found that Yap deletion in kidney tubular cells stimulates autophagic flux. We also demonstrated that the autophagic flux is maintained in tubules lacking Yap after unilateral ureteral obstruction (UUO) and this maintenance was shown to have a protective role in the development of renal lesions after UUO.

- Lane 335-336: WB are necessary to claim autophagy flux inhibition here as 4 dyn shear stress has not been applied in earlier experiments.

Again, we are surprised by this comment as this point was not raised during the first round of reviewing. We now provide LC3 WB (Figure 8f and 8g, new experiment) showing an inhibition of the autophagic flux in KECs submitted to pathological shear stress (4 dyn/cm²) compared to physiological one (1 dyn/cm²).

- Figure 9a: Channels need to be split.

We now provide new Figure 9a with splitted channels.

- General increase in YAP levels is observed in DKD.

We indeed observed that YAP levels increase in DKD. However, we always measured here the ratio of nuclear/cytoplasmic YAP, which is not impacted by YAP total levels. This point was discussed in the point 7 for Reviewer 3 in the previous rebuttal letter.

- Also, no correlative analysis between nuclear yap and LC3-LAMP have been performed to support claim on Lane 350. Similarly, fusion of autophagosomes with lysosomes has not been investigated. Conclusions and tittle of figure legend 9 is not supported by the data.

We respectfully disagree with the reviewer. It is worth mentioning that it is technically impossible to perform triple (LC3/LAMP1/YAP) labeling experiments on the same human kidney section since, to our knowledge, there is no working antibody for immunofluorescence and commercially available against either LC3 or LAMP1 or YAP, produced in animals other than rabbit and mouse. Here, we have performed two different immunofluorescence experiments (first one using YAP rabbit antibody and second one using mouse LC3 and rabbit LAMP1 antibodies) on two different (but serial) tissue sections, which belong to the same kidney from the same DKD patient. Here we also investigated the fusion of autophagosomes with lysosomes by analyzing the colocalization between LC3-positive structures with acidic compartments (LAMP1-positive structures) and observed a decrease of the fusion process in DKD patients.

- Figure 10 (g-i): Pictures show a decrease in LC3 staining in UUO.

First, we assume here that the reviewer is talking about the overall cytosolic LC3 (LC3-I and not LC3-II) staining. We think that the decrease of LC3-I staining in UUO condition could be mainly due to the increase of LC3-II puncta formation. We now modified Fig.10g to avoid any confusion.

- Reduced puncta size in YAP ko mice is not convincing. Statement in text re- autophagosomes accumulation is not supported by data.

We now provide new pictures (Figure 10g) to better support the quantification shown on Figure 10i). We believe that the reviewer and readers will now better appreciate the changes in YAP KO mice.

- No lysosomal marker has been evaluated.

Unfortunately, our lysosomal antibody (used for immunofluorescence on human kidney sections Figure 9) did not work on mouse kidney sections, that's why we have decided to perform p62 immunofluorescence.

- P62 levels by itself cannot be used for flux studies. WB quantifications are needed.

Fluorescence analysis and quantification of SQSTM1/p62 puncta is a well-accepted method for the detection of autophagic flux in tissues samples (see "Guidelines for the use and interpretation of assays for monitoring autophagy." 2021 PMID:33634751 published in Autophagy and other many methodological reviews published on this topic.

- Ext Fig 10- Many aspects of that hypothesis need to be yet tested.

We respectfully disagree with the reviewer. Here, it is important to mention that the overall schematic summary not only refers to data obtained in this manuscript but also findings from previous publications from us and others. We have previously shown that physiological fluid flow induces mitochondrial biogenesis and lipophagy in a primary cilia-dependent manner in kidney epithelial cells (Miceli et al, Nat Cell Biol 2020; PMID:32868900). In the same paper, we have also demonstrated that two divergent signaling events emanate downstream of AMPK to control mitochondrial biogenesis and lipophagy, respectively. The first one was shown to be dependent on PGC1 α (downstream of the PC-AMPK axis). The lipophagic pathway and mitochondrial biogenesis are important to direct metabolic adaptation. This schematic summary also refers to data obtained from two other publications (Orhon et al, Nat Cell Biol, 2016; PMID: 27214279) and (Boehlke C et al, Nat Cell Biol 2010; PMID:20972424) showing that primary cilia induce autophagy in an LKB1/AMPK-dependent pathway in kidney epithelial cells submitted to physiological shear stress. Regarding the schematic summary of the pathological condition, we have used dashed lines where links (for example the link between AMPK and PGC1 α) remain hypothetical.

Reviewer #2 (Remarks to the Author):

All my concerns have been addressed by the authors in this revised MS. I fully support the publication of this MS.

We would like to thank the reviewer for his/her very insightful comments in the 1st round of revisions to strengthen our study and now for supporting their publication.

Reviewer #3 (Remarks to the Author):

In the revised manuscript, the authors have performed several additional studies to address some of the concerns of this reviewer. Few major concerns, however, still remain unanswered.

- It is quite puzzling that renal YAP involvement on autophagy is not consistent between UUO and IR-induced renal injury according to new data. It is evident from previous studies that YAP upregulation plays a protective role in the context of acute kidney injury (PMID 6115662) while YAP induction plays a causative role in the context of UUO-driven fibrosis. These observations raise concerns regarding broad in vivo relevance of YAP involvement in renal autophagy during CKD or YAP involvement in autophagy is dependent on kidney disease etiology. Although authors provide extensive human studies linking YAP upregulation to autophagy in diabetic kidney disease, pertinent animal studies in diabetic nephropathy (most dominant cause of CKD) are still lacking which could clarify above confusion.

We agree with the reviewer that the divergent role of YAP in acute and chronic kidney disease (CKD) is indeed intriguing. However, this phenomenon is not an isolated case. Notably, molecules like YAP are not the only ones displaying such contrasting behavior. For instance, several works have established that the activation of the EGFR pathway is detrimental in CKD (Terzi et al., J Clin Invest 2000, PMID: 10903338; Lautrette et al., Nat Med 2005, PMID: 16041383; Yang et al., Am J Physiol Renal Physiol 2012, PMID: 22169008), yet it promotes tubular regeneration following an ischemic injury (Humes et al., J Clin Invest 1989, PMID: 2592559; Chen et al., Kidney Int 2012, PMID: 22418982). Similar dual effects have been observed for Lcn2 (Mori et al., J Clin Invest 2005, PMID: 15711640; Viau et al., J Clin Invest 2010, PMID: 20921623; El Karoui et al., Nat Commun 2016, PMID: 26787103). This intriguing phenomenon can be explained by the fact that the same cellular events triggered by these molecules, including YAP, yield different outcomes depending on the specific pathological context. For instance, cell proliferation is advantageous in replacing dead cells after an acute injury, but it has been shown to promote fibrosis in the context of CKD (Yang et al., Nat Med 2010, PMID: 20436483; Lovisa et al., Nat Med 2015, PMID: 26236991; Grande et al., Nat Med 2015, PMID: 26236989).

We also agree with the reviewer that it would be interesting to study the function of YAP in experimental models of diabetic nephropathy. However, the reviewer will agree that it is even more compelling to establish the relevance of our experimental findings to humans. With access to a unique human bio-bank of kidney biopsies from diabetic patients, we considered imperative to investigate the interplay between YAP and autophagy in this specific context. Remarkably, our results validate our in vitro observations and show that increased nuclear localization of YAP correlates with decreased fusion of autophagosomes with acidic compartments, suggesting a blockade of the autophagy flux in DKD. That said, we would like to mention, as already quoted in the text, that the team led by Raymond Harris has already elegantly demonstrated that specific deletion of Yap in renal tubular cells improves interstitial fibrosis in a model of diabetic

nephropathy in mice (Chen et al., Diabetes 2020, PMID: 32843569). These collective findings underscore the significant role of YAP in kidney disease, particularly in the context of DKD, in both experimental models and human patients.

• IF data in panels a and c of Figure 2 showed the cells with nuclei of different sizes. Are these images taken at same magnification?

These images were taken with the same objective (63X) but the reviewer is right, after processing the images, a small variation in image cropping has been done, explaining why the scale bars were a bit different. We have now adjusted this (Figure 2a,c). It is also worth mentioning that we have previously shown that 4 days of shear stress decreases cell size (and nuclei) (Orhon et al, Nat Cell Biol, 2016; PMID: 27214279), explaining why some cells and nuclei may already appear smaller after 1 day of shear (compared to static condition).

Autophagic flux determination is mostly based on the punctated expression of LC3 protein. The quantification of puncta is not shown for these figures.

The quantification of puncta shown in figure 2a and c are shown in figure 2b and 2d.

• YAP involvement in shear stress is well studied in the context of various diseases, but these studies are not cited. For example.

1: Lee, H., Diaz, M., Price, K. et al. Fluid shear stress activates YAP1 to promote cancer cell motility. Nat Commun 8, 14122 (2017). <https://doi.org/10.1038/ncomms14122>

2. Lee, H. J., Ewere, A., Diaz, M. F., & Wenzel, P. L. (2018). TAZ responds to fluid shear stress to regulate the cell cycle.

3. Yu, H., He, J., Su, G., Wang, Y., Fang, F., Yang, W., Gu, K., Fu, N., Wang, Y., Shen, Y., & Liu, X. (2021). Fluid shear stress activates YAP to promote epithelial-mesenchymal transition in hepatocellular carcinoma. Molecular oncology, 15(11), 3164–3183. <https://doi.org/10.1002/1878-0261.13061>

4. Kim KM, Choi YJ, Hwang J-H, Kim AR, Cho HJ, Hwang ES, et al. (2014) Shear Stress Induced by an Interstitial Level of Slow Flow Increases the Osteogenic Differentiation of Mesenchymal Stem Cells through TAZ Activation. PLoS ONE 9(3): e92427. <https://doi.org/10.1371/journal.pone.0092427>

5. Wang KC, Yeh YT, Nguyen P, Limqueco E, Lopez J, Thorossian S, Guan KL, Li YJ, Chien S. Flow-dependent YAP/TAZ activities regulate endothelial phenotypes and atherosclerosis. Proc Natl Acad Sci U S A. 2016 Oct 11;113(41):11525-11530. doi: 10.1073/pnas.1613121113.

The 1st reference was already cited in our manuscript (Ref 32 in the previous version) but we have now cited the other requested references (page 4).

Reviewer #4 (Remarks to the Author):

The author have fully addressed my concerns.

We would like to thank the reviewer for his/her help to strengthen our data.

REVIEWERS' COMMENTS

Reviewer #3 (Remarks to the Author):

I have no additional questions for the authors at this stage.

Reviewer #5 (Remarks to the Author):

This is an extensive study with a ton of work by the authors to support their hypothesis. Energy depletion induces AMP-LKB1-AMPK axis can promote autophagy through inhibiting mTORC1 activity or directly regulating ULK1 complex, this is well known pathway in regulating autophagy. This study identified a novel pathway that LKB1-AMPK regulates YAP S61 phosphorylation, thus inhibiting autophagy by transcription of Rubicon. I only have some minor suggestions hope can improve the manuscript.

The reviewers have asked many questions, especially reviewer #1, on the YAP regulates autophagy. I also have another question on this point. The authors claim that "inhibiting AMPK activity with dorsomorphin can increase nuclear YAP and alongside the inhibition of lipophagy". Why did they mention lipophagy here but the whole MS is about autophagy? They also can't conclude that dorsomorphin inhibits autophagy through YAP as they can't exclude AMPK's roles in regulating autophagy. LKB1-AMPK regulation of autophagy needs to be discussed.

Some figures need more work. WB in Fig 6i, the input and IP usually run together to indicate the WB works with input, more strict controls like IgG and/or blank bead should be used. The band for Ac-K in input not sharp and clear.

Fig7a, bright field light and LC3-RFP, and Wt1b-GFP fluorescence with lower magnification pictures should be re-taken to show the zebrafish embryo and pronephros morphology/structure.

Fig 7e, the staining of YAP on UUO kidneys look not specific to me, the authors can try looking at other group published YAP stain on UUO kidneys with the same antibody at <https://insight.jci.org/articles/view/146243/figure/2>. Or they can try different YAP antibody from Novus (Nb11058358) as this paper <https://pubmed.ncbi.nlm.nih.gov/26015453/#&gid=article-figures&pid=figure-1-uid-0>, this antibody both works for human and mouse tissues.

REVIEWER COMMENTS

Reviewer #3 (Remarks to the Author):

I have no additional questions for the authors at this stage.

We would like to thank the reviewer for his/her very insightful comments in the 1st and 2nd round of revisions to strengthen our study.

Reviewer #5 (Remarks to the Author):

This is an extensive study with a ton of work by the authors to support their hypothesis. Energy depletion induces AMP-LKB1-AMPK axis can promote autophagy through inhibiting mTORC1 activity or directly regulating ULK1 complex, this is well known pathway in regulating autophagy. This study identified a novel pathway that LKB1-AMPK regulates YAP S61 phosphorylation, thus inhibiting autophagy by transcription of Rubicon. I only have some minor suggestions hope can improve the manuscript.

The reviewers have asked many questions, especially reviewer #1, on the YAP regulates autophagy. I also have another question on this point. The authors claim that “inhibiting AMPK activity with dorsomorphin can increase nuclear YAP and alongside the inhibition of lipophagy”. Why did they mention lipophagy here but the whole MS is about autophagy? They also can't conclude that dorsomorphin inhibits autophagy through YAP as they can't exclude AMPK's roles in regulating autophagy. LKB1-AMPK regulation of autophagy needs to be discussed.

In a previous study we showed that shear stress induces lipophagy (Miceli et al 2020 Nat Cell Biol). We agree with the reviewer that the present study is not focused on lipophagy. However, in Figure 5g,h, we showed an accumulation of lipid droplets when YAP is retained in the nucleus under shear stress. In a broader picture, it would be important in future studies to delineate the respective role of lipophagy and bulk autophagy in the physiological adaptation of kidney epithelial cells to shear stress. However, this is beyond the scope of the present manuscript. According to the referee's comment, we have therefore modified the text (see page 17) and we have decided to replace the word “lipophagy” by “autophagy” in our model (Supplementary Fig.11).

As now mentioned in the discussion, the LKB1-AMPK controls autophagy in response to shear stress at two steps: the initiation step as previously shown (Orhon et al 2016 NCB, Miceli et al 2020 NCB) and at the maturation step by regulating the localization of YAP and consequently the expression of Rubicon (this manuscript). In Dorsomorphin-treated cells, YAP is mainly localized in the nucleus. Under these conditions, Rubicon is expressed. The inhibition of autophagy by dorsomorphin is a consequence of AMPK inhibition. Since AMPK promotes the initiation of autophagy, it is not possible with the use of dorsomorphin to observe an effect only on the maturation step.

Some figures need more work. WB in Fig 6i, the input and IP usually run together to indicate the WB works with input, more strict controls like IgG and/or blank bead should be used. The band for Ac-K in input not sharp and clear.

First, it is worth mentioning that this IP experiment is extremely difficult to perform in shear condition because cells are seeded (at $5 \cdot 10^5$ cells per microslide) in a very small and confined environment in which we cannot harvest them by scraping. For this technique, we have to use a huge number of microslides to harvest enough cells. In addition, it is also important to mention that this IP experiment provided in the first round of reviewing (see point 3 referee #3) was never asked by any referees. We thus propose to remove this piece of data if needed, and this proposition would not change our previous rebuttal letter since our response (for point 3 referee #3) was mainly based on (i) WB experiments on in vivo samples and (ii) new luciferase experiments. On the top, it is worth noting that referee #3 does not have anymore “additional questions for the authors” (as mentioned previously) and therefore supports this manuscript for publication.

Fig7a, bright field light and LC3-RFP, and Wt1b-GFP fluorescence with lower magnification pictures should be re-taken to show the zebrafish embryo and pronephros morphology/structure.

We now provide lower magnification pictures (see Supplementary figure 8). The reviewer and readers should now appreciate the morphology/structure of the zebrafish embryo and pronephros.

Fig 7e, the staining of YAP on UUO kidneys look not specific to me, the authors can try looking at other group published YAP stain on UUO kidneys with the same antibody at <https://insight.jci.org/articles/view/146243/figure/2>. Or they can try different YAP antibody from Novus (Nb11058358) as this paper <https://pubmed.ncbi.nlm.nih.gov/26015453/#&gid=article-figures&pid=figure-1-uid-0>, this antibody both works for human and mouse tissues.

We respectfully disagree with the reviewer. To convince the reviewer, we now provide new pictures (only for the reviewer, figure R1) showing YAP labeling (using YAP antibody from Cell Signaling Technology, #14074) in $Yap^{\Delta Tub}$ and $Yap^{fl/fl}$ kidneys respectively. The reviewer will now appreciate the specificity of our YAP antibody. Nevertheless, we slightly changed the background of the YAP staining shown on Figure 7e to better show the specific signal. The reviewer and readers should now better see the specific YAP staining.

Figure R1 : Specific YAP staining on renal tubules from *Yap^{flox}* (*Yap flox*) mice
Representative images of immunohistochemistry analysis on *Ksp/CreERT2-Yap^{flox}* mice (*Yap^{Δtub}*) compared to control *Yap^{flox}* (***Yapflox***) mice using anti-YAP antibody (Cell Signaling Technology, #14074).